

# Hyperbolic compactification of M-theory and de Sitter quantum gravity

**Giuseppe Bruno De Luca[1], Eva Silverstein[1] and Gonzalo Torroba[2]**

**1** Stanford Institute for Theoretical Physics, Stanford University, Stanford, CA 94306, USA
**2** Centro Atómico Bariloche and CONICET, Bariloche, Argentina

## Abstract

We present a mechanism for accelerated expansion of the universe in the generic case of negative-curvature compactifications of M-theory, with minimal ingredients. M-theory on a hyperbolic manifold with small closed geodesics supporting Casimir energy – along with a single classical source (7-form flux) – contains an immediate 3-term structure for volume stabilization at positive potential energy. Hyperbolic manifolds are well-studied mathematically, with an important rigidity property at fixed volume. They and their Dehn fillings to more general Einstein spaces exhibit explicit discrete parameters that yield small closed geodesics supporting Casimir energy. The off-shell effective potential derived by M. Douglas incorporates the warped product structure via the constraints of general relativity, screening negative energy. Analyzing the fields sourced by the localized Casimir energy and the available discrete choices of manifolds and fluxes, we find a regime where the net curvature, Casimir energy, and flux compete at large radius and stabilize the volume. Further metric and form field deformations are highly constrained by hyperbolic rigidity and warping effects, leading to calculations giving strong indications of a positive Hessian, and residual tadpoles are small. We test this via explicit back reacted solutions and perturbations in patches including the Dehn filling regions, initiate a neural network study of further aspects of the internal fields, and derive a Maldacena-Nunez style no-go theorem for Anti-de Sitter extrema for a range of parameters. A simple generalization incorporating 4-form flux produces axion monodromy inflation. As a relatively simple de Sitter uplift of the large-N M2-brane theory, the construction applies to de Sitter holography as well as to cosmological modeling, and introduces new connections between mathematics and the physics of string/M theory compactifications.

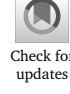

# 1   Introduction, motivation and summary of results

The observational discovery of the accelerated expansion of the universe demands a much more complete theoretical understanding. This requires a full formulation in quantum gravity, along with its implications for cosmological observables. The accelerated expansion in $\Lambda CDM$ cosmology is well modeled phenomenologically by a cosmological constant at late times and by a range of viable inflationary dynamics in the early universe. Inflationary cosmology is ultraviolet-sensitive, and its modeling extends to quantum gravity via string theoretic realizations of metastable de Sitter and inflation, some with novel features and testable signatures [1–6]. These models also illuminate the more abstract problem of formulating quantum gravity in cosmology.[1] There is clearly much more to learn about the structure of string theory and cosmological observables, building from the existing results.

In this work, we will introduce a new class of models of four dimensional de Sitter and accelerated expansion in M theory based on negatively curved internal geometry.[2] We are motivated in part by the typicality of negative curvature among Riemannian manifolds, there being infinitely many topologies at a fixed dimensionality. Although more generic in that sense than most previously studied compactifications, this setup also enjoys several simplifying features. The hyperbolic metric is known explicitly and the tree level potential is positive.[3] This combined with the automatically generated Casimir energy and 7-form flux $F_7 = dC_6$ provides a simple power-law volume stabilization mechanism. The crucial negative contribution from the Casimir energy is readily tunable to compete with the two classical contributions via discrete choices of finite-volume hyperbolic manifolds. In particular, we find that the net integrated curvature including variations of the warp and conformal factors can be tuned small enough to enable the quantum Casimir energy to compete in the volume stabilization mechanism, illustrating this with back reacted solutions in large patches of the space. Moreover, negative curvature spaces enjoy rigidity properties that enhance stability [14–16], as does the warped product structure [17, 18].

The Casimir contribution arises from small circles in relatively localized regions of the internal manifold. Two concrete examples of this are the Einstein spaces obtained from a higher-dimensional analogue of Dehn filled cusps [19][4] and the 'inbreeding' construction [20–22]. The localized support of the Casimir energy requires a detailed analysis of the effect on internal fields, including a warp factor and perturbations away from the fiducial hyperbolic metric $g_{\mathbb{H}}$:

$$ds^2 = e^{2A(y)}ds^2_{dS_4} + e^{2B(y)}(g_{\mathbb{H}ij} + h_{ij})dy^i dy^j . \tag{1.1}$$

We will make use of Douglas' elegant formulation of a well defined off-shell effective potential $V_{eff}[B,h,C_6]$ [17],[5] incorporating the dominant one-loop Casimir stress-energy tensor $T^{(\text{Cas})M}{}_N$. This is a functional of the internal fields obtained by solving the internal part of the constraint equation

$$\delta_A \mathcal{S}_{11,classical} = -\langle T^{(\text{Cas})\mu}{}_\mu \rangle , \tag{1.2}$$

---

[1]For example, the metastable structure of string-theoretic de Sitter as an uplift of AdS/CFT provides a striking consistency check on de Sitter and FRW holography [7–9] along with concrete microphysical entropy counts such as [10].

[2]See also [11] for previous AdS constructions.

[3]Useful properties of negatively curved internal spaces were exploited in earlier works such as [12] and [13].

[4]A cusp is a region covered by a metric of the form (4.1) with a shrinking 6-torus; a filling consists of a smooth cigar geometry in which one cycle of the 6-torus shrinks smoothly and the remaining torus stays finite. See appendix A.2 for a quick summary.

[5]For other work in this direction see e.g. [23] and [24].

for the warp factor $u = e^{2A}$, with a finite four dimensional Newton constant:

$$\frac{1}{G_N} = \frac{1}{\ell_{11}^9} \int d^7 y \, \sqrt{\det(g_{\mathbb{H}} + h)} e^{7B} e^{2A}. \tag{1.3}$$

The formulation [17] and appropriate generalizations will be useful for analyzing the background fields, their second order perturbations, and more general field configurations including inflationary ones.

In the region of the localized Casimir energy, the screening effect of the warp factor alone [17] stabilizes the naïve instability toward arbitrarily small Casimir circle. The Casimir energy sources a particular profile of warp and conformal factors in the bulk, dressing the underlying hyperbolic metric. We find a backreacted solution in the region of a cusp which illustrates the stability of the Casimir circle and the expected warp and conformal factor variation, along with other features, at the level of radial variation in that region. Manifolds such as those in [25] contain a substantial fraction of their volume in cusps.

Second order metric variations are highly constrained by the rigidity properties of negatively curved Einstein spaces [14–16] and effects of the warp factor. For dimension $n > 2$, finite volume hyperbolic $n$-manifolds do not come in continuous families, and there is a positive gapped Hessian for $-\int \sqrt{g}R$ deformed along modes orthogonal to the conformal factor. On the latter, a key effect derived in [17] shows explicitly that the warp factor $e^{2A}$ serves to stabilize conformal mode fluctuations [17]. We generalize this to obtain a model-independent statement of the positive warping contribution to the Hessian, expanding near a small cosmological constant. Using this and related methods, we argue for a positive Hessian overall expanding around the dressed hyperbolic metric, and present a nontrivial test of this expanding around the backreacted cusp solution although we stop short of a full calculation of the Hessian in our problem. Our mechanism requires a certain parameter regime for integrated quantities, one which we find to be readily available by working with known classes of hyperbolic manifolds such as [25–27], summarized in appendix A.

Moreover, via a simple comparison of two integrated combinations of the equations of motion, we derive a Maldacena-Nunez style no-go theorem for *Anti*-de Sitter extrema for a range of parameters. Altogether these results indicate metastable de Sitter solutions, including examples fairly close to our fiducial hyperbolic metric in the bulk of the internal space. Further, the homology of suitable hyperbolic manifolds supports a simple generalization to axion monodromy inflation [28–32] (along the lines of [33] or [34] but with fewer ingredients). We summarize the mechanism and spatial structure of our compactifications in figures 1-2.

It is interesting to explore the internal fields as explicitly as possible, both to concretely study the back-reaction coming from the localized Casimir energy, and to characterize the landscape arising from more general field configurations for these topologies. For this, we analyze the equations of motion,

$$\frac{\delta V_{eff}}{\delta B} = 0 = \frac{\delta V_{eff}}{\delta h} = \frac{\delta V_{eff}}{\delta C_6}, \tag{1.4}$$

with the constraint equation $\delta_A \mathcal{S}_{11} + T^{(Cas)} = 0$ determining $A$ as a functional of $B, h$, and $C_6$. Including the constraint, this constitutes a set of nonlinear partial differential equations (PDEs) for deformations $\{A(y), B(y), h(y), C_6\}$, including deviations from the fiducial hyperbolic metric (1.1). This problem is well posed, as many concrete examples of appropriate finite volume hyperbolic spaces are known explicitly [15,25–27]. One way such spaces can be characterized is as a joined set of hyperbolic polygons [15]. This gives an explicit formulation of the PDEs and boundary conditions.

In this setup, we can similarly characterize regimes of more general accelerated expansion as well as metastable de Sitter. Approximate solutions of the PDEs (1.4) correspond to more

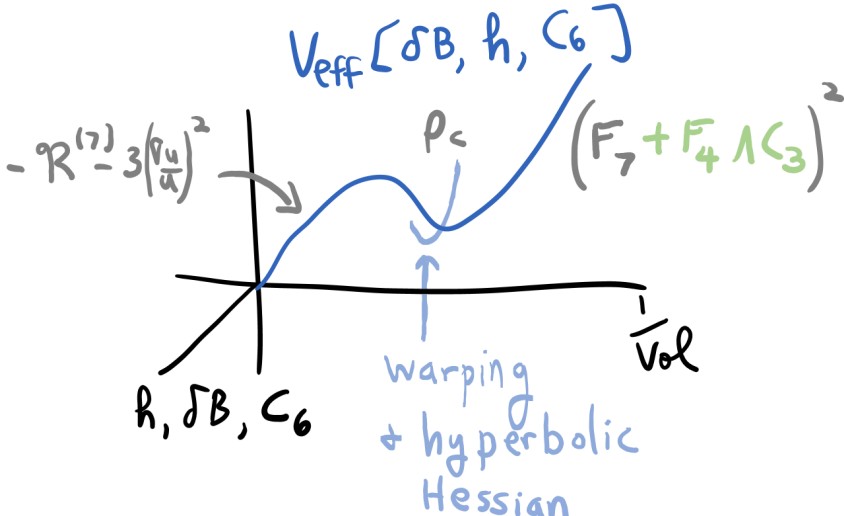

Figure 1: Schematic of our stabilization mechanism. The overall volume is stabilized by the indicated three-term mechanism involving curvature (including warp factor gradients), Casimir energy, and flux, in a regime of large radius control (5.88). The additional directions in 4$d$ field space are stabilized via a combination of hyperbolic rigidity and warping effects. Near the small circles supporting a strong negative contribution to $V_{eff}$ from Casimir energy, the exponential decay of the warp factor $u = e^{2A}$ prevents a runaway instability toward larger $|\rho_C|$ and stabilizes the circle size $R_{c*}$ as discussed around (5.27). The integrated gradient squared of the warp factor enters in combination with the integrated internal curvature, but with opposite sign, to yield a tunable leading positive term in the potential, parameterized by $a \ll 1$ (4.6). Hyperbolic manifolds exhibit a positive Hessian for deformations $h$ in (1.1) [16], and warping effects render the net Hessian for the conformal mode $B$ positive along the lines explained in [17]. These effects extend to our system as described in §3.1, §5.4 suggesting a positive Hessian overall, although a detailed calculation of the Hessian in our problem is beyond the scope of this paper. In the absence of four-form flux, the potential $C_6$ relaxes to the solution (3.10)(3.11). If we include quantized $F_4$ fluxes, the generalized flux term produces multifield axion monodromy inflation as explained in §7. The features required for our construction exist in concrete classes of hyperbolic manifolds as described in §A. Although the quantum effect of Casimir energy is a leading contribution to the inflationary and de Sitter potential, quantum corrections to other quantities such as the 4$d$ Newton constant are suppressed as explained in §4.2.

general accelerated expansion. The slow roll parameters (given in terms of the canonically normalized fields $\phi_{cI}$)

$$\varepsilon_V = \frac{1}{2} \sum_I \left( \frac{\partial_{\phi_{cI}} V_{eff}}{V_{eff}} \right)^2 \frac{1}{G_N}, \quad \eta_{V,IJ} = \frac{\partial_{\phi_{cI}} \partial_{\phi_{cJ}} V_{eff}}{V_{eff}} \frac{1}{G_N} \tag{1.5}$$

are proportional to the PDEs and their derivatives. We derive an explicit functional corresponding to $\varepsilon_V$ in our landscape, and comment on applications.

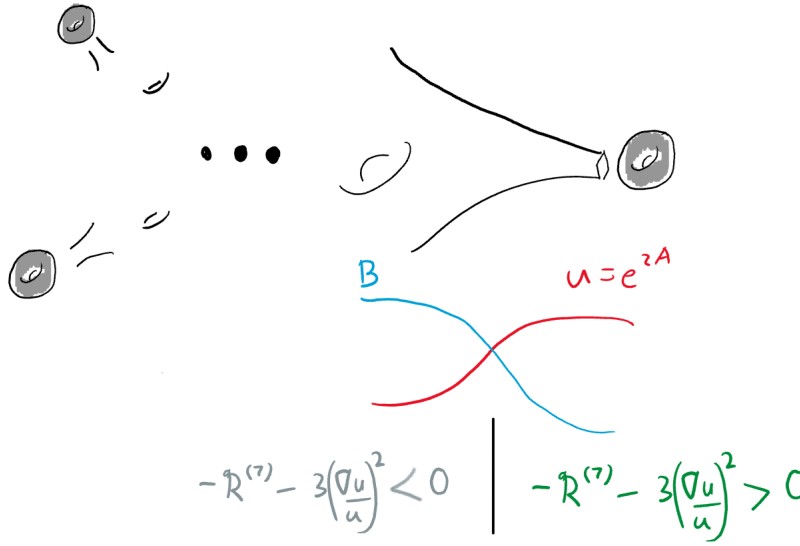

Figure 2: Schematic of the spatial structure of our compactification. Casimir energy is concentrated in regions with small circles, e.g. filled [19] cusps as pictured here, or other types of systoles [20] ( the *systole* of a manifold being its shortest closed noncontractible geodesic). The constraint equation (3.7) is related to a Schrödinger problem. The solution contains gradients of the analogue wavefunction, the warp factor $u = e^{2A}$, as discussed e.g. in §5.1.1-5.1.2, and the conformal factor $B$ also varies internally, enabling a small value of the net curvature term in the effective potential. We schematically sketch the hyperbolic space dressed with the varying warp and conformal factors; explicit solutions illustrating the back reacted filled cusp geometry appear in section 5.3. The direction of variation of the deformation $B$ away from the hyperbolic space arises because the negative curvature pushes outward on the volume, whereas the Casimir energy contributes a force in the opposite direction. The effective Schrödinger potential develops a potential barrier for sufficiently large Casimir energy. The warp factor dynamics also keeps the Casimir circle size $\gg$ the eleven dimensional Planck length $\ell_{11}$ since the wavefunction dies rapidly under the Schrödinger potential barrier $\sim |\rho_C|$ if the latter gets too large. The discrete parameters determining the volume and the size of the filled region enable independent variation of the integrated curvature and gradient terms as discussed in §4, §5.1, and appendix A, which details explicit classes of hyperbolic manifolds which satisfy our requirements (including more precise illustrations in the upper half space). The leading term in the effective potential $\sim \int \sqrt{g^{(7)}} u^2 (-R^{(7)} - 3(\frac{\nabla u}{u})^2)$ is tuned to small positive value via these choices, enabling the quantum Casimir energy to compete.

In studying the internal dynamics, we focus on analytic methods, but also include some numerics. We will touch on neural network techniques for solving PDEs and exploring the potential landscape $V_{eff}$ [17], including a novel approach using the slow roll functionals as loss functions to minimize in machine learning. The concreteness of hyperbolic manifolds yields a number of well-posed problems in this area that could benefit from a much more extensive study of the internal fields in the landscape of hyperbolic compactifications of M theory.

After explaining our framework in the bulk of the paper, we will comment on other future directions and applications of our results to aspects of observational cosmology and de Sitter quantum gravity, along with connections to mathematics.

## 2 Setup and volume stabilization mechanism

We compactify M theory on a manifold admitting a finite-volume hyperbolic metric, obtained as a freely acting orbifold $\mathbb{H}_7/\Gamma$ with constant curvature radius $\ell$. In upper half space coordinates, we can write the $\mathbb{H}_7$ metric as

$$ds^2_{\mathbb{H}_7} = \ell^2 \frac{dz^2 + ds^2_{\mathbb{R}^6}}{z^2} = dy^2 + e^{-2y/\ell} ds^2_{\mathbb{R}^6} \,. \tag{2.1}$$

Each example in our class of finite-volume compactifications is obtained by modding this out by a freely acting group $\Gamma$ of isometries [15], with specific features needed in our application.

The main specification we will need is one or more regions in the geometry with small circles of slowly varying size $R_c \ll \ell$, as occurs in hyperbolic cusps (combined with a higher dimensional analogue of Dehn filling [19]) or in other constructions in systolic geometry[6] such as [20–22]. We will describe this in more detail as we go, connecting them with standard features of hyperbolic spaces [15]. To be specific we will realize them in a class of examples obtained from the simple and explicit constructions introduced recently in [25].

Including a warp factor and allowing for deformations away from the hyperbolic metric yields the metric (1.1) appropriate for seeking four dimensional de Sitter solutions. There is no additional dilaton field (as would arise in perturbative string theory). We also introduce $N_7$ units of 7-form flux, and will take into account Casimir stress-energy generated at the quantum level. This will be straightforward to calculate in regions of the manifold with slowly varying circles. As we will review in detail below in §4, the Casimir energy $\rho_C$ is negative for suitable fermion boundary conditions, and in the region of small circle size $R_c$ it behaves like $\rho_C \sim -\frac{1}{R_c^{11}}$ as an 11d energy density. So its averaged contribution will be $\sim -\frac{1}{R_c^{11}} \frac{\mathrm{Vol}_C}{\mathrm{Vol}_7}$ where $\mathrm{Vol}_C$ is the volume over which the Casimir energy has its leading support, with $\mathrm{Vol}_7$ the full internal volume. The bosonic part of the 11d (super)gravity limit of M-theory including the classical sources is described by the action

$$\mathcal{S}^{(classical)}_{11(\mathrm{bosonic})} = \frac{1}{\ell_{11}^9} \int d^{11}x \sqrt{-g^{(11)}} \left( R^{(11)} - \frac{1}{2} |F_7|^2 \right) + S_{CS} \,, \tag{2.2}$$

in $(- + \ldots +)$ signature, where $S_{CS}$ is a Chern-Simons term that will not play a direct role in our de Sitter construction. Around a background configuration, quantum fluctuations of field perturbations will build up a quantum Casimir stress-energy which we will specify below. This will enter into the four dimensional effective theory in an important way.

The hyperbolic geometry, as an Einstein space, is an extremum of the Einstein-Hilbert action $\int_{11d} \sqrt{-g^{(11)}} R^{(11)}/\ell_{11}^9$ for all metric deformations except the overall volume direction (equivalently the curvature radius $\ell$), which would run away to large radius in the absence of stress energy sources. We will start by showing that the averaged effect of the flux and Casimir contributions gives a potential of the right shape to stabilize this would-be runaway direction. This may happen in a regime of large-radius control, given a large parameter in the problem which enables the Casimir energy to compete with the classical sources. We will obtain this control parameter in detail, once we incorporate the warping effects [17] required for a complete treatment of compactification. In that framework, we will analyze

---

[6]the *systole* of a manifold being the shortest closed noncontractible geodesic

and bound the effects of the inhomogeneity in the Casimir energy, finding strong indications of overall meta-stability from the combined effects of warping [17] and the Hessian from the internal curvature [16]. For this full problem, we will work extensively with the complete four dimensional effective potential $V_{eff}[B, h, C_6]$ derived in [17] (which we will introduce below in §3).

First, let us explain the basic motivation, a simple mechanism for volume stabilization. Dimensionally reducing (2.2) to four dimensions and taking into account the one-loop Casimir contribution, yields a four dimensional Einstein frame potential for the curvature radius – equivalently the volume – of the form[7]

$$
\begin{aligned}
V(\ell) \;\sim\;\; & M_4^4 \frac{1}{v_7 \hat{\ell}^7} \left( \frac{a}{\hat{\ell}^2} + \frac{\int_{\mathbb{H}_7/\Gamma} d^7 y \sqrt{g_{\mathbb{H}}} \rho_C \ell_{11}^4}{v_7 \hat{\ell}^7} + \frac{N_7^2}{v_7^2 \hat{\ell}^{14}} \right) + \text{warping} + \text{inhomogeneities} \\
=\;\; & M_4^4 \frac{1}{v_7 \hat{\ell}^7} \left( \frac{a}{\hat{\ell}^2} - \frac{K}{\hat{\ell}^{11}} + \frac{N_7^2}{v_7^2 \hat{\ell}^{14}} \right) + \text{warping} + \text{inhomogeneities} ,
\end{aligned}
\tag{2.3}
$$

where

$$
M_4{}^2 \sim v_7 \frac{\ell^7}{\ell_{11}^9} = v_7 \frac{\hat{\ell}^7}{\ell_{11}^2}
\tag{2.4}
$$

is the four dimensional Planck mass. Here we define $\hat{\ell} \equiv \ell/\ell_{11}$ in terms of the eleven dimensional Planck length $\ell_{11}$, and define $v_7$ by

$$
\text{Vol}(\mathbb{H}_7/\Gamma) = v_7 \ell^7 = \text{Vol}_7 ,
\tag{2.5}
$$

and

$$
K \sim \left( \frac{\ell}{R_c} \right)^{11} \frac{\text{Vol}_C}{v_7 \ell^7} ,
\tag{2.6}
$$

where as defined above $\text{Vol}_C$ represents the internal volume over which the Casimir energy density has its leading support $\sim -1/R_c^{11}$.[8] The first term in (2.3) descends from the 11d Einstein-Hilbert action. With no backreaction, $a = 42$, but we will see in §5 that effects from warping and inhomogeneities allow us to get $a \ll 1$. This will play an important role in the stabilization mechanism, helping to increase the relative importance of the Casimir term. The third term is the quantized 7-form flux squared descending from the 11d $C_6$ kinetic term. The middle term arises from the aforementioned Casimir energy whose contribution we will discuss in detail in §4-5. We note that the 11d supergravity fields that give rise to the Casimir term also induce other quantum corrections, such as corrections to $1/G_N$ and higher derivative terms. But as we will explain in detail below, these are suppressed by higher powers of $\ell_{11}/R_c$ and $\ell_{11}/\ell$, which will be controllably small in our setup.

We will later treat the warping and inhomogeneity effects in the potential, finding them consistent with the volume stabilization mechanism suggested by the first three terms with appropriate tunes to obtain large radius control. The first step is to note that the shape of the first 3 terms in the potential (2.3) is consistent with a metastable minimum at positive potential, given sufficiently large values of $K$ and $N_7^2$ scaling like

$$
\frac{N_7^2}{v_7^2} \sim \frac{K^{4/3}}{a^{1/3}} .
\tag{2.7}
$$

---

[7]We use a positive sign for the Casimir term, noting that the Casimir energy density $\rho_C$ for bosons (which dominate due to the fermion boundary conditions) is negative.

[8]Note that both $K$ and $v_7$ are dimensionless as defined in (2.3) and (2.4).

A large flux quantum number is straightforward to prescribe. For the quantum Casimir energy to compete with the other terms will require an input large number. If we were to balance the hyperbolic curvature term against the Casimir term in (2.3), this requires

$$\frac{a}{\ell_{11}^9 \ell^2} \sim \frac{1}{R_c^{11}} \frac{\text{Vol}_C}{v_7 \ell^7}. \tag{2.8}$$

If $K/a$ in (2.3) and (2.6) is fixed and large and positive – corresponding to net negative Casimir energy – then there is a minimum for $\ell$ at large radius compared to $\ell_{11}$:

$$\hat{\ell} = \frac{\ell}{\ell_{11}} \sim \left(\frac{K}{a}\right)^{1/9} \gg 1. \tag{2.9}$$

This enhancement of the negative Casimir energy such that is competes with the term of order $\frac{a}{\ell^2 \ell_{11}^9}$ with $a \ll 1$ will be readily available in concrete hyperbolic manifolds with small cycles $R_c \ll \ell$, consistently with $R_c \gg \ell_{11}$. The latter criterion $R_c \gg \ell_{11}$ avoids potential instabilities from the M-branes arising in the UV completion of 11d supergravity. It is worth stressing that since the curvature radius $\ell$ is much larger than the small circle size $R_c$, this criterion is not needed for weak curvature but it does eliminate light degrees of freedom from wrapped branes.

In a similar local geometry studied in [35], wrapped strings cause an instability when a Scherk-Schwarz circle size reaches the string scale. This is analogous to the potential instabilities that could occur if $R_c$ reached the $\ell_{11}$ scale. If the circle reaches the scale where the wrapped extended object condenses, the end result may be, as in [35], a solution capped off at a scale $\ell$. As explained in detail in [35], this transpires via a process where first the wrapped string condenses, removing the region with small $R_c$ to produce a thin cigar geometry, and then the geometry retracts to a cigar geometry with a curvature scale $\sim \ell$. The latter geometry is analogous to the cigar geometry in a Euclidean black hole; since it contains no structure at a scale smaller than $\ell$, it would not support strong Casimir energy. [9] However, this wrapped string/brane instability does not set in until the circle size reaches the fundamental scale. Conversely, if the circle is large in Planck units, the process is a Euclidean quantum gravity effect, the Witten bubble [36], which is exponentially suppressed in $R_c^2/\ell_{11}^2$. Our mechanism, including key effects of the warping, will meta-stabilize $R_c \gg \ell_{11}$, so that any instability of this sort is a negligible non-perturbative effect.

A typical potential shape is depicted in Fig. 3. The squared 4$d$ Hubble constant proportional to the value of the potential at the minimum scales like $1/\ell^2$ with no further tuning (since all terms are then comparable, including the internal curvature). Further tuning is available via our parameters $K$ and $N_7$. Altogether we have a small 4$d$ curvature

$$H_{dS}^2 \sim \frac{V_{min}}{M_4^4} \le \frac{1}{\ell^2} \ll \frac{1}{\ell_{11}^2} \ll M_4^2, \tag{2.10}$$

in units of both the eleven and four dimensional Planck scales. The final inequality here arises from the large internal volume.

Finally, let us stress again that in motivating our setup in this section, we have discussed the homogeneous terms in the expression (2.3) for the potential $V^{(0)}$ that arises from the averaged sources. However, it will be important to take into account effects from warping and inhomogeneities, and we will do this in our more complete treatment of the full potential $V_{eff}$ in what follows. This will lead us to an approximation scheme in which the required strong localized Casimir contribution is stabilized by warping effects, and for bulk fields the metastable minimum shifts by a controllably small amount as a result of tadpoles introduced by the inhomogeneity.

---

[9]We thank Juan Maldacena for discussions related to this point.

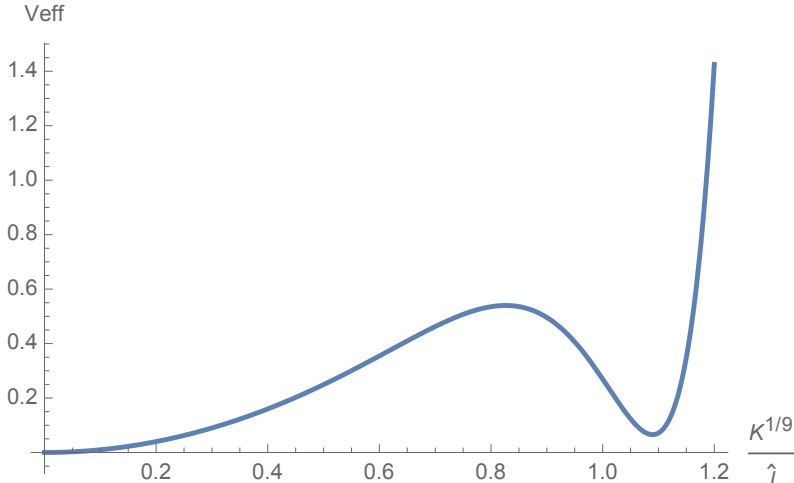

Figure 3: Effective potential with 3-term structure from (2.3). The intermediate negative Casimir contribution balances against curvature (dominant at small $1/\hat{\ell}$), and flux. This produces a dS minimum according to the averaged effects of our stress energy sources, We will incorporate inhomogeneities and warping effects in later sections and find that this volume-stabilization structure persists, with a strong localized Casimir source and controllably small shifts in the bulk fields.

# 3 Off shell effective potential and slow roll functionals

We will now incorporate the full effective potential, which captures the warping and inhomogeneity effects indicated schematically above in (2.3). This is presented in equations (3.1-3.2) of [17], which we reproduce here for our setup. Let us define the metric components as in (1.1),

$$ds^2 = g^{(4)}_{\mu\nu}dx^\mu dx^\nu + g^{(7)}_{ij}dy^i dy^j = e^{2A(y)}ds^2_{\text{symm}} + e^{2B(y)}\tilde{g}^{(7)}_{ij}dy^i dy^j, \tag{3.1}$$

with the metric $ds^2_{\text{symm}}$ being any maximally symmetric 4d metric, (A)dS$_4$ or Minkowski, with curvature $R^{(4)}_{\text{symm}}$. Furthermore, $\tilde{g}_{ij}$ is an arbitrary fiducial metric, although for part of our analysis we will be interested in fluctuations around a fiducial hyperbolic metric,

$$\tilde{g}^{(7)}_{ij} = g_{\mathbb{H}ij} + h_{ij}. \tag{3.2}$$

It is very useful for some purposes to work directly with the warp factor, defining the variable

$$u(y) = e^{2A(y)}. \tag{3.3}$$

In terms of this, the effective potential in four dimensions is (see App. B)

$$V_{eff}[g^{(7)}, C_6] = \frac{1}{2\ell^9_{11}}\int d^7y \sqrt{g^{(7)}}u^2|_c \left(-R^{(7)} - \frac{1}{4}\ell^9_{11}T^{(\text{Cas})\mu}_{\quad\mu} + \frac{1}{2}|F_7|^2 - 3\left(\frac{\nabla u}{u}\right)^2\Big|_c\right)$$
$$+ \frac{C}{2}\left(\frac{1}{G_N} - \frac{1}{\ell^9_{11}}\int \sqrt{g^{(7)}}u|_c\right), \tag{3.4}$$

where from (4.2),

$$\langle T^{(\text{Cas})\mu}_{\quad\mu}\rangle = -4\rho_C(R_c), \tag{3.5}$$

$C$ is a Lagrange multiplier enforcing a fixed 4d Newton constant

$$\frac{1}{G_N} = \frac{1}{\ell_{11}^9} \int \sqrt{g^{(7)}} u|_c \,, \tag{3.6}$$

$\nabla$ is the gradient with respect to $g_{ij}^{(7)}$, and $u|_c$ denotes the functional $u[g^{(7)}, C_6]$ which satisfies the constraint equation

$$\left(-\nabla^2 - \frac{1}{3}\left(-R^{(7)} - \frac{1}{4}\ell_{11}^9 T^{(\mathrm{Cas})\mu}{}_\mu + \frac{1}{2}|F_7|^2\right)\right) u = -\frac{C}{6} \,, \tag{3.7}$$

with appropriate boundary conditions, along with the normalization condition (3.6). For a hyperbolic manifold decomposed into hyperbolic polygons, the boundary conditions are continuity of $u$ and its normal derivative at the totally geodesic interfaces between polygons.

Upon solving for $u$ in this way, the value of $C$ is proportional to the potential (and hence the four dimensional curvature):

$$V_{eff} = \frac{C}{4G_N} = \frac{R_{\mathrm{symm}}^{(4)}}{4G_N} \,. \tag{3.8}$$

This expression for $V_{eff}$ is valid off-shell in the effective theory, and depends only on the internal fields. Setting to zero its variation with respect to the internal fields, while solving for $u$ via (3.7), reproduces the 11d field equations for $C_6$ and the metric (3.1). In App. B, we collect the equations of motion for $A$ (equivalently $u$), $B$, and $C_6$ around any fiducial geometry $\tilde{g}_{ij}^{(n)}$.

We can also write this in a way that makes contact with the standard Einstein frame 4d potential, by integrating out $C$:

$$V_{eff}[g^{(7)}, C_6] = \frac{\ell_{11}^9}{2G_N^2} \frac{\int d^7 y \sqrt{g^{(7)}} u^2|_c \left(\left[-R^{(7)} - 3\left(\frac{\nabla u}{u}\right)^2\big|_c\right] - \frac{1}{4}\ell_{11}^9 T^{(\mathrm{Cas})\mu}{}_\mu + \frac{1}{2}|F_7|^2\right)}{(\int d^7 y \sqrt{g^{(7)}} u|_c)^2} \,. \tag{3.9}$$

This has the overall volume scaling as in (2.3), with the addition of the $-3(\frac{\nabla u}{u})^2$ term which scales with $\ell$ like the internal curvature term.

The organization of these equations into the constraint equation and variations of the effective potential $V_{eff}$ has several practical advantages (in addition to its conceptual importance) [17]. For fixed internal metric and other sources, the constraint equation (3.7) can be understood in terms of a Schrödinger problem [17], with the analogue Schrödinger potential getting positive contributions from negative stress energy sources, and vice versa. Thus the analogue wavefunction $u$ gets its support in regions where positive stress energy sources dominate, and conversely it dies off rapidly where negative sources dominate, like a quantum mechanical wavefunction in a classically disallowed region. As such, it screens would-be unstable regions of negative energy, as we will see in detail in §5.1. At a perturbative level, [17] showed that the warp factor staunches a naïve second order instability in the direction of the conformal factor of the internal metric. We will generalize this shortly to derive a model-independent formula for this effect on the Hessian. A key restriction that we will take into account is that $u$ must be real non-negative in our problem.

Our system contains 7-form magnetic flux $F = dC_6$. It is useful to record here the equation of motion for $C_6$ and a simple solution $\bar{F}^{(7)}$ for the flux:

$$0 = \partial_{j_1}\left(\sqrt{g^{(7)}} u^2 g^{(7)j_1 i_1} \cdots g^{(7)j_7 i_7} F_{i_1,\dots,i_7}\right),$$

$$\bar{F}^{(7)}_{i_1,\dots,i_7} = f_0 \frac{\sqrt{g^{(7)}}}{u^2} \epsilon_{i_1,\dots,i_7} \,. \tag{3.10}$$

On a compact space we must impose flux quantization[10],

$$\frac{(2\pi)^{1/3}}{\ell_{11}^6} \int_{\Sigma_7} \bar{F}^{(7)} = 2\pi N_7 \Rightarrow f_0 \sim \ell_{11}^6 \frac{N_7}{\int_{\Sigma_7} \frac{\sqrt{g^{(7)}}}{u^2}}, \tag{3.11}$$

where $\Sigma_7$ is the internal manifold; see e.g. [37] where the key role of fluxes in string com-pactifications was articulated. With this in place, we note that the overall scale of $u = e^{2A}$, equivalently the zero mode of $A$, does not enter in $\bar{F}^{(7)}$. This is the general solution for $C_6$ given maximal symmetry in $4d$ and no magnetic $F_4$ flux internally.

Let us explain this statement. There is a unique 4-form flux that is closed and compatible with maximal symmetry:

$$F_{\mu_1 \ldots \mu_4}^{(4)} = f_0 \sqrt{g_{\text{symm}}^{(4)}} \epsilon_{\mu_1 \ldots \mu_4}, \tag{3.12}$$

with $f_0$ given by the functional (3.11) which respects the maximal symmetry and ensures flux quantization. Dualizing this gives the solution in (3.10).

It is also useful to consider the energetics directly in terms of the magnetic description. We want to compare two configurations:

$$\begin{aligned}
\bar{F}_{i_1,\ldots,i_7}^{(7)} &= \ell_{11}^6 \frac{N_7}{\int \sqrt{g^{(7)}} e^{-4A}} \sqrt{g^{(7)}} e^{-4A} \epsilon_{i_1,\ldots,i_7}, \\
\bar{F}_{i_1,\ldots,i_7}' &= \ell_{11}^6 \frac{N_7}{\int \sqrt{g^{(7)}}} \sqrt{g^{(7)}} \epsilon_{i_1,\ldots,i_7}.
\end{aligned} \tag{3.13}$$

The first one contributes an energy density

$$E = \int \sqrt{g^{(7)}} e^{4A} |\bar{F}^{(7)}|^2 \sim \frac{N_7^2}{\int \sqrt{g^{(7)}} e^{-4A}}. \tag{3.14}$$

So in fact regions with strong warping $u \to 0$ tend to decrease the contribution of this config-uration to the energy. For the second one,

$$E' \sim \frac{N_7^2 \int \sqrt{g^{(7)}} e^{4A}}{(\int \sqrt{g^{(7)}})^2}. \tag{3.15}$$

Thus

$$\frac{E'}{E} = \frac{1}{(\int \sqrt{g^{(7)}})^2} \left( \int \sqrt{g^{(7)}} e^{4A} \right) \left( \int \sqrt{g^{(7)}} e^{-4A} \right) \geq 1, \tag{3.16}$$

by Cauchy-Schwarz. So $\bar{F}_7$ is energetically preferred over the unwarped ansatz $\bar{F}_7'$.

If we work with the properly quantized flux solution (3.10) and (3.11) from the start, then the constraint equation (3.7) becomes a nonlinear integro-differential equation:

$$\hat{H}u = -\frac{C\ell^2}{6}, \quad \hat{H}[u] = -\nabla_w^2 - \frac{1}{3} \left( -R^{(7)}\ell^2 + \ell_{11}^9 \ell^2 \rho_C + \frac{\ell^2 \ell_{11}^{12}}{2} \frac{N_7^2}{u^4 (\int \sqrt{g^{(7)}} \frac{1}{u^2})^2} \right). \tag{3.17}$$

We stress that the $u$-dependence here is consistent with the screening effect just noted fol-lowing [17]. A rapidly decreasing 'wavefunction' $u$ in a classically disallowed region of the effective Schrödinger potential is consistent with finite flux in that region, as a result of the $\int \sqrt{g} \frac{1}{u^2}$ factor in the denominator of the final term in (3.17). For some purposes, it is useful to begin with a fixed flux configuration (independent of $u$) such as $\bar{F}_7'$ in (3.13), to maintain

---

[10]In the rest of the paper we are going to omit these factors of $2\pi$.

the linearity of the constraint equation, and later shift $C_6$ to its minimum. This also puts the $C_6$ perturbations in the same footing as the metric perturbations.

This formulation of an off-shell effective potential facilitates exploration of the landscape beyond the fiducial hyperbolic metric. Another application which we will develop below is to formulate a functional version of the inflationary slow-roll parameters related to variations of $V_{eff}$. As we will see, these expressions will enable useful analytic upper bounds on deviations from de Sitter in appropriate configurations in this landscape that can extend well beyond the hyperbolic geometry. Moreover, numerical approaches based on machine learning can naturally explore the $V_{eff}$ landscape as well as a loss landscape built from the squared equations of motion or the functional slow roll parameters, as we will see in §8.

In the remainder of this paper, we will first study our system starting from the fiducial hyperbolic metric, taking into account the inhomogeneities introduced by the localized Casimir energy in the solution and in the analysis of the Hessian, which we find to be positive overall. It is interesting to further analyze the internal field equations, in part to explore field configurations well beyond this nearby de Sitter minimum. This is a well posed problem given explicit hyperbolic manifolds obtained by gluing polygons. We will discuss a warmup example in this direction below, but largely leave it to future work. In general, our setup raises numerous directions for ongoing analysis.

## 3.1 Warping contributions to the Hessian

We can use this formalism to derive model-independent consequences for small fluctuations about a solution to the equations of motion with small cosmological constant. The result will be a universal stabilizing (positive) contribution to the Hessian arising directly from the warping dependence in $V_{eff}$ (3.4).

Motivated by the analogy to a Schrödinger problem, let us write

$$2\ell_{11}^9 V_{eff} = -u^I \mathcal{H}_{IJ} u^J = -\langle u|\mathcal{H}|u\rangle \,, \tag{3.18}$$

with

$$\mathcal{H} = \sqrt{g^{(7)}}\left(R^{(7)} - \ell_{11}^9 \rho_C - \frac{1}{2}F_7^2 - 3\nabla^2\right) = 3\sqrt{g^{(7)}}\hat{H}/\ell^2 \,. \tag{3.19}$$

Suppose that we consider a system for which the ground state wavefunction $u_0$ satisfies

$$\mathcal{H}|u_0\rangle \approx 0 \,, \tag{3.20}$$

so that the ground state energy $\lambda_0$ is close to zero. Such a wavefunction $u_0$ gives a good approximation to a solution of the constraint equation (3.7) for $C\ell^2 \ll 1$, a regime of interest for compactifications.

Let us denote metric deformations by $\gamma$ (corresponding to $h, \delta B$ in the compactifications (1.1) we are studying in this work). In this language, the $\gamma$ equation of motion at fixed $u = u_0$ is

$$\langle u|(\partial_\gamma \mathcal{H})|u\rangle = 0 \,. \tag{3.21}$$

In standard quantum mechanical perturbation theory, this corresponds to a vanishing first order correction to the small ground state energy in (3.20). This in turn means that (3.20) is preserved under this first order perturbation, so we can write

$$\frac{\delta}{\delta\gamma}(\mathcal{H}|u\rangle) \simeq 0 \implies \mathcal{H}|\partial_\gamma u\rangle \simeq -(\partial_\gamma \mathcal{H})|u\rangle \,. \tag{3.22}$$

We would like to analyze the Hessian for small fluctuations about such a solution.

### 3.1.1 Universal positive contribution

The mass term for $\gamma$ taking into account the warp factor variation is (using the chain rule and eliminating terms that vanish on shell)

$$2\ell_{11}^9 \frac{\delta^2 V_{eff}}{\delta\gamma^2} = -\langle u|(\partial_\gamma^2\mathcal{H})|u\rangle - 2\langle\partial_\gamma u|\mathcal{H}|\partial_\gamma u\rangle - 4\langle\partial_\gamma u|(\partial_\gamma\mathcal{H})|u\rangle\,. \tag{3.23}$$

Generically the warp factor Hamiltonian has no exactly zero eigenvalues and is invertible, so from (3.22) we have

$$|\partial_\gamma u\rangle = -\mathcal{H}^{-1}(\partial_\gamma\mathcal{H})|u\rangle\,. \tag{3.24}$$

Plugging this into (3.23) gives

$$2\ell_{11}^9 \frac{\delta^2 V_{eff}}{\delta\gamma^2} = \left\langle u \left| \left(-\partial_\gamma^2\mathcal{H} + 2\partial_\gamma\mathcal{H}\,\mathcal{H}^{-1}\partial_\gamma\mathcal{H}\right) \right| u \right\rangle\,. \tag{3.25}$$

Denote the eigenstates as $\mathcal{H}|u_\alpha\rangle = \lambda_\alpha|u_\alpha\rangle$, so that the "propagator"

$$\mathcal{H}^{-1} = \sum_\alpha \frac{1}{\lambda_\alpha}|u_\alpha\rangle\langle u_\alpha|\,. \tag{3.26}$$

If we now approximate the warp factor by the ground state, $|u\rangle \approx |u_0\rangle$, with $\lambda_0\ell^2 \ll 1$, we see that only excited states contribute to the second term in (3.25), since $\langle u_0|(\partial_\gamma\mathcal{H})|u_0\rangle \approx 0$ from (3.21). Therefore

$$2\ell_{11}^9 \frac{\delta^2 V_{eff}}{\delta\gamma^2} \approx \left\langle u_0 \left| \left(-\partial_\gamma^2\mathcal{H} + 2\partial_\gamma\mathcal{H}\left(\sum_{i\neq 0}\frac{1}{\lambda_i}|u_i\rangle\langle u_i|\right)\partial_\gamma\mathcal{H}\right) \right| u_0 \right\rangle \tag{3.27}$$

$$= \left\langle u_0 \left|(-\partial_\gamma^2\mathcal{H})\right| u_0 \right\rangle + \sum_{i\neq 0}\frac{1}{\lambda_i}|\langle u_0|\partial_\gamma\mathcal{H}|u_i\rangle|^2\,.$$

The same result arises from applying standard quantum mechanical perturbation theory, with a perturbation $\Delta\hat{H} = \gamma\partial_\gamma\hat{H}$ leading to a perturbation $\Delta u$ orthogonal to the unperturbed wavefunction. For $\lambda_0$ small and negative, and for larger level spacing (of order $1/\ell^2$ in our application), all the $\lambda_i > 0$ and thus the warping correction to the mass squared is always positive.

A key example of this appears in section 4.1 of [17], showing explicitly how the full Hessian including warping effects avoids an instability that would otherwise arise from short modes of the conformal factor in any compactification. In our application, there are two contributions that are automathically positive to the Hessian: this warping contribution, and the positive $\frac{\delta^2}{\delta h^2}\int\sqrt{g}(-R^{(7)})$ [16] arising from the negative sectional curvatures of our fiducial manifold. We will develop this further below in §5.4, using trial wavefunctions in the sense of the analogue Schrödinger problem and also analyzing the perturbations around our explicit patchwise solutions from §5.3.

## 3.2 The slow roll functionals in our landscape

In our de Sitter construction, the metastable minima of $V_{eff}$ are needed, requiring the formula (3.4) [17] for the full effective potential. For the purpose of more general cosmological evolution, other aspects of the four dimensional effective theory are of interest. Before leaving the subject of the effective four dimensional theory, we note that one can use this framework to derive an expression for the slow roll parameters (1.5). We can think of these as functionals of the internal fields, $\varepsilon_V[\delta B, h, C_6]$ and $\eta_V[\delta B, h, C_6]$ in the parameterization (1.1) of our landscape.

The general formula for the slow-roll parameter $\varepsilon_V$ is

$$\varepsilon_V = \frac{1}{2} \sum_I \left( \frac{\partial_{\phi_{c,I}} V_{eff}}{V_{eff}} \right)^2 \frac{1}{G_N}, \tag{3.28}$$

in terms of the canonically normalized four-dimensional fields $\phi_{c,I}$. Before continuing, we note that the model-independent parameter of most general interest is $\varepsilon = -\dot{H}/H^2$ (in terms of Hubble $H(t) = \dot{a}/a$ for FRW metric $-dt^2 + a(t)^2 d\vec{x}^2$ with scale factor $a(t)$). The model-independent quantity $\varepsilon$ directly determines whether accelerated expansion occurs: it does so if $\varepsilon < 1$ [38]. In general, this can arise due to a combination of kinetic and potential effects, not requiring a flat potential. However $\varepsilon_V \ll 1$ is a sufficient condition for accelerated expansion, and for simplicity we will focus on this quantity here.

In appendix §C we will incorporate the structure of the kinetic terms to derive explicit formulas for contributions to $\varepsilon_V$ from $\delta B$ and certain deformations $h$ (1.1); similar methods capture the full $\varepsilon_V$ and $\eta_V$. The resulting expressions are amenable to both analytic estimates and numerical evaluation.

## 4  Casimir contribution

Since it is essential for the volume stabilization mechanism, we next explain the origin of the strong Casimir contribution in our compactifications. As explained above in §2, the Casimir stress energy will dominate in regions of the internal space with a small, slowly varying circle of size $R_c$. The study of short geodesics is known as systolic geometry, with various explicit constructions given in hyperbolic geometry [20–22] and in more general Einstein spaces obtained by filling in hyperbolic cusps [19]. These constructions, which feature a small circle supported in a local Einstein geometry are suitable for our application. In this section, we will describe this in detail, focusing for specificity on the case of a hyperbolic cusp filled in as in [19]. We will incorporate the warping effects encoded in the effective potential $V_{eff}$ [17] reviewed in the previous section.

A hyperbolic manifold can contain thin regions, including near-cusps whose metric is

$$ds^2_{cusp} = dy^2 + e^{-2y/\ell} ds^2_{T^6_{comoving}}, \quad y_0 < y < y_c. \tag{4.1}$$

The upper endpoint $y_c$ prescribed here can effectively arise via Anderson's generalization of Dehn filling [19], with the metric (4.1) joined to a twisted Euclidean AdS black hole geometry as described in [19], smoothed out to form an Einstein space. We review this construction in more detail in appendix A.2. In that construction, or other realizations of thin regions containing a small, finite-size circle, the local geometry is a small circle $S^1$ times a region of $\mathbb{R}^6$. We can view $y_c$ in (4.1) as a regulator modeling the finite size of the minimal circle in these constructions.

This filled cusp is, for now, a fiducial geometry in our compactification; we will discuss the back reacted warped geometry – including the dynamics of the $S^1$ size, and its stability –in detail in what follows. In sections 5.1.1 and 5.1.2 we illustrate some of the dynamics of our fields in simpler analogues of the geometry [19], in section 5.2 we derive the sourced fields away from the Casimir region, and in section 5.3 we present a back reacted cusp solution smoothly joining these regions.

At smaller $y$, the cusp joins to the central manifold. In (4.1) we have described this with a lower endpoint $y_0$, but we note that there is a more natural interface that is totally geodesic: a hyperbolic space can be obtained by gluing a set of hyperbolic polygons via a pairing of their totally geodesic facets, including those with ideal vertices that join together to form cusps [15];

see e.g. [25] for recent examples. For our present purposes of calculating the Casimir energy, the large $y$ region of the cusp dominates: the small proper size of the $T^6$ for large $y$ leads to a large contribution to the Casimir energy $\sim -1/R_c^{11}$ localized in that region.

We will take antiperiodic fermion boundary conditions on the minimal-length cycle(s) of the $T^6$, leaving bosons as the dominant contribution to the Casimir energy. The proper size $R_{T^6}$ of the cycles in the $T^6$ are small at large $y$, and in that regime they shrink slowly: $\frac{dR_{T^6}}{dy} \propto e^{-y/\ell}$. This leads to a simple result for the expectation value of the Casimir stress-energy for each bosonic fluctuation, as reviewed in e.g. [39]:

$$\langle T^{(\text{Cas})\mu}{}_{\nu}\rangle = -\rho_C(R_c)\,\delta^{\mu}_{\nu}, \quad \langle T^{(\text{Cas})y}{}_{y}\rangle = -\rho_C(R_c), \quad \langle T^{(\text{Cas})a}{}_{b}\rangle = -\left(\rho_C(R_c) + \frac{1}{N}R_c\rho'_C(R_c)\right)\delta^a_b, \tag{4.2}$$

with

$$\rho_C(R_c) \sim -\frac{1}{R_c^{11}} \sim -\frac{e^{11y/\ell}}{\lambda_c^{11}}, \tag{4.3}$$

where $\lambda_c$ is the length of the shortest cycle on the $T^6$ cross section in the comoving metric $ds^2_{T^6}$ in (4.1). Here $N$ is the number of dimensions with the minimal cycle size $R_c$, and $\mu, \nu$ range over both the four external dimensions of the putative $dS_4$ as well as the remaining $6-N$ torus directions. This expression also assumes that the warp factor in (1.1) does not vary rapidly enough to affect the modes entering into this calculation of the Casimir stress energy. We will see below that this is the case in our backgrounds.

Let us denote the volume of the comoving torus as $v_T \ell^6$. We can estimate the integral that enters into the second term of (2.3) as:

$$
\begin{aligned}
\frac{\int \sqrt{g_{\mathbb{H}}}\rho_C\,\ell_{11}^4}{v_7\hat{\ell}^7} &\sim -n_c\frac{v_T\hat{\ell}^6}{v_7\hat{\ell}^7}\int_{\hat{y}_0}^{\hat{y}_c}\frac{d\hat{y}}{\hat{\lambda}_c^{11}}e^{-6\hat{y}/\hat{\ell}}e^{11\hat{y}/\hat{\ell}} \simeq -\frac{v_T n_c}{5v_7\hat{\lambda}_c^{11}}e^{5y_c/\ell} \\
&\sim -\frac{1}{\hat{\ell}^{11}}\left(\frac{\hat{\ell}}{\hat{R}_c}\right)^5\frac{\text{Vol}(T^6_{comoving})}{\lambda_c^6}\frac{n_c}{v_7}
\end{aligned}
\tag{4.4}
$$

times 128, the number of bosonic species, where again hats indicate variables in units of the 11d Planck length $\ell_{11}$, and $n_c$ is the number of cusps like this. This incorporates our minimal cycle size $R_c \ll \ell$, cutting off the divergence in the Casimir energy. This expression reproduces the general behavior discussed above in (2.6). Without including additional effects, balancing this against the curvature term would yield a stabilization mechanism

$$\hat{\ell}^4 \sim \frac{1}{\hat{R}_c^5}\left[\frac{128}{42}\frac{n_c}{v_7}\frac{\text{Vol}(T^6_{comoving})}{\lambda_c^6}\right] \quad \text{(no warping or backreaction)}. \tag{4.5}$$

From this we see that obtaining $\ell \gg R_c \gg \ell_{11}$ would require the factor in square brackets to be $\gg 1$. The final factor $\frac{\text{Vol}(T^6_{comoving})}{\lambda_c^6}$, in itself, grows with increasing cusp asymmetry, e.g. a square torus with one circle radius $R_c$ much less than the others. It follows from Mostow rigidity that cusp shapes of hyperbolic manifolds do not vary continuously, but there are many discrete choices of finite-volume spaces $\mathbb{H}_7/\Gamma$ depending on the choice of isometry subgroup $\Gamma$. Mathematical work has established the existence of a dense set of cusp cross sections in various cases including [40] and [41]. It is straightforward to obtain a parametrically large asymmetry in an elementary way by taking covers of explicit manifolds with symmetric cusps such as the 7-manifold in [25] to construct hyperbolic manifolds with arbitrarily asymmetric cusps.[11] In that procedure, however, we find that the volume per cusp grows with the asymmetry in such a way that the factor in square brackets remains of order 1.

---

[11]We thank Ian Agol for this suggestion.

This behavior may be more general. In the mathematical literature, there is much discussion of the spectrum of volumes of hyperbolic manifolds. Of particular interest to us is an inverse relationship between systole size and volume: such a bound derived in [21, 22] in the context of the inbreeding construction for small systoles [20] has the same effect of maintaining an order 1 value of the factor in brackets in (4.5). This suggests an interesting relationship between bounds on negative energy in physics and systolic geometry in mathematics.

However, in our physical context, this is not the final result: additional effects on $R_c$ and the Casimir energy are important. These involve the full four dimensional effective potential $V_{eff}$ introduced in the previous section. There are two basic effects that will be important to consider: a contribution of warp and conformal factor gradients reducing the curvature term in the potential, and a tadpole for $R_c$ that will be stabilized by a combination of the rigidity and warp factor screening. In particular, the gradient of the warp factor in the effective potential (3.4) can reduce the magnitude of the curvature term allowing the Casimir compete with it in the regime $\ell \gg R_c \gg \ell_{11}$. Indeed, define

$$a = \frac{\int \sqrt{g^{(7)}} u^2|_c [-R^{(7)} - 3\left(\frac{\nabla u}{u}\right)^2 \big|_c]}{\int \sqrt{g^{(7)}} u^2|_c \, 42/\ell^2} , \tag{4.6}$$

with $R^{(7)}$ the full internal curvature. Balancing the warping-corrected full curvature term and the Casimir contribution now requires

$$a \int \sqrt{g^{(7)}} u^2|_c \frac{42}{\ell^2} \sim \ell_{11}^9 \int \sqrt{g_7} u^2|_c |\rho_c| , \tag{4.7}$$

equivalently,

$$\hat{\ell}^4 \sim \frac{1}{a} \frac{1}{42} \hat{\ell}^6 \ell_{11}^{11} \frac{\int \sqrt{g^{(7)}} u^2|_c |\rho_c|}{\int \sqrt{g^{(7)}} u^2|_c} . \tag{4.8}$$

The unwarped analysis above is obtained in the limit $u \to 1$ ($a \to 1$), with the right hand side of (4.8) proportional to the averaged Casimir contribution $\sim \frac{\int \sqrt{g^{(7)}} |\rho_c|}{Vol_7}$ estimated in (4.4) and leading to the unwarped relation (4.5). However, we see that $a \ll 1$ helps the two terms to compete, stabilizing at $\hat{\ell} \gg 1$, even if the last factor is of order 1.

We will explain more about this tuning shortly, but first let us incorporate the second effect. With the full effective potential in place, we can also give a glimpse of the dynamics of the Casimir circle $R_c$ and its role in the stabilization mechanism. The first point to note is that the Casimir energy becomes more negative as $R_c$ shrinks. As a result, starting from the fiducial metric, given any asymmetry in the hyperbolic cusp, there is a nonzero force $-R_c \partial_{R_c} V_{eff} < 0$. To determine the fate of this tadpole, we must again incorporate the warping effects in the full $V_{eff}$. This contains dependences on the internal fields that enter via the solution $u|_c$ to the constraint equation (3.7) appearing in (3.9). To study the net effect of the tadpole $R_c \partial_{R_c} V_{eff}$ and the warping, we will use the fact that the constraint equation takes a Schrödinger form for small $C\ell^2$, with $u$ proportional to the analogue wavefunction with analogue energy eigenvalue near zero. In §5.1, we will study this in detail. In particular, we will explain how the parameters required to ensure $C\ell^2 \ll 1$ arise in our setup, analyzing the relevant integrated quantities that enter into the volume stabilization, and also enable a perturbative treatment of other metric deformations. In the remainder of this section, we summarize more schematically this dynamics and how it applies to our stabilization mechanism.

In the analogue Schrödinger problem, the increasing magnitude of the localized Casimir energy as $R_c$ shrinks leads to a stronger potential barrier $\sim |\rho_C(y)|$ for the 'wavefunction' $u$. For a sufficiently small $R_c$, the barrier is high enough that in the region of support of $\rho_C$, the wavefunction is in the classically disallowed regime, below this barrier. Once this happens,

the analogue wavefunction becomes exponentially suppressed $\sim \text{Exp}[-\sqrt{|\rho_C|}]$. In (3.9), the integrand contains a factor of $u^2$. As a result, the Casimir term in the potential for the unstable mode $\hat{R}_c$ behaves schematically as $V_{eff}[\hat{R}_c] \sim -b\hat{R}_c^{-11}\exp[-b\hat{R}_c^{-11/2}]$ for sufficiently small $R_c$. This, combined with the Hessian, is enough to prevent an unbounded instability for small $R_c$ as we will discuss further below in §5.1, and we will exhibit a backreacted cusp solution with stabilized $R_c$ in §5.3.

Let us now return to the question of what provides the large parameter needed in the $\ell$ stabilization to ensure that all length scales can be $\gg \ell_{11}$, now working with the full effective potential (3.9). Recall that above in (4.5), we were left with a factor $\left[\frac{n_c}{v_7} \frac{\text{Vol}(T^6_{comoving})}{\lambda_c^6}\right]$ of order 1, but we had not yet taken into account any tuning of $a$ as defined in (4.6), or the tadpole along which $R_c$ shrinks, along with the warping effects that ultimately stabilize this tadpole at a lower value of $V_{eff}$. We will now incorporate those effects and check the relative sizes of $\ell, R_c,$ and $\ell_{11}$. Since all terms in (3.9) will compete in the solution, we get a relation between our parameters by balancing the first term $-R^{(7)} - 3((\nabla u)/u)^2$ against the Casimir term, giving

$$\hat{\ell}^4 \sim \frac{1}{\hat{R}_{c*}^5} \left[\frac{128}{42} \frac{b}{a} u_*^2 \frac{n_c}{v_7}|_* \frac{\Delta Y}{\ell} \frac{\text{Vol}(T^6)}{R_c^6}|_*\right], \tag{4.9}$$

where we denote with a $*$ the values of geometric quantities at the stabilized value of $R_c$, including the proper volume of $T^6$ in the Casimir region of size $\Delta Y$ and support $u^2$; the reduced $R_c$ has an effect also on the total volume $v_7$, reducing it somewhat as $R_c$ shrinks. We also defined

$$b = \frac{\int d^7y \sqrt{g^{(7)}} u^2(y)|\rho_C(y)|\Big|_c}{128\, u_*^2 \text{Vol}_C/R_{c,*}^{11}}. \tag{4.10}$$

This constant $b$ will be order 1 in the numerical solutions below and we will often drop it in what follows. This reduces to the previous expression in the unwarped Einstein metric (4.5) if we set $u_* = 1 = a = b = \Delta Y/\ell$ and identify $R_{c*}$ with its value in the Einstein metric. With the warping effects included, this factor

$$\frac{1}{\epsilon} = \left[\frac{128}{42} \frac{b}{a} u_*^2 \frac{n_c}{v_7}|_* \frac{\Delta Y}{\ell} \frac{\text{Vol}(T^6)}{R_c^6}|_*\right] \tag{4.11}$$

may be $\gg 1$. Starting from a configuration with $\epsilon \sim 1$, as we descend the potential landscape in the direction of the $R_c \partial_{R_c} V_{eff}$ tadpole, the number $n_c$ of cusps stays constant and the remaining factors lead to some increase in $1/\epsilon$, although the effects of other fields including the warp factor stabilize it, as we will see below in §5.1 and §5.3.

The method that we will focus on to ensure that $\epsilon \ll 1$ is to tune $a \ll 1$. To achieve this, we need to incorporate variations in the warp and conformal factors. One general result in compactifications with $C \geq 0$ is that in a region where the Casimir contribution is negligible, the solution of the warp and conformal factor equations of motion implies a net negative value pointwise of the potential energy density appearing in the formula for $V_{eff}$ [42],

$$-R^{(7)} - 3\left(\frac{\nabla u}{u}\right)^2 = 4\ell_{11}^9|\rho_C| - \frac{C}{u} - \frac{5}{2}F_7^2. \tag{4.12}$$

This combination $-R^{(7)} - 3\left(\frac{\nabla u}{u}\right)^2$ contributing to $a$ may be positive with sufficiently strong Casimir energy, near the small circles in our construction. Conversely, where the Casimir energy is negligible, we will have a negative contribution to $a$. In general, the contributions to $a$ coming from the variation of various metric components depend on different aspects of the

compactification (related to the group $\Gamma$ and cusp fillings) and flux quanta, suggesting they are tunable via small variations of contributions to $a$:

$$|\Delta \int \sqrt{g^{(7)}} u^2 \left( -R^{(7)} - 3(\frac{\nabla u}{u})^2 \right)| \ll |\int \sqrt{g^{(7)}} u^2 \left( -R^{(7)} - 3(\frac{\nabla u}{u})^2 \right)|. \qquad (4.13)$$

We can analyze this more concretely as follows.

Below in section 5.3 we will present a class of back reacted solutions in the cusp which smoothly joins the Casimir and bulk regions (with positive and negative contributions to $a$, respectively). These solutions evolve purely radially (depending on $y$ in (4.1)), whereas the full solution must depend on all directions near the gluing to the central manifold, but the solution illustrates several important features. In these solutions by themselves, one can (as we show explicitly below) tune $0 < a \ll 1$. In the underlying hyperbolic space, the cusps contain a significant fraction of the volume of the manifold, so the existence of cusp solutions with small $a > 0$ suggests that this tuning is possible in the full space.

## 4.1 Tuning $a \ll 1$

Indeed, we can identify parameters enabling us to tune $a$. Suppose we start from a solution in which $a$ is too large. Starting from manifolds such as [25], summarized in appendix A, we may add bulk regions which contribute negatively to $a$. Specifically, using the filling prescription [19] we can obtain (via choice of simple closed geodesic as reviewed in §A.2) short cusps which add volume without significant Casimir energy; we can also take covers of the manifold to proliferate or extend either type of cusp, as described in the appendix. Conversely, starting from a value of $a$ that is too small, we can increase it by reducing the flux quantum number $N_7$. Since this appears on the right hand side of (4.12), reducing it will reduce the negative contribution to $a$ in the bulk regions. In the explicit solutions in section 5.3, the strong Casimir region does not get much contribution from flux, so this adjustment of $N_7$ predominantly affects the bulk region, increasing $a$. Below in §5.5 we will see that the adjustment of $a$ indeed leads to large length scales in all directions, including the Casimir circle. This mechanism, together with the wide availability of the relevant sequences of geometric and flux parameters, also suggests that this effect is parametric, that it is possible to parametrically control all the length scales.

## 4.2 Bounding additional quantum effects

Having described the crucial effect of the Casimir energy in our stabilization mechanism, let us now assess other quantum effects in the theory. In addition to $V_{eff}$, the kinetic terms get quantum corrections. The renormalization of the four-dimensional Newton constant is the most UV sensitive of these, but as we will see here this is suppressed compared to its classical value. We can study this both from the bottom up 4d perspective and the 11d perspective.

In the 11d language we obtain a correction

$$\sim \frac{n_c \text{Vol}_C}{v_7 \ell^7} \frac{R^{(11)}}{R_{c*}^9} \qquad (4.14)$$

to the equations of motion, with $R^{(11)}$ the 11d curvature scalar. This can be thought of as a curvature expansion of the Casimir stress energy calculation (which starts at order $\frac{1}{R_{c*}^{11}} \frac{n_c \text{Vol}_C}{\text{Vol}_7}$ as above). The $R^{(4)}$ part of this contains a renormalization to the Einstein term. Since we have

$$n_c \frac{\text{Vol}_C}{\text{Vol}_7} \sim \frac{n_c}{v_7} \frac{R_{c*}^6}{\ell^6} \frac{\text{Vol}(T^6)}{R_c^6}|_*, \qquad (4.15)$$

we obtain a ratio of quantum contributions to $1/G_N$ to classical ones of order

$$\frac{\Delta 1/G_N}{1/G_N} \sim \frac{\ell_{11}^9}{\ell^6 R_{c*}^3} \frac{n_c}{v_7} \frac{\mathrm{Vol}(T^6)}{R_c^6}\big|_* . \tag{4.16}$$

This ratio is small since we work in a regime where $\ell \gg R_{c*} \gg \ell_{11}$.

In the four dimensional effective theory, the Newton constant is given up to one loop order by

$$\frac{1}{G_N} = \frac{1}{\ell_{11}^9} \int \sqrt{g^{(7)}} u + \text{quantum} \sim \frac{v_7 \ell^7}{\ell_{11}^9} + \frac{N_{species}}{L_{SUSY-breaking}^2} , \tag{4.17}$$

where we used the fact that in the bulk of our space, where the classical contribution to $1/G_N$ gets its dominant contribution, the variation of $u$ is not large in the sense we have described above (as detailed in §5.2 below). The cutoff on the effective theory is of order the SUSY breaking scale $\frac{1}{L_{SUSY-breaking}}$, with an effective number of species $N_{species}$ contributing.

The SUSY breaking scale is of order $1/R_{c*}$ for species localized in the region of the small circles dominating the quantum corrections,

$$\frac{N_{species}}{L_{SUSY-breaking}^2} \sim \frac{1}{R_{c*}^2} n_c \left\{ \frac{\ell}{R_{c*}} \frac{\mathrm{Vol}(T^6)}{R_c^6}\big|_* \right\} . \tag{4.18}$$

As indicated here, the number of species contributing up to the scale $1/R_{c*}$ is the number of cusps $n_c$ times the factor in curly brackets which gives the number of KK modes of mass $1/R_{c*}$ in the region with the small circle of size $R_{c*}$. This region is parametrically of proper length $\sim \ell$ in the radial direction down each (filled) cusp. Given this, the 4d description reproduces the ratio in (4.16).

The result is that although Casimir energy competes with the classical terms in $V_{eff}$ via our mechanism (4.9), the quantum correction to the Newton constant is subdominant to its classical value. Related to this, there is no suppression of the bare Newton constant term, analogous to the available tuning $a \ll 1$ in (4.6) that applies to the leading (curvature) term in $V_{eff}$. Similarly, the quantum corrections to the kinetic terms of 4d scalar fields are suppressed compared to the classical contribution.

# 5 Effect of inhomogeneities near the hyperbolic metric

We have seen that the averaged Casimir energy, combined with flux and curvature, leads to a stabilized volume provided that we can tune $a \ll 1$ (4.6). Having laid out the appropriate formalism and our strategy for large radius control, our next step is to determine the effect of the inhomogeneity of the Casimir contribution to the effective theory. This involves four interrelated parts:

($i$) A more detailed analysis of the geometry in the region toward the end of the cusp with a strong Casimir energy, to check the robustness of the localized Casimir source described in §4.

($ii$) More generally check of the effect of warping and a varying conformal factor on $V_{eff}$ including bulk regions away from the end of the cusp, given the localized Casimir source.

These studies (i) and (ii) enter into the tunability of our parameter $a$ (4.6), with contributions to it from different regions.

($iii$) An analysis of the Hessian around the a background consisting of our hyperbolic metric dressed with warp and conformal factor variations. We have two universal positive contributions: the rigidity of non-conformal deformations of the hyperbolic metric, and the

positive contribution from warping calculated model-independently in §3.1. Extending those analyses leads to evidence for a positive Hessian overall as described below in §5.4.

($iv$) Applying (iii) to obtain a check of the magnitude of the deformation in the bulk fields needed to absorb residual tadpoles arising from the inhomogeneity. In order to make a general estimate of the effects of the inhomogeneity-induced tadpoles on $\delta B$ and $h$ in (1.1), we need to compare the linear and quadratic terms upon expanding the potential in these fluctuations. The shift in a given field $\sigma$ is given by

$$\Delta \sigma = \frac{\partial_\sigma V_{eff}}{m_\sigma^2},\tag{5.1}$$

if the Hessian $\mathcal{H}_{IJ} = \partial_I \partial_J V_{eff}$ is diagonalized, or more generally

$$\partial_I \partial_J V_{eff} \Delta \sigma_J = \partial_I V_{eff} \Rightarrow \Delta \sigma_J = \mathcal{H}^{-1} \partial V_{eff}.\tag{5.2}$$

We present this estimate below in §5.5, finding a small shift from the dressed bulk hyperbolic metric.

## 5.1 Geometry and Casimir source near small circle

In this section we will elaborate on the dynamics summarized in §4, filling in some of the requisite details. Here we focus on warp factor effects, in sections 5.2 and 5.3 we fill in details of the full dressed background solutions, and in section 5.4 we will describe the Hessian around this background.

Let us start by putting our analysis in a broader context. The metastability of M/string theory compactifications depends on the strength of the intermediate negative source in the expansion about weak coupling and large radius [18, 43–45]. Such sources must compete with the other terms to create a dip in the potential as in Figure 3, without engendering any runaway instability in the theory as a whole. For example, negative mass Schwarzschild solutions should be absent to avoid rampant pair production instabilities [46]. This requirement relates to fascinating aspects of quantum field theory and quantum gravity involving energy conditions [47–51], a connection that would be interesting to explore more systematically in the future. In the context of the compactifications we are studying here, this also connects to mathematics in an interesting way: small circles supporting strong Casimir energy come with lower bounds on the overall volume of the internal manifold [22, 52].

In perturbative string limits, orientifold planes play this role beautifully, appearing at the first subleading order in the expansion in the string coupling. Their long range effect on spacetime and their non-dynamical positions prevent pair production. Although limited in number, they proliferate in setups of interest, e.g. at large dimensionality [45, 53] and in the presence of triply intersecting 7-branes [13, 54, 55]. They can play a useful role in systems with stronger gradients such as [56].

In the present construction, Casimir energy plays this role. To understand its behavior including the effects of gravitational interactions, it is useful to employ the effective potential formalism in §3. In the constraint equation (3.7), viewed as an analogue Schrödinger equation, a region of negative energy introduces a potential barrier. The warp factor $u$, proportional to the analogue wavefunction, decreases exponentially in the corresponding classically disallowed region. This screens such regions [17], preventing a runaway instability.

This raises the question of whether such screening limits the negative stress energy sources to the extent of eliminating the dip in $V_{eff}$ essential to metastability of de Sitter. We can eliminate this possibility in general, as follows (and we will see this in action in examples). In order to assess the strength of our negative source in $V_{eff}$, we must solve the constraint (3.7) and include the resulting $u$-dependences in (3.4). In this section, we will show that a significant

negative contribution survives this step, preserving a three-term stabilization structure for the volume. This follows both from general estimates of Schrödinger wavefunctions – applicable to small systoles wherever they appear – as well as from explicit warped solutions in the cut off cusp and the filled cusp including a backreacted solution for the warp and conformal factors there.

We can derive this in general as follows. Consider a solution of the constraint equation (3.7) with small $C$. Now integrate this equation against $u$, obtaining

$$\ell_{11}^9 \int \sqrt{g^{(7)}} u^2 |_c \rho_C = -\int \sqrt{g^{(7)}} u^2 |_c \left( \left[ -R^{(7)} - 3 \left( \frac{\nabla u}{u} \right)^2 \Big|_c \right] + \frac{1}{2} |F_7|^2 - \frac{C}{2u|_c} \right). \quad (5.3)$$

Here we dropped boundary terms, working on a closed manifold, e.g. with the Casimir energy localized near a small systole. As we will see in detail in the next section, $\int \sqrt{g^{(7)}} (\nabla u)^2 |_c$ gets its support in the region where the Casimir energy dominates; in the bulk of the manifold, $\nabla^2 A \gg (\nabla A)^2$ and this term is subdominant in $V_{eff}$, satisfying the criterion articulated in [8]. In this region, positive curvature develops – as we saw above in (4.12), when Casimir energy is negligible the equations of motion imply $-R^{(7)} - 3 \left( \frac{\nabla u}{u} \right)^2 < 0$ pointwise [42]. We will exhibit a tunably positive contribution to $-R^{(7)} - 3 \left( \frac{\nabla u}{u} \right)^2$ from the end region. For sufficiently small contribution from the bulk volume, the net effect is

$$\int \sqrt{g^{(7)}} u^2 |_c \left[ -R^{(7)} - 3 \left( \frac{\nabla u}{u} \right)^2 \right] > 0. \quad (5.4)$$

As a result, if we solve the constraint (3.7) with $C \simeq 0$[12], and satisfy (5.4), it is inevitable that $\rho_C$ will compete with the other terms, cancelling them as in the original 3-term structure. We may work with a fiducial flux contribution, $\bar{F}^{(7)\prime}$ in (3.13) to begin with, so that the constraint equation is linear. As we will see shortly, the true flux solution $\bar{F}^{(7)}$ is a small perturbation from this. Viewing the constraint as a Schrödinger problem, $C \simeq 0$ corresponds to demanding an energy eigenvalue $\simeq 0$, a one parameter tune. Discrete parameters in the group $\Gamma$ in $\mathbb{H}_7/\Gamma$ and in the flux quantum number $N_7$ can be tuned to achieve this and (5.4).

In addition to the competitive size of $\int \sqrt{g^{(7)}} u^2 \rho_C$ just addressed, we need to check that the $\ell$-dependence of $u|_c$ does not overcome the 3-term stabilization mechanism that is based on the variation with respect to $\ell$ of curvature, Casimir energy, and flux. In other words, since $u|_c$ depends on $g_{ij}^{(7)}$ (in particular the overall scale $\ell$), $V_{eff}$ contains additional dependence on $\ell$ beyond these three sources. We will now show that this new $\ell$ dependence can be subdominant to that of the original sources in our parameter regime of interest.

We will use perturbation theory, writing $\ell = \ell_0 + \delta\ell$, expanding all quantities in $\delta\ell$ and determining which contributions are dominant in the variation of $V_{eff}$. We start from a fiducial configuration given by the hyperbolic metric and flux

$$F_{fid,j_1,\ldots,j_7}^{(7)} = \ell_{11}^6 \frac{\tilde{N}_7}{\text{Vol}_7} \sqrt{g^{(7)}} \epsilon_{j_1,\ldots,j_7}. \quad (5.5)$$

Note that in (5.5) we are not yet working with the $u$-dependent solution for $F_7$ described above in (3.10), and we distinguish the flux quantum number $N_7$ from $\tilde{N}_7$ which will be an auxiliary parameter in our analysis. This fiducial configuration is a conceptually useful starting point

---

[12]More precisely, we will be interested in a regime $C\ell^2 \ll a$ as described below in (5.21). Note that the overall scale of the warp factor $u$ is fixed in terms of the Newton constant by (3.6), and given the resulting relation (3.8) we identify $C$ as the proper curvature in the four dimensional theory, of order Hubble$^2$ in our dS and inflationary models. In other words, we will mostly work in a regime in which 4d Hubble is weaker than the internal curvature scale.

because it preserves the linearity of the warp factor constraint (3.7). We will then incorporate effects from $C_6$ and metric tadpoles.

Because of the form of the constraint equation, part of our problem is similar to ordinary quantum mechanical perturbation theory. We start from this equation, rescaled with a factor of $\ell^2$ to take the form

$$\left(-\nabla_w^2 - \frac{1}{3}\left(-\ell^2 R^{(7)} + \ell_{11}^9 \ell^2 \rho_C + \frac{\ell^2}{2}|F_{fid}^{(7)}|^2\right)\right)u = -\frac{C\ell^2}{6}, \tag{5.6}$$

where $w_i = y_i/\ell$ is a dimensionless coordinate. The effective Schrodinger Hamiltonian $\hat{H}$ is perturbed as

$$\hat{H} = -\nabla_w^2 - \frac{1}{3}\left(-\ell^2 R^{(7)} + \ell_{11}^9 \ell^2 \rho_C + \frac{\ell^2}{2}|F_{fid}^{(7)}|^2\right) = H^{(0)} + \frac{\delta\ell}{\ell}H^{(1)}, \tag{5.7}$$

with

$$\hat{H}^{(0)} = -\nabla_w^2 - \frac{1}{3}\left(-\ell_0^2 R^{(7)} + \ell_{11}^9 \ell_0^2 \rho_C^{(0)} + \frac{\ell_0^2}{2}(F_{fid}^{(0)})^2\right), \tag{5.8}$$

and

$$\hat{H}^{(1)} = \ell_0^2\left(3\ell_{11}^9 \rho_C^{(0)} + 2(F_{fid}^{(0)})^2\right). \tag{5.9}$$

This last expression follows from the power law scalings with $\ell$ of the Casimir and flux terms (multiplied by $\ell^2$ in (5.6)); the curvature scales like $1/\ell^2$ and does not contribute to $\hat{H}^{(1)}$.

We start from a configuration as just laid out in the discussion surrounding (5.3), with warp factor $u^{(0)}(w)$ satisfying the constraint with the $C$ term subdominant to the contributions from individual sources. Perturbing around that, we have

$$\begin{aligned}
\ell &= \ell_0 + \delta\ell, \\
u &= u^{(0)} + \frac{\delta\ell}{\ell}u^{(1)}, \\
u_k &= u_k^{(0)} + \frac{\delta\ell}{\ell}u_k^{(1)}, \\
\lambda_k &= \lambda_k^{(0)} + \frac{\delta\ell}{\ell}\lambda_k^{(1)},
\end{aligned} \tag{5.10}$$

where the last two lines are the eigenfunctions and eigenvalues of $\hat{H}$:

$$\begin{aligned}
\hat{H}u_k &= \lambda_k u_k, \\
\hat{H}^{(0)}u_k^{(0)} &= \lambda_k^{(0)}u_k^{(0)},
\end{aligned} \tag{5.11}$$

with orthonormalization condition similar to that in [17],

$$\frac{1}{\text{Vol}_7}\int \sqrt{g}\, u_j u_k = \delta_{jk}. \tag{5.12}$$

We find the familiar expression for the first-order shift in eigenvalues: at order $\delta\ell$, the eigenvalue equation is

$$\hat{H}^{(1)}u_k^{(0)} + \hat{H}^{(0)}u_k^{(1)} = \lambda_k^{(1)}u_k^{(0)} + \lambda_k^{(0)}u_k^{(1)}. \tag{5.13}$$

Integrating this against $\int \sqrt{g^{(0)}}u_j^{(0)}/\text{Vol}_7^{(0)}$ and integrating by parts gives

$$\begin{aligned}
\lambda_j^{(1)} &= \frac{1}{\text{Vol}_7^{(0)}}\int \sqrt{g^{(0)}}u_j^{(0)}\hat{H}^{(1)}u_j^{(0)} \\
&= \frac{1}{\text{Vol}_7^{(0)}}\int \sqrt{g^{(0)}}u_j^{(0)}\ell_0^2\left(3\ell_{11}^9 \rho_C^{(0)} + 2(F_{fid}^{(0)})^2\right)u_j^{(0)},
\end{aligned} \tag{5.14}$$

where in the second line we used (5.9).

From [17] equation (2.38), we then have

$$u(w) = -\frac{1}{6}C\ell^2 \sum_k u_k(w) \frac{1}{\lambda_k} \frac{1}{\text{Vol}_7} \int d^7y \sqrt{g} u_k \,, \tag{5.15}$$

and substituting this into $\frac{1}{G_N} = \frac{1}{\ell_{11}^9} \int \sqrt{g} u$ and using $V_{eff} = \frac{C}{4G_N}$, we have (cf. (2.43) of [17])

$$V_{eff} = -\frac{1}{G_N^2} \frac{3 \, \ell_{11}^9 \text{Vol}_7}{2\ell^2} \frac{1}{\sum_k \frac{1}{\lambda_k} |\int \sqrt{g} u_k|^2} \,. \tag{5.16}$$

We can also express the effective potential upon integrating out the constraint as

$$V_{eff} = \frac{C}{4G_N} = \frac{C}{4\ell_{11}^9} \int \sqrt{g} u \,. \tag{5.17}$$

We are starting from a configuration described above around (5.3) with $C$, and hence $V_{eff}$ near zero; $0 \lesssim C \ll -R^{(7)}$. In (5.16) we see that this corresponds to having a small negative eigenvalue $\lambda_0 \lesssim 0$. We would like to impose that this corresponds to good approximation to a local *minimum* of $V_{eff}$ in the $\ell$ direction, to confirm that the 3-term structure remains consistent with the warping. As such, we need the variation with $\ell$ of $V_{eff}$ to be much smaller than that of its individual terms, such as the curvature term $\propto 42/\ell^9$ in (2.3). More precisely, we will require this variation to be much less than the $[-R^{(7)} - 3\left(\frac{\nabla u}{u}\right)^2 \big|_c]$ term (3.9) for which

$$\delta V_a \sim \frac{\delta\ell}{\ell} \frac{9\ell_{11}^9}{G_N^2\ell^9} \frac{\text{Vol}_7 \int d^7y \sqrt{g^{(0)}}\,(u^{(0)}|_c)^2 a}{(\int d^7y \sqrt{g^{(0)}} u^{(0)}|_c)^2} \,, \tag{5.18}$$

where $a$ was defined above in (4.6). Since we have a small eigenvalue, expanding $V_{eff}$ in the form (5.16) gives (using $\text{Vol}_7 \sim \ell^7$)

$$\delta V_{eff} \simeq -\frac{\delta\ell}{\ell} \frac{3\ell_{11}^9}{2G_N^2\ell^9} \frac{\lambda_0^{(1)}}{(\int d^7y \, \ell^{-7} \sqrt{g^{(0)}} u^{(0)}|_c)^2} + \mathcal{O}(\lambda_0^{(0)}) \,. \tag{5.19}$$

Comparing (5.19) and (5.18) we see that our criterion for a small variation with respect to $\ell$ in our configuration near $V_{eff} = 0$ is satisfied provided that we tune

$$\lambda_0^{(1)} \ll a \,. \tag{5.20}$$

This is given by (5.14), and the signs work for this tune to be available. Altogether, the conditions we require on our discrete parameters – the group $\Gamma$ defining the hyperbolic manifold $\mathbb{H}_7/\Gamma$, the size of the Dehn/Anderson fillings [19] reviewed in appendix A.2, and the fiducial flux quantum number $\tilde{N}_7$ are

$$a \;\gg\; \frac{1}{\text{Vol}_7^{(0)}} \int \sqrt{g^{(0)}} (u_0^{(0)})^2 \ell_0^2 \left(3\ell_{11}^9 \rho_C^{(0)} + 2(F_{fid}^{(0)})^2\right),$$

$$a \int \sqrt{g^{(7)}} u_0^2 (42/\ell^2) \;\gg\; \int \sqrt{g^{(0)}} (u_0^{(0)})^2 \left(-R_7^{(0)} - 3\left(\frac{\nabla u^{(0)}}{u^{(0)}}\right)^2 + \ell_{11}^9 \rho_C^{(0)} + \frac{1}{2}(F_{fid}^{(0)})^2\right) > 0,$$

with

$$a \int \sqrt{g^{(7)}} u_0^2 (42/\ell^2) \;\simeq\; \int \sqrt{g^{(0)}} (u_0^{(0)})^2 \left(-R_7^{(0)} - 3\left(\frac{\nabla u^{(0)}}{u^{(0)}}\right)^2\right) \gtrsim 0, \tag{5.21}$$

with the last requirement just as in the above discussion of the fiducial configuration. Note that these are integrated conditions (rather than local conditions), tunable with constant parameters. As a check, substituting $\rho_c = -\frac{2}{3}F_{fid}^2$ from the top condition into the second, the coefficient of the $F_7^2$ term is then $-\frac{1}{6} < 0$, so the three conditions are consistent. With the parameters in $\Gamma$, the Dehn/Anderson filling [19], and $\tilde{N}_7$, we have the freedom to tune this. The mathematical discrete choices available to tune the third line in (5.21) are described above around (4.13) and in appendix A. We can change the volume, e.g. using k-fold covers of the manifolds [25], independently of the parameters chosen for the Anderson filling of cusps [19] which can be used in combination with the choice of $\tilde{N}_7$ to ensure the first two lines of (5.21).

With these relations we have established that the fiducial solution satisfying the constraint preserves our basic mechanism for stabilizing $\ell$. This fiducial configuration does not satisfy the other equations of motion; $V_{eff}[\delta B, h, C_6]$ will contain tadpoles generically. It is important to study the fate of these instabilities, taking into account the solution to the constraint as we move along the directions of the tadpoles.

Let us first treat this for the $C_6$ potential field. Here the tadpole arises from the difference between the fiducial configuration (5.5) we have used so far, and the $u$-dependent solution for $F_7$ described above in (3.10). We denote the flux quantum number of the latter by $N_7$, which does not need to be the same as $\tilde{N}_7$. Let us write

$$\hat{H}_{fid} = -\nabla_w^2 - \frac{1}{3}\left(-\ell^2 R^{(7)} + \ell_{11}^9 \ell^2 \rho_C + \frac{\ell^2}{2}|F_{fid}^{(7)}|^2\right). \tag{5.22}$$

Define $u_{0,fid}$ as the ground state of the Hamiltonian $\hat{H}_{fid}$, which is determined by a linear Schrödinger problem. Let us choose $\tilde{N}_7/\ell^7$ such that

$$0 < -\lambda_{0,fid} = \int \sqrt{g}\, u_{0,fid}\, \hat{H}_{fid}\, u_{0,fid} = \mathcal{O}(C\ell^2) \ll a. \tag{5.23}$$

This means that if $\hat{H}_{fid}$ is a good approximation to the effective Schrödinger Hamiltonian, the corresponding warp factor solution yields a small $V_{eff}$. To check this, let us write

$$\hat{H}[\tilde{u}] = -\nabla_w^2 - \frac{1}{3}\left(-\ell^2 R^{(7)} + \ell_{11}^9 \ell^2 \rho_C + \frac{\ell^2 \ell_{11}^{12}}{2}\frac{N_7^2}{\tilde{u}^4(\int \sqrt{g}\frac{1}{\tilde{u}^2})^2}\right) = \hat{H}_{fid} + (\hat{H}[\tilde{u}] - \hat{H}_{fid}). \tag{5.24}$$

Consider $\hat{H}[u_{0,fid}]$. Note that in the classically disallowed region where $u_{0,fid}$ decays exponentially in $\sqrt{|\rho_C|}$, the flux term in $\hat{H}[u_{0,fid}]$ remains bounded because the second factor in the denominator compensates for the first; indeed this term integrates to a magnitude smaller than the integrated fiducial flux term as we saw above in (3.16). We choose $N_7/\ell^7$ to tune

$$\int \sqrt{g}\, u_{0,fid}(\hat{H}[u_{0,fid}] - \hat{H}_{fid})^2 u_{0,fid} \ll 1, \tag{5.25}$$

ensuring that $(\hat{H}[u_{0,fid}] - \hat{H}_{fid})$ is a small perturbation, in the sense that in acting on the ground state $u_{fid,0}$ of $\hat{H}_{fid}$, the perturbation $(\hat{H}[u_{0,fid}] - \hat{H}_{fid})$ produces a vector of small magnitude. As such, it preserves the solution $u_{0,fid}$ to the constraint equation up to a small correction. Thus for the purpose of solving the constraint (3.7), we can view $\hat{H}[u_{0,fid}]$ as a small perturbation of $\hat{H}_{fid}$, with the latter more easily solvable as it contains constant background fields aside from the Casimir energy. In so doing, we are free to separately tune $\tilde{N}_7/\ell^7$ and $N_7/\ell^7$ to ensure that the configuration $\tilde{u} = u_{0,fid}$ in the full problem will give a small eigenvalue $\lambda_0$, and hence a small $V_{eff}$ (as in (5.16)). This feature, a small $V_{eff}$, is what entered into the $\delta\ell$ analysis; note that the tune (5.23) is the same as line 2 of (5.21).

Next we will study tadpoles for metric deformations in the Casimir region; in the following sections §5.2- §5.5 we will treat these in the bulk.

To analyze this concretely, we can return to the interpretation of $u$ as a wavefunction in a Schrödinger problem [17], described above in §3. An instability in the end region of the cusp would decrease the contribution to $V_{eff}$ there. But as the localized stress energy gets more negative, the 'wavefunction' $u$ dies more quickly. That reduces the magnitude of the negative contribution. As an extreme example we can immediately exclude an instability with energy in this region diverging $\to -\infty$: a vertical wall in the effective Schrödinger potential would then force $u$ to zero, entirely screening the would-be decay mode.

Analyzing this more generally, we find a stabilizing effect of the warping, as follows. The Casimir energy becomes more negative as we decrease the size $R_c = R_{c0}e^{\delta\sigma_c}$ of the smallest circle contributing to it. From the point of view of the naive potential, i.e. without solving the constraint equation for the warping, we would have relevant terms

$$V_{naive} \sim \int \sqrt{g^{(7)}}\left(c_1 e^{-11\delta\sigma_c}\rho_{c0}(y_i) + c_2\delta\sigma_c^2 + \dots\right), \tag{5.26}$$

where $c_1$ and $c_2$ are positive constants. Here, the first term is the Casimir energy $\rho_C \sim e^{-11\delta\sigma_c}\rho_{c0}(y_i)$, with $\rho_{c0}(y) < 0$ calculated from (4.2) in the fiducial Einstein metric. The second term represents the positive mass squared from the curvature term that we get in the hyperbolic metric [16], something we will discuss in the next section, along with any gradient energy in the modes of $\delta\sigma_c$. This naive potential by itself would have a runaway instability to negative $\delta\sigma_c$ even perturbatively, despite the Hessian from the curvature.

The full effective potential is instead of the form

$$V_{eff} \propto \int \sqrt{g^{(7)}}u[\delta\sigma_c]^2\left(c_1 e^{-11\delta\sigma_c}\rho_{c0}(y_i) - 3\left(\frac{\nabla u[\delta\sigma_c]}{u[\delta\sigma_c]}\right)^2 + c_2\delta\sigma_c^2 +\right)\dots, \tag{5.27}$$

where $u$ depends on $\delta\sigma_c$ via the constraint, reproduced here:

$$\left(-\nabla^2 - \frac{1}{3}\left(-R^{(7)} + \ell_{11}^9\rho_C + \frac{1}{2}|F_7|^2\right)\right)u = -\frac{C}{6}, \tag{5.28}$$

and we have in mind working at small $C$ as discussed above. The instability suggested in (5.26) increases the Schrödinger potential barrier in this equation, forcing $u$ to fall faster in the classically disallowed region. The effective energy in the Schrödinger problem is much smaller than the potential barrier from $-\ell_{11}^9\rho_C$, so we can use the WKB approximation under the barrier. The solution for $u$ dies like

$$\exp(-|\rho_c|^{1/2}) \sim \exp\left(-\int e^{-11\delta\sigma_c/2}|\rho_{c0}(y_i)|\right), \tag{5.29}$$

doubly exponentially in $\delta\sigma_c$ in the classically disallowed region. All of the terms in $(c_1 e^{-11\delta\sigma_c} - 3(\frac{\nabla u[\delta\sigma_c]}{u[\delta\sigma_c]})^2 + c_2\delta\sigma_c^2)$ are at most singly exponentially growing. The resulting contribution they make to the spacetime effective potential in the Casimir region is of the form

$$\sim \int_y -e^{-2\sqrt{\kappa_c\gamma}}\kappa_c\gamma + \left(\frac{\log\gamma}{11}\right)^2, \tag{5.30}$$

with $\gamma \sim -\rho_{c0}(y_i)e^{-11\delta\sigma_c}$ and $\kappa_c$ a positive constant. This leads to a potential which has a minimum rather than a runaway. To be more precise, we can start from the effective potential (3.9) and vary it with respect to a mode of $\delta\sigma_c$ with support in the Casimir region and also the bulk (if it had support in just the Casimir region, its Kaluza-Klein mass would be large,

stabilizing even the naive potential (5.26)). In the full system, we can view $V_{eff}$ as a functional of both $u$ and deformations such as $\sigma_c$ and write

$$\frac{dV_{eff}}{d\delta\sigma_c} = \frac{\partial V_{eff}}{\partial u}\frac{du}{d\delta\sigma_c} + \frac{\partial V_{eff}}{\partial\delta\sigma_c} = \frac{\partial V_{eff}}{\partial\delta\sigma_c}, \qquad (5.31)$$

evaluated at the solution of the constraint, $u = u|_c$, which kills the first term in the middle expression here. This takes the schematic form

$$V' \sim 11e^{-2\sqrt{\kappa_c\gamma}}\kappa_c\gamma + 2\frac{\log\gamma}{11^2}, \qquad (5.32)$$

coming from the explicit $\delta\sigma_c$ dependence in the Casimir and mass terms in $V_{eff}$. This expression has a zero at positive $\gamma$ corresponding to a local minimum of $V_{eff}$. Both the warping and the mass term [16] (to be discussed further below in §5.4) play a role in this. We note that the constants here are such that the Casimir contribution (5.30) is net negative, since it arises from a tadpole direction, descending in $V_{eff}$ until $u$ drops below the Schrodinger barrier and yields the stabilization mechanism (5.30). In the full backreacted patch solution in §5.3 other fields beyond the warp factor play a role, exhibiting a stabilized value of $R_c$.

As mentioned above, our expression for the Casimir energy in terms of $R_c$ (4.2) assumes that the warp factor gradient $A' = u'/2u$ does not strongly affect the behavior of the modes that enter into the Casimir stress-energy calculation. Given that the potential (5.27) quickly suppresses deformations for which the wavefunction dies rapidly under the barrier, we do not expect strong variation of $u$. We will verify this, and its consistency with the competitive contribution of the warp factor gradient in suppressing our parameter $a$ (4.6) below in §5.1.2.

The result is a bounded, but sufficiently strong, source of negative Casimir energy in our problem. This analysis is reasonably model-independent, and applies to small systoles as in the 'inbreeding' construction [20–22] or Anderson filling [19].

One may also consider a full hyperbolic cusp where the fiducial metric contains arbitrarily small cycles in the $T^6$ cross section. This setup by itself produces an ultimately singular geometry, albeit screened by a rapidly dying warp factor. It would be interesting to resolve such singularities, which might eliminate the need for filling the cusps at some finite $y_f$. In the next subsections we will illustrate the behavior of the internal fields, and the integrated quantities appearing in $V_{eff}$, using the cut off cusp metric (4.1) as a mockup of a filled cusp.

### 5.1.1 Fiducial solution in a cut off cusp

To illustrate some of the features just described in a technically simpler setting, consider the geometry (4.1), with a fiducial constant flux. This yields a concrete solution to the constraint (3.7) in terms of Bessel functions, which we can use to assess the contributions of the various integrated quantities appearing above in (5.21). We will repeat this exercise in a toy model of a Anderson/DF geometry in §5.1.2 and in a fully backreacted cusp geometry in §5.3.

The region (4.1) contains a part in which the Casimir energy density is of order $-1/R_c(y_c)^{11}$, and for sufficiently small $y_0$ it also has a region where the Casimir energy becomes subdominant. The constraint equation (3.7) becomes simply

$$-\frac{1}{6}C = \left(-\partial_y^2 + \frac{6}{\ell}\partial_y + \frac{1}{3}[R^{(7)} - \frac{1}{2}|F_7|^2 + \ell_{11}^9\frac{1}{4}Tr_{(4d)}T_{Cas}]\right)u, \qquad (5.33)$$

$$-\frac{1}{6}C\ell^2 = \left(-\partial_w^2 + 6\partial_w + [-\tilde{a} + \tilde{b}e^{11w}]\right)u, \quad w = y/\ell, \quad \tilde{a} = 14 + \frac{(1/6)N_7^2}{\nu_7^2\hat{\ell}^{12}}, \quad \tilde{b} = \hat{\ell}^2\frac{c_\rho}{\hat{\lambda}_c^{11}},$$

where $C$ is the 4d curvature and $c_\rho$ is an order one positive constant.

As explained above, we can work in a regime where $V_{eff}$ is much smaller than its individual contributions, and we can neglect the $C$ term on the left hand side here. Then the general solution is given in terms of Bessel functions.

The correct linear combination which exhibits the decay in $u$ at the end is given in terms of the modified Bessel function of the second kind $K_\nu$:

$$u = N_u e^{3w} K_\nu(z), \tag{5.34}$$

with

$$z = \frac{2}{11}\sqrt{\tilde{b}}\,e^{11w/2}, \quad \nu = \frac{2}{11}\sqrt{9-\tilde{a}} = \frac{2}{11}i\sqrt{5+\frac{(1/6)N_7^2}{v_7^2\hat{\ell}^{12}}}. \tag{5.35}$$

The normalization $N$ in (5.34) can be traded for $1/G_N$ as in (1.3). From this solution (5.34), we can study the cusp contribution to integrated quantities appearing in our de Sitter mechanism, as summarized in (5.21). From this one finds a variety of regimes of positive, tunable values of $a$ at the level of the analysis of this section (incorporating the warp factor variation but not yet all fields). This includes regimes where $a$ is much less than the integrated curvature, and the integrated Casimir energy competes with the net $\int\sqrt{g^{(7)}}u^2[-R^{(7)}-3(\frac{\nabla u}{u})^2]$. Moreover, we obtain a finer tuning of the integrated Casimir energy by varying $y_c$, which mocks up the choice of simple closed geodesic determining the size of the $T^6$ at where the filling geometry starts to deviate from the cusp for Anderson's Dehn filling of the cusp [19] (see appendix A.2).

### 5.1.2 Fiducial solution in a simple filled region

In the previous sections, we have described general features of the Schrödinger problem associated to the constraint equation (3.7) and analyzed it in a cut off cusp, where analytic results can be obtained. There we mocked up the presence of a minimal circle in [19] (figure 14) with the cutoff $y_c$ in (4.1). In this section, we analyze the constraint equation in a smooth geometry somewhat closer to the filled geometry of [19]. Here we continue to focus on the warping generated by the constraint equation; below in section 5.3 we will treat the fully backreacted cusp. For the present section, we replace the cusp geometry (4.1) by the line element

$$ds^2_{\text{filled-cusp}} = dy^2 + R_c^2 ds^2_{T^{(n-2)}} + R^2 d\theta^2, \tag{5.36}$$

where we have isolated one of the circles from the $(n-1)$-dimensional transverse torus in (4.1). In this filled geometry the $\theta$ circle shrinks smoothly at a certain $y = y_{\text{DF}}$, where the volume of the now $(n-2)$-dimensional transverse torus reaches its minimal value. Explicitly, this is realized by the functions

$$R_c = R_c(0)e^{\frac{y}{\ell}}\left(\frac{e^{-\frac{(n-1)(y-y_{\text{DF}})}{\ell}}+1}{e^{\frac{(n-1)y_{\text{DF}}}{\ell}}+1}\right)^{\frac{2}{n-1}},$$

$$R = R(0)e^{\frac{y}{\ell}}\left(\frac{1-e^{-\frac{2(n-1)(y-y_{\text{DF}})}{\ell}}}{1-e^{\frac{2(n-1)y_{\text{DF}}}{\ell}}}\right)^{\frac{1}{n-1}}\left(\frac{\tanh\left(\frac{(n-1)(y-y_{\text{DF}})}{2\ell}\right)}{\tanh\left(\frac{(1-n)y_{\text{DF}}}{2\ell}\right)}\right)^{\frac{n-2}{n-1}}. \tag{5.37}$$

Here we introduced a parameter $y_{DF}$. This would be determined by matching to the bulk manifold; it is not a free parameter. In the more general construction of [19], there is a discrete parameter that plays a similar role: a choice of simple closed geodesic determines how far down the cusp the filling deviates significantly from the cusp metric. Note that in that construction, however, the metric (A.5) is not diagonal as it is here in (5.36). In this section, we will work with a $y_{DF}$ of order $\ell$ and explore the behavior of the constraint equation (3.7) in

the background (5.36). This by itself – without special tuning of $y_{DF}$ – will yield results similar to what we require for our mechanism, although the ability to tune via the discrete choices described around (5.25) and in appendix A ensures that each of our conditions (5.21) can be met. Locally, (5.36) and (5.37) describe an $n$-dimensional Einstein space with curvature normalized as $R_{(n)} = -\frac{n(n-1)}{\ell^2}$. To guide the intuition, we notice that for $n = 3$ the expressions (5.37) simplify to $R_c^2 \sim \cosh^2(\frac{y-y_{DF}}{\ell}), R^2 \sim \sinh^2(\frac{y-y_{DF}}{\ell})$, which for $\frac{y_{DF}}{\ell} \gg 1$ and $y \ll y_{DF}$ reduce to the single exponentials in the cusp geometry (4.1). This is true for any $n \geq 3$, and from now on we specialize the discussion to $n = 7$. An example of the resulting geometry is plotted in Figure 4a. In a region near small $y$ we glue (5.36) to the bulk of the hyperbolic manifold. The boundary conditions for gluing hyperbolic polygons are naturally set at their facets, which are totally geodesic submanifolds [15]. As reviewed in more detail in appendix A, these are vertical walls and hemispheres centered at $z = 0$ (and any $\vec{x}$) in the upper half space model $ds^2 = \frac{dz^2 + d\vec{x}^2}{z^2}$, with $y = \log(z)$. A submanifold at constant $y$ is not totally geodesic, and a proper gluing will require to introduce extra angular dependencies, which we are going to neglect in this section.

In the compact space defined by (5.36), (5.37) for $0 \leq y \leq y_{DF}$ we will now study the Schrödinger equation described around (3.17), which we repeat here for convenience of our reader

$$(\Delta + V)u_i = \lambda_i u_i,$$
$$V = \frac{1}{3}\left[R^{(7)} - \frac{1}{2}|F_7|^2 + \frac{\ell_{11}^9}{R_c^{11}}\right], \tag{5.38}$$

where $\Delta \equiv -\nabla^2 = -\frac{1}{\sqrt{g^{(7)}}}\partial_m\left(\sqrt{g^{(7)}}g^{(7)mn}\partial_n\right)$ and we use the fiducial flux $|F_7|^2 = \ell_{11}^{12}\frac{N_7^2}{\text{Vol}_7^2}$. We have also set to 1 the constant coefficient in front of the Casimir term. We normalize the eigenfunctions $u_i$ as in (5.12): $\delta_{ij} = \langle u_i, u_j \rangle \equiv \frac{1}{\text{Vol}_7}\int_M \sqrt{g^{(7)}}u_i u_j$. Given the eigenstates $u_i$, the solution $u$ to the constraint equation (3.7) is given by (5.15).

Before resorting to numerics, let us recall here a few elementary properties of Schrödinger operators on compact spaces. First of all, we have the lower bound

$$\lambda_i = \lambda_i \frac{1}{\text{Vol}_7}\int_M \sqrt{g^{(7)}}u_i^2 = \frac{1}{\text{Vol}_7}\int_M \sqrt{g^{(7)}}u_i(\Delta + V)u_i$$
$$= \frac{1}{\text{Vol}_7}\int_M \sqrt{g^{(7)}}|\nabla u_i|^2 + \frac{1}{\text{Vol}_7}\int_M \sqrt{g^{(7)}}Vu_i^2 \tag{5.39}$$
$$\geq \frac{1}{\text{Vol}_7}\int_M \sqrt{g^{(7)}}Vu_i^2.$$

Since for small $\lambda_0$ the sum in (5.15) is dominated by the ground state (see also Figures 4d, 4e), in order to have a positive $u$ for dS ($C > 0$) we need $\lambda_0 < 0$. From (5.39) we see that this requires $V$ to be negative in some regions. In the full geometry this happens in the bulk of the manifold, where the Casimir energy does not contribute (cf. figure 2). In the local region we study in this section, the filled geometry (5.36), this is true near $y = 0$, where the Casimir contribution to the potential is suppressed as $\ell^2\ell_{11}^9 R_c(0)^{-11}$. Also, from the variational principle we have

$$\lambda_0 = \min_\varphi \frac{\int_M \sqrt{g^{(7)}}(|\nabla\varphi|^2 + V\varphi^2)}{\int_M \sqrt{g^{(7)}}\varphi^2} \leq \frac{1}{\text{Vol}_7}\int_M \sqrt{g^{(7)}}V, \tag{5.40}$$

meaning that we also require $V$ to be not too negative, in order to allow for $\lambda_0 \sim 0$. In Figure 4b we show that the potential (5.38) specialized to the geometry (5.36), (5.37) has all these features.

We now solve numerically the Schrödinger problem (5.38) in the geometry (5.36), (5.37). First of all, we need to impose boundary conditions. At $y = 0$ everything goes to a constant, and we impose Neumann boundary conditions to match to a solution $u \sim$ const. in the bulk. This boundary condition is consistent with the fact that we have discarded boundary terms at $y = 0$ in the expressions (5.39),(5.40). Approaching $y = y_{DF}$, the volume density vanishes as $\sqrt{g^{(7)}} \sim c_1(y - y_{DF}) + c_2(y - y_{DF})^3$, for some constants $c_i$. Assuming also for the eigenfunctions $u_i$ a power-like behavior $u_i \sim u_{i,0}|y - y_{DF}|^\alpha + u_{i,1}|y - y_{DF}|^\beta + \dots$, by plugging it in the Schrödinger equation (5.38) we obtain a single class of solutions

$$u_i \sim u_{i,0} + u_{i,1}(y - y_{DF})^2 + \dots. \tag{5.41}$$

Thus, we impose Neumann boundary conditions also at $y = y_{DF}$. We note that this behavior is familiar from textbook quantum mechanics problems with a finite radially symmetric potential barrier.

Working in 11d Planck units, the Schrödinger problem (5.38) reads

$$\hat{\Delta} u + \frac{1}{3}\left[-\frac{42}{\hat{\ell}^2} - \frac{1}{2}\hat{f}_7^2 + \frac{1}{\hat{R}_c^{11}}\right]u = \hat{\lambda}_i u, \tag{5.42}$$

where $\hat{\Delta}$ and $\hat{R}_c$ are constructed from (5.36) (5.37) with $\hat{y} \equiv y/\ell_{11}$. Similarly, $\hat{f}_7^2 \equiv \ell_{11}^{14}\frac{N_7^2}{Vol_7^2} \equiv \frac{N_7^2}{v_7^2\hat{\ell}^{14}}$ and $\hat{\lambda}_i \equiv \ell_{11}^2\lambda_i$. Notice that equation (5.42) is invariant under the parametric rescaling

$$\hat{y} \to e^c\hat{y}, \qquad \hat{\ell} \to e^c\hat{\ell}, \qquad \hat{R}_c \to e^{\frac{2}{11}c}\hat{R}_c, \qquad \hat{f}_7 \to e^{-c}\hat{f}_7, \qquad \hat{\lambda}_i \to \hat{\lambda}_i e^{-2c}, \tag{5.43}$$

for a real constant $c$. The rescaling of $\hat{R}_c$ is obtained by also rescaling $y_{DF} \to e^c y_{DF}$ and $R_c(0) \to e^{\frac{2}{11}c}R_c(0)$ in (5.37). Thus, acting with $c \gg 1$ on a given solution of (5.42) produces a new one with hierarchy $\hat{\ell} \gg \hat{R}_c \gg 1$, tuning at the same time $\hat{\lambda}_i \to 0$. The relative scaling under $c$ of the various terms in the Schrödinger potential in (5.42) is consistent with the parametric estimates (2.7) and (2.9). Finally, notice that this rescaling is quantized, since it also acts on $N_7$. For this reason it can also be used to generate a solution with an integer $N_7$ starting from a non-quantized one. In Figure 4 we show a particular numerical solution of the Schrödinger problem (5.42). Evaluating on this numerical solution we obtain

$$\frac{\int \sqrt{g^{(7)}}u^2\left[-R^{(7)} - 3\left(\frac{\nabla u}{u}\right)^2 + \frac{1}{2}|F_7|^2\right]}{-\int \sqrt{g^{(7)}}u^2\,\ell_{11}^9 R_c^{-11}} \sim -1, \tag{5.44}$$

confirming a competitive Casimir energy. Also, the ratio

$$\frac{\int \sqrt{g^{(7)}}u^2\left[-R^{(7)} - 3\left(\frac{\nabla u}{u}\right)^2\right]}{-\int \sqrt{g^{(7)}}u^2 R^{(7)}} \sim .04 \tag{5.45}$$

shows that already a single filled cusp contains a nontrivial contribution from the warp factor gradient contribution to $V_{eff}$, here reducing $a$ (4.6) by a significant amount, while keeping it positive. At the same time, this solution illustrates that the warp factor gradient does not significantly affect the Casimir energy calculation (4.2). For this we need to check that the radial friction term in the equation of motion for the field fluctuations does not compete with the dominant modes of momentum $\sim 1/R_c$ that generate the stress energy (4.2). This condition

$$\frac{1}{\ell}\frac{u'}{u} \ll \frac{1}{R_c^2} \tag{5.46}$$

is satisfied in the present solution (as anticipated in the discussion above around (5.30)). In our full compactification, we make use of geometric choices to tune $a$ as discussed above and in appendix §A.

As a technical note, we add that thinking about the (linear) Schrödinger problem helps with finding numerical solutions with the required physical properties, especially since we have shown in our perturbation theory analysis above that the ground state is a very good approximation to the full solution. Indeed, an alternative way to look for numerical solutions to the constraint equation (3.7) in the filled geometry (5.36), (5.37) could have been to tune the radially-initial parameters in order to directly solve the non-linear equation shooting from $y = 0$. However, this problem is complicated by the fact that for generic values of the parameters the solution for $u$ usually blows up. A similar phenomenon would appear trying to solve the Schrödinger equation as an initial value problem with the wrong energy, since only for a measure-zero set of $\lambda_i$ the Neumann boundary problem admits a solution. On the other hand, solving the linear Schrödinger equation directly as a boundary problem, and using it as seed for the non-linear problem, allows us to quickly find smooth solutions to the full non-linear integro-differential equation (3.17), as we show in the example in Figure 5. This confirms that there are no obstructions to finding smooth solutions to the non-linear constraint equation (3.17) arising when the flux is properly taken into account. Moreover, in the solution in Figure 5, it is still true that the Casimir term competes with the classical ones in the potential since

$$\frac{\int \sqrt{g^{(7)}} u^2 \left[ -R^{(7)} - 3 \left( \frac{\nabla u}{u} \right)^2 + \frac{1}{2} |F_7|^2 \right]}{-\int \sqrt{g^{(7)}} u^2 \ell_{11}^9 R_c^{-11}} \sim -1 \,. \tag{5.47}$$

The next step would be to solve numerically all the equations of motion. This would incorporate the shifting due to the metric tadpoles, whose effect we have bounded in the rest of the work. This is harder to do explicitly since, as we noted above, already the fiducial configuration would require us to work with a more complicated setup that includes the angular dependence. Failing to do so could produce spurious singularities, similarly to the untuned situation discussed above. In Section 5.3, however, we will present a nonsingular full backreacted solution in an example of a complete filled cusp (without the general angular dependence of [19]), and in Section 8 we will discuss how neural network methods can be applied to this problem; it is well posed for complete filled hyperbolic manifolds obtained by joined polygons.

## 5.2  Warp and conformal factors sourced by localized Casimir

Casimir stress-energy (4.2) is localized in our compactifications, near a systole such as those in [20–22] or down a filled hyperbolic cusp. In this subsection, we will treat the fields in the bulk regions which are sourced by the concentrated Casimir energy. In the case of a cusp, we may more specifically separate our system into two parts: the central manifold combined with any region in the bulk of the cusp for which $\ell_{11}^9 |\rho_C| \ll -R^{(7)}$, and the end of the cusp where $|\rho_C|$ competes with the other energy sources.

In order to have a valid de Sitter compactification, there must exist a pointwise solution to the equations of motion in the internal geometry. This requires nonzero gradients of the warp factor and conformal factor in (1.1), as we saw above in (4.12) and will describe in detail below in §5.2.1. The physical reason for this is straightforward for both the warp factor $u = e^{2A}$ and the conformal factor. Around any fiducial metric, the warp factor includes the Newtonian potential sourced by the localized Casimir stress-energy. In particular, the constraint equation takes the form:

$$\nabla^2 \delta A \sim (\rho_C - \bar{\rho}_C) + \mathcal{O}((\nabla A)^2) \,, \tag{5.48}$$

where $\delta A$ denotes the deformation away from the fiducial metric with constant $A = A_0$. The simplest regime – which we will see shortly applies in the bulk of our setup – is one where

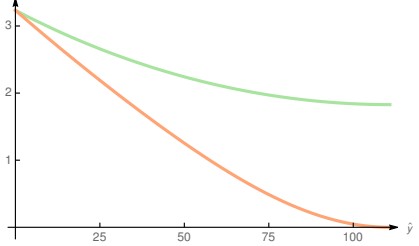

(a) Background geometry: the Casimir circle $\hat{R}_c^2$ (green), and the smoothly-shrinking circle $\left(\hat{R}\frac{R_c(0)}{\hat{R}(0)}\right)^2$ (orange).

(b) Different contributions to the Schrödinger potential in units of $\ell_{11}$

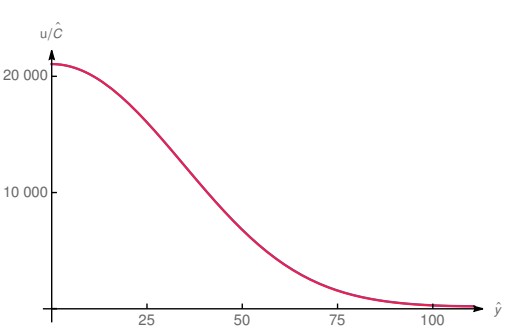

(c) First four normalized eigenstates and their energies.

(d) The contribution of the ground state (blue) to (5.15) is indistinguishable from the sum of the first ten eigenfunctions (red).

(e) A close-up near the end point of the partial sums in panel (d).

Figure 4: An instance of the Schrödinger problem (5.42) with $\hat{\ell} \sim 221$, $\hat{y}_{\mathrm{DF}} = \hat{\ell}/2$, $N_7/v_7 \sim 2 \times 10^{15}$. In panel 4a we see the slowly-varying Casimir circle $\hat{R}_c$, and the smoothly-shrinking circle $\hat{R}$. Comparing with 4b, we see that when $\hat{R}_c$ decreases its contribution to the potential gives rise to a finite potential barrier. The total potential has a slightly negative region, compatible with the bounds in the main text. In panel 4c we show the first eigenstates. From panels 4d, 4e we see that the ground state is a very good approximation to $u$, and from the close-up in panel 4e we can check that $u$ stays finite on the smooth point.

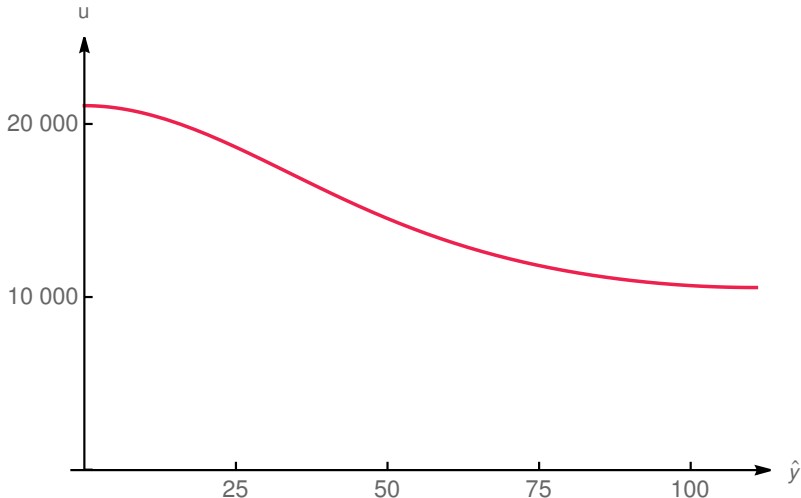

Figure 5: A smooth solution to the non-linear version of the Constraint equation (3.17) in the filled the geometry (5.36).

the $\mathcal{O}((\nabla A)^2) \sim (\frac{\nabla u}{u})^2$ contribution is subdominant to the other contributions. This means that bulk contributions to the integrated $(\frac{\nabla u}{u})^2$ term in $V_{eff}$ are subdominant to the other terms in (3.4), meaning that the gradients of the sourced field $A$ do not introduce significant corrections to the 4d theory from the bulk regions of the manifold, only from the Casimir-dominated regions in the way we saw above.

The basic physical reason for variation of the conformal factor $B$ in (1.1) relative to the hyperbolic metric is the following. $B$ includes the overall volume mode $B_0$, which is stabilized at the level of the averaged sources. Writing $B = B_0 + \delta B$, the deformations $\delta B$ will be subject to opposite forces in the region dominated by Casimir energy as compared to the regime dominated by curvature and flux. Negative fiducial curvature $R_{\mathbb{H}}^{(7)}$ and flux each push the compactification toward larger volume, whereas the negative Casimir energy pushes it toward smaller volume. Since these dominate in different regions (the bulk versus the end of the cusp), gradients of $B$ will be induced.

Now let us explain why we can remain in the simplest regime where $\nabla^2 A > (\nabla A)^2$. We start by comparing to flat spacetime of size $\ell$ with a localized source. In that case, dimensional analysis – consistently with a calculation along the lines of [8] via the sum of power-law Green's functions arising in flat space – estimates this ratio to be of order 1, suggesting no such hierarchy. A naive generalization to our case would give the same result, as follows. In the bulk of the space, the Casimir contribution is absent. As a result, in that region only the homogeneous terms $\frac{1}{3}\left(R^{(7)} - \frac{1}{2}|F_7|^2\right)$ contribute to the effective Schrödinger potential in (3.7). Using the fact that this is of order of the curvature contribution in our stabilization mechanism, we must have $\nabla^2 \delta A \sim |R^{(7)}| \sim 1/\ell^2$ in order for it to balance against the other terms in the equations away from the Casimir source. By dimensional analysis, this entails $\delta A$ of order 1, and so $\nabla^2 \delta A \sim 1/\ell^2 \sim (\nabla \delta A)^2$.

However, for the hyperbolic case, there is an extra effect that helps to suppress $\delta A$ away from the localized sources: the spreading of geodesics in hyperbolic geometry encoded in the propagator $(\nabla^2)^{-1}$. For example, in the cusp the geodesics spread exponentially as we move toward the central manifold from the end where the Casimir energy dominates. This suppresses the Green's functions compared to the power law ones we just discussed in the flat space analogue, with an IR regulating effect as in the treatment of Euclidean field theory on hyperbolic space in [57]. The fully backreacted solution of §5.3 will combine a region of negative curvature in the approximate cusp joined to a central part where the internal curvature

becomes positive. In this latter region the warp factor solution will be near a minimum and hence still $(\nabla A)^2 < \nabla^2 A$.

### 5.2.1 Details of warp factor in bulk and the regime $\nabla^2 A > (\nabla A)^2$

Let us now check the previous intuition in the bulk of the cusp where the Casimir energy is negligible. We focus on the warp and conformal factors $A$ and $B$ in (1.1), and will present an analysis of tadpoles in the other directions $h$ below in §5.5. We work in terms of variables $u$ and $v$

$$u = e^{2A}, \qquad v = e^{\frac{5}{2}B}, \tag{5.49}$$

with the equations for $u$ and $v$ (equivalently $A$ and $B$) in the form

$$-\frac{\nabla^2 u}{u} + \frac{R^{(7)}}{3} - \frac{1}{6}F_7^2 + \frac{C}{6u} \approx 0, \tag{5.50}$$

$$-R^{(7)} + \frac{9}{5}(\frac{\nabla u}{u})^2 - \frac{7}{10}F_7^2 + \frac{24}{5}\frac{\nabla^2 u}{u} - \frac{7}{5}\frac{C}{u} \approx 0, \tag{5.51}$$

with as above $C = R^{(4)}_{\text{symm}}$. To specialize these equations to the $h = 0$ case of (1.1), with the internal metric conformally hyperbolic, we replace

$$R^{(7)} \to v^{-4/5}(R^{(7)}_{\mathbb{H}} - \frac{24}{5}\frac{\nabla^2_{\mathbb{H}} v}{v}), \tag{5.52}$$

and

$$\nabla^2 f \to v^{-\frac{4}{5}}(\nabla^2_{\mathbb{H}} f + 2\frac{\nabla_{\mathbb{H}} v}{v}\nabla_{\mathbb{H}} f). \tag{5.53}$$

Note that these bulk equations do not admit a physical solution (with real flux) with exactly constant $u > 0$. This reflects the fact that the 3-term solution from the averaged sources in §2 requires the Casimir contribution, but in bulk it is not present. This is a familiar feature in top down models, and is what leads to the need for warp factor variations in the bulk geometry [8,17]; see also [23,58]. However, as described above and articulated in general terms in [8], there is a distinction between a regime where all the gradient terms participate equally and a regime where $\nabla^2 A \gg (\nabla A)^2$ (or the opposite, WKB like regime where $(\nabla A)^2 \gg \nabla^2 A$). It is our purpose in this section to assess which of these regimes we are in in the bulk of our compactification.

We can assess this within the simpler geometry of the cusp (4.1), since if $\delta A$ has died down in that region, then it is small at the matching to the central manifold, indicating that the hyperbolic suppression of its response function has been realized. The equations simplify further if we seek a solution which respects a translation symmetry along the directions of the cusp cross section $T^6$ transverse to the radial $y$ direction (4.1). The Casimir stress energy has this property, so this is consistent on that end. Dependence on the $T^6$ directions is ultimately required to extend our solution into the central manifold. One can organize that problem in terms of hyperbolic polygons which join together to form the complete internal manifold. The boundaries of these polygons are totally geodesic submanifolds, unlike the constant $y$ surface. However, incorporating the appropriate Fourier modes on the $T^6$ direction does not spoil the overall suppression of $\delta A$ far from the localized Casimir source which is the aim of the present analysis. Those are like masses and should further suppress $\delta A$ as we propagate farther from the source.

These equations can be solved numerically, but similar results are obtained analytically with a few simplifications. We keep the contributions from $\nabla^2 u/u$ and $\nabla^2 v/v$ but not the quadratic gradient terms, and then check that the latter terms would be subdominant in the

solution. We also neglect the nonlinear factors from $v^{4/5}$. With these specifications, we can solve (5.50) and (5.51) algebraically for $\nabla^2 u/u$ and $\nabla^2 v/v$, finding

$$\frac{\nabla_{\mathbb{H}}^2 u}{u} = \frac{Cv^{4/5}}{2u} + \frac{2F_7^2}{3}, \qquad \frac{\nabla_{\mathbb{H}}^2 v}{v} = -\frac{5Cv^{4/5}}{24u} - \frac{25F_7^2}{48} + \frac{5R_{\mathbb{H}}^{(7)}}{24}. \tag{5.54}$$

Although we have neglected the Casimir source since we are working in the bulk of the cusp, it affects the solution we pick for $u$ and $v$ via the need to match those to the decaying warp factor solution in the Casimir region near the end of the cusp as described above in §5.1.

From (4.12) we should obtain positive curvature in this region since we have negligible Casimir energy. This is indeed true as follows immediately by combining (5.53) and (5.52). We would like to solve these equations. Before doing that, we can make another simplification and neglect the terms proportional to $C$ here, using our 3-term structure as explained above in sections §2 and 4. For now, we will also treat the $F_7^2$ term as a constant, and check later that this is a reasonable approximation even in the full zero mode solution (3.10), (3.11).

With these additional simplifications, the equations to solve are

$$u''(y) - 6u'(y) = \frac{2}{3}\ell^2 F_7^2 u(y), \tag{5.55}$$

and

$$v''(y) - 6v'(y) = -\frac{25}{48}\ell^2 F_7^2 v(y) + \frac{5}{24}\ell^2 R_{\mathbb{H}}^{(7)} v(y). \tag{5.56}$$

These equations have solutions

$$u(y) = a_1 e^{3\frac{y}{\ell}\left(1-\sqrt{1+\frac{2\ell^2 F_7^2}{27}}\right)} + a_2 e^{3\frac{y}{\ell}\left(1+\sqrt{1+\frac{2\ell^2 F_7^2}{27}}\right)}, \tag{5.57}$$

and

$$v(y) = b_1 e^{3\frac{y}{\ell}\left(1-\sqrt{1-\frac{25\ell^2 F_7^2}{432}+\frac{5\ell^2 R_{\mathbb{H}}^{(7)}}{216}}\right)} + b_2 e^{3\frac{y}{\ell}\left(1+\sqrt{1-\frac{25\ell^2 F_7^2}{432}+\frac{5\ell^2 R_{\mathbb{H}}^{(7)}}{216}}\right)}, \tag{5.58}$$

where $a_1, a_2, b_1,$ and $b_2$ are integration constants. We choose the $a_2 = 0$ solution for which the warp factor $u$ decreases as we go further down the cusp in the direction of increasing $y$.

In this solution, we can evaluate the key ratio $(\nabla A)^2/\nabla^2 A$, or its equivalent in terms of $u$, finding

$$\frac{u'(y)^2}{u(y)(u''(y)-\frac{6}{\ell}u'(y))} \sim \mathcal{O}(1/10), \tag{5.59}$$

for the range of applicable flux values.

In these equations, we have neglected not only the $\frac{(\nabla_{\mathbb{H}}u)^2}{u^2}$ term (which we just explicitly showed is subdominant), but also the cross term $\frac{\nabla_{\mathbb{H}}u}{u}\frac{\nabla_{\mathbb{H}}v}{v}$ which appears in $\nabla_{\mathbb{H}}^2 u$ via (5.53). We can similarly check whether this is self-consistently small in our current solution. We find

$$\frac{u'(y)v'(y)}{v\nabla_{\mathbb{H}}^2 u} \sim -0.4 \tag{5.60}$$

in this solution. This is not as small as the ratio (5.59). We can address this by generalizing the equation (5.55) to include this cross term, with $v'$ introducing radial friction. That equation is also solvable and reproduces the small ratio (5.59). We also find that the $\propto 1/u^4$ behavior of the flux term in the zero mode solution does not affect this qualitative result: it varies slowly throughout the bulk of the cusp, so the approximation of constant $F_7^2$ in the above equations is justified. Numerical solutions without the simplifying approximations taken here give qualitatively similar results.

## 5.3 Full backreacted solution in the cusp region

In this section, we solve the full set of equations of motion in the cusp and near-cusp region. We will do it by employing a radial ansatz, where we assume that all the functions only depend on the coordinate which extends longitudinally in the cusp, and we will work with a special Dehn filling in which the cross sectional torus remains rectangular throughout, though the general case includes a radial varying shape [19] as reviewed below in appendix A. This radial approximation breaks down going towards the central manifold, where the transverse gradients of the metric functions are important. Nonetheless it allows us to capture the main deviations from the fiducial hyperbolic metric in the cusp and near-cusp region, confirming at the same time the existence of smooth solutions to the full set of equations of motion with a strong Casimir contribution and with net positive parameter $a$ (4.6). The analysis here will be similar to the one in Section 5.1.2 but it will deviate from it by the fact that we will now allow the cusp geometry to backreact, by solving the complete set of equations of motion. We thus start again with the metric (5.36), which we rewrite here in full for convenience

$$ds_{11}^2 = e^{2A}ds_4^2 + dy^2 + R_c^2 ds_{T^5}^2 + R^2 d\theta^2 \,. \tag{5.61}$$

The equations of motion are now composed of the radial constraint

$$0 = 4A'\left(\frac{5R_c'}{R_c} + \frac{R'}{R}\right) + 6(A')^2 - \frac{1}{4}e^{-8A}f_0^2 - \frac{1}{2}e^{-2A}C + \frac{5R'R_c'}{RR_c} - \frac{|\rho_c|}{2R_c^{11}} + \frac{10(R_c')^2}{R_c^2} \,, \tag{5.62}$$

and the second order equations for the metric fields, which can be organized as

$$A'' = -A'\left(4A' + \frac{5R_c'}{R_c} + \frac{R'}{R}\right) + \frac{1}{3}e^{-8A}\left(\frac{3}{4}e^{6A}C + f_0^2\right) - \frac{|\rho_c|}{2R_c^{11}} \,, \tag{5.63}$$

$$\frac{R_c''}{R_c} = -\frac{R_c'\left(4A' + \frac{5R_c'}{R_c} + \frac{R'}{R}\right)}{R_c} - \frac{1}{6}e^{-8A}f_0^2 + \frac{3|\rho_c|}{5R_c^{11}} + \frac{(R_c')^2}{R_c^2} \,, \tag{5.64}$$

$$\frac{R''}{R} = -\frac{R'\left(4A' + \frac{5R_c'}{R_c} + \frac{R'}{R}\right)}{R} + \frac{1}{6}\left(-e^{-8A}f_0^2 - \frac{3|\rho_c|}{R_c^{11}}\right) + \frac{(R')^2}{R^2} \,, \tag{5.65}$$

where $f_0$ is a constant flux parameter, as defined in (3.10). For the purposes of this section we work in Planck units, by setting $\ell_{11} = 1$, and from now on we will also set $|\rho_c| = 1$. To remind ourselves of these units, we will henceforth put hats on all the physical quantities.

The rescaling symmetry (5.43) of the local fiducial geometry is not broken in the nonlinear equations. In the gauge (5.61) it acts as

$$\hat{y} \to e^c \hat{y} \,, \qquad \hat{R}_c \to e^{\frac{2}{11}c}\hat{R}_c \,, \qquad \hat{f}_0 \to e^{-c}\hat{f}_0 \,, \qquad \hat{C} \to e^{-2c}\hat{C} \,, \tag{5.66}$$

for a real constant $c$.

We are seeking solutions to the equations of motion where $\hat{R}$ shrinks so as to cap off the geometry smoothly. To construct this class of solutions we first analyze the equations locally near this smooth point, which we take to be at $\hat{y} = 0$, and we will then evolve them numerically. Smoothness is imposed by requiring that $\hat{R}$ goes to zero linearly and that the first derivatives of all the other functions vanish. These conditions result in the following power expansion of

the metric functions:

$$e^{2A} = e^{2a_0} + \hat{y}^2 \left( -\frac{e^{2a_0}}{4r_{c0}^{11}} + \frac{1}{6}e^{-6a_0}\hat{f}_0^2 + \frac{\hat{C}}{8} \right) + \tag{5.67a}$$

$$+ \hat{y}^4 \left( -\frac{11e^{-6a_0}\hat{f}_0^2}{384r_{c0}^{11}} + \frac{67e^{2a_0}}{480r_{c0}^{22}} - \frac{1}{6}e^{-8a_0}\hat{f}_0^2\frac{1}{12}\hat{C} - \frac{7}{432}e^{-14a_0}\hat{f}_0^4 - \frac{3}{16}e^{-2a_0}\frac{1}{144}\hat{C}^2 - \frac{1}{64r_{c0}^{11}} \right)$$

$$+ O(\hat{y}^5),$$

$$\hat{R}_c^2 = r_{c0}^2 + \hat{y}^2 \left( \frac{3}{10r_{c0}^9} - \frac{1}{12}e^{-8a_0}\hat{f}_0^2 r_{c0}^2 \right) + \tag{5.67b}$$

$$+ \hat{y}^4 \left( -\frac{e^{-8a_0}(54e^{6a_0}\frac{1}{12}\hat{C} + 5\hat{f}_0^2)}{360r_{c0}^9} + \frac{1}{54}e^{-16a_0}\hat{f}_0^2 r_{c0}^2(e^{6a_0}\frac{3}{4}\hat{C} + \hat{f}_0^2) - \frac{17}{200r_{c0}^{20}} \right) + O(\hat{y}^5),$$

$$\hat{R}^2 = \hat{y}^2 + \hat{y}^4 \left( -\frac{5}{36}e^{-8a_0}\hat{f}_0^2 - \frac{1}{6}e^{-2a_0}\hat{C} - \frac{1}{3r_{c0}^{11}} \right) + O(\hat{y}^6). \tag{5.67c}$$

We have displayed in (5.67) only the first few perturbative orders, but the expansion can be analytically computed to arbitrarily high order. Since a constant shift of $A$ is unphysical, we set $a_0$ to 1. Also, we can trade $\hat{C}$ for $c$ by using (5.66). Thus, at a fixed $c$, the expansion only depends on the flux parameters $\hat{f}_0$ and on $r_{c0}$, the size of the Casimir circle at the end of the cusp.

We can now ask how these smooth local solutions extend away from the end of the cusp. A common strategy is to evaluate them at a very small $\hat{y}$, where the expansion is reliable, in order to obtain the initial conditions for starting a radial numerical evolution. By scanning over the parameters, we have obtained that if $r_{c0}$ is too big the Casimir contribution is always subdominant. However, for $r_{c0}$ below a certain threshold the integrated Casimir energy competes with the other sources, producing solutions such as the one in Figure 6.

On the right, this solution is glued to the core of the manifold, where the radial ansatz breaks down and the transverse gradients become important. More precisely, $u = e^{2A}$ reaches its minimum, and after this point a region starts where $(A')^2 \ll \nabla^2 A$. As we expect from general results, the integrated Casimir energy is competitive with the other sources also in the backreacted geometry. Indeed, evaluating the ratio (5.44) for the solution in Figure 6, we obtain

$$\frac{\int \sqrt{g^{(7)}}u^2 \left[ -R^{(7)} - 3\left(\frac{\nabla u}{u}\right)^2 + \frac{1}{2}|F_7|^2 \right]}{-\int \sqrt{g^{(7)}}u^2 \ell_{11}^9 R_c^{-11}} \sim -1.06. \tag{5.68}$$

In this solution, we can also check that $\int \sqrt{g^{(7)}}u^2 \left[ -R^{(7)} - 3\left(\frac{\nabla u}{u}\right)^2 \right] > 0$. Let us normalize it by comparing with the flux, obtaining for the solution in Figure 6 the ratio

$$\frac{\int \sqrt{g^{(7)}}u^2 \left[ -R^{(7)} - 3\left(\frac{\nabla u}{u}\right)^2 \right]}{\int \sqrt{g^{(7)}}u^2 |F_7|^2} \sim .25. \tag{5.69}$$

Notice that the ratios (5.68) and (5.69) are unaffected by the rescaling (5.66), meaning that they have the same values in the rescaled solutions where $\hat{R}_c \gg 1$.

## 5.4 The Hessian in our problem

Let us now address second order deformations away from our background backreacted configuration. We will study this in two ways, first a general method and then a test of the result obtained by perturbing the explicit patchwise solution in §5.3.

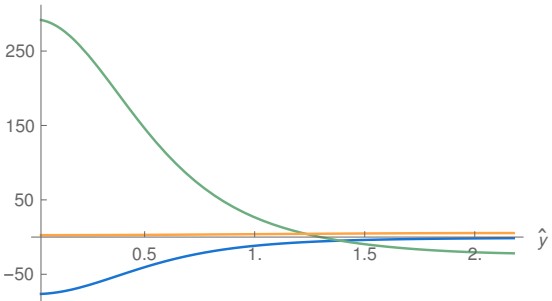
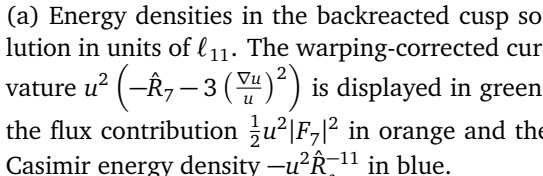
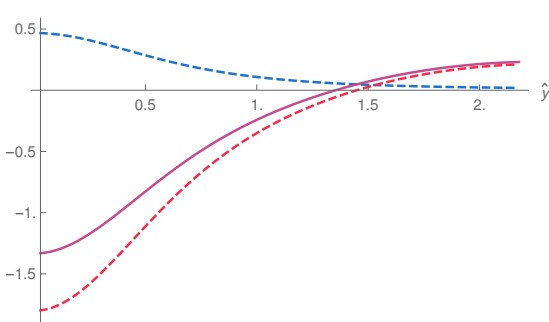

(a) Energy densities in the backreacted cusp solution in units of $\ell_{11}$. The warping-corrected curvature $u^2 \left( -\hat{R}_7 - 3 \left( \frac{\nabla u}{u} \right)^2 \right)$ is displayed in green; the flux contribution $\frac{1}{2} u^2 |F_7|^2$ in orange and the Casimir energy density $-u^2 \hat{R}_c^{-11}$ in blue.

(b) Different contributions to the on-shell Schrödinger potential (5.38) in units of $\ell_{11}$: curvature+flux(red), Casimir (blue), and their sum (purple). Notice that when the flux is on-shell the equation satisfied by $u$ becomes non-linear, but the non-linearity is mild enough that the intuition from the Schrödinger problem still applies.

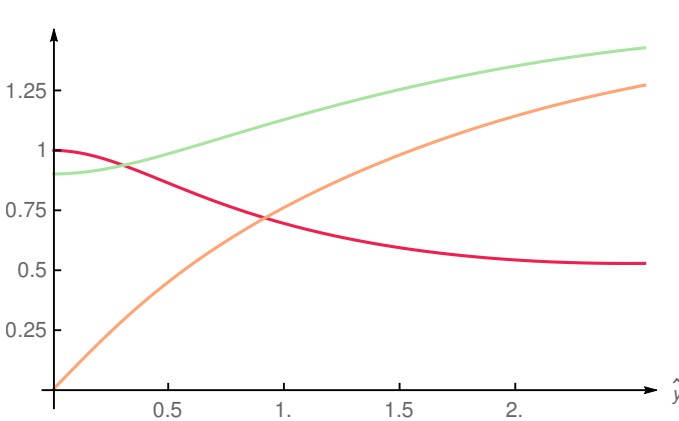

Figure 6: A numerical solution of the equations of motion in the cusp region. At $\hat{y} = 0$, the function $\hat{R}$ (orange) goes to zero linearly, with the circle parametrized by the coordinate $\theta$ shrinking smoothly. At the same point, both $e^{2A}$(red, rescaled with $e^{-2a_0}$) and $\hat{R}_c$(green), reach a finite value, in agreement with the expansion (5.67). On the right, the solution is glued to the bulk of the manifold, as discussed in the main text. Panel (a) and (b) show the backreacted energy densities and contributions to the $u$ equation of motion respectively. This numerical solution is obtained for $c = 2.2$, $r_{c0} = e^{\frac{2}{11}c} 0.65$, $\hat{f}_0 = e^{-c} 150$. For clarity we are displaying it for a small $c$, but increasing $c$ acts as a simple rescaling as in (5.66). This can be used to make the manifold big, ensuring at the same time $\hat{R}_c \gg 1$ and quantizing the integral of $F_7$.

Here we will make use of the analogue Schrödinger problem [17] structure of the constraint equation (3.7), via trial wavefunctions. Pursuing this method, developed earlier in §3.1, we start by noting that a trial wavefunction $u_t$ will yield an approximation to the ground state energy that is no better than the true ground state:

$$\langle u_t | (-\hat{H}) | u_t \rangle \leq \langle u_0 |_c | (-\hat{H}) | u_0 |_c \rangle \simeq 2\ell_{11}^9 \ell^2 V_{eff} \,, \tag{5.70}$$

where we now use an inner product

$$\langle f_1 | f_2 \rangle = \int \sqrt{g^{(7)}} f_1 f_2 \,. \tag{5.71}$$

In writing the last relation of (5.70) we work, as above, in the regime where $C\ell^2 \ll 1$ so that the the warp factor $u$ is well approximated by the ground state wavefunction.

We start in a solution with $V_{eff} G_N^2 \gtrsim 0$ and we would like to incorporate second order variations away from this solution, determining if these keep $V_{eff}$ positive or if they lead to tachyons sending the system to negative $V_{eff}$.

Let us write the internal metric in the form $g^{(7)} = e^{2B} G$, yielding curvature $e^{-2B}(R_G^{(7)} - 12\nabla_G^2 B - 30(\nabla_G B)^2)$. $V_{eff}$ becomes

$$\begin{aligned} V_{eff} = &-\frac{1}{2\ell_{11}^9} \int d^7 y \sqrt{G} e^{5B} e^{4A}|_c \left( R_G^{(7)} + \frac{1}{4}\ell_{11}^9 e^{2B} T^{(\text{Cas})\mu}{}_\mu - \frac{1}{2} e^{2B}|F_7|^2 \right. \\ &\left. +12(\nabla A)^2\Big|_c - 12\nabla^2 B - 30(\nabla B)^2 \right) + \frac{C}{2}\left( \frac{1}{G_N} - \frac{1}{\ell_{11}^9} \int \sqrt{g^{(7)}} u|_c \right). \end{aligned} \tag{5.72}$$

With an integration by parts on the $\nabla^2 B$, this becomes

$$\begin{aligned} V_{eff} = &-\frac{1}{2\ell_{11}^9} \int d^7 y \sqrt{G} e^{5B} e^{4A}|_c \left( R_G^{(7)} + \frac{1}{4}\ell_{11}^9 e^{2B} T^{(\text{Cas})\mu}{}_\mu - \frac{1}{2} e^{2B}|F_7|^2 \right. \\ &\left. +12(\nabla A)^2\Big|_c + 30(\nabla B)^2 + 48\nabla B\nabla A|_c \right) + \frac{C}{2}\left( \frac{1}{G_N} - \frac{1}{\ell_{11}^9} \int \sqrt{g^{(7)}} u|_c \right) \end{aligned} \tag{5.73}$$

as in [17] equation (4.2).

Let us begin by considering a trial wavefunction $u_{t0} = e^{2A_{t0}}$ given by

$$A_{t0} = -2B \,, \tag{5.74}$$

up to a constant addition where $B$ includes the background solution for the conformal factor along with small deformations $\delta B$ away from it which enter into the Hessian. Plugging this into the formula (5.73) for $V_{eff}$ yields

$$\langle u_{t0} | (-\hat{H}) | u_{t0} \rangle = \ell^2 \int \sqrt{G} e^{-3B} \left( 18(\nabla B)^2 - R_G^{(7)} + e^{2B}(\ell_{11}^9 \rho_C + \frac{1}{2}F_7^2) \right). \tag{5.75}$$

The integrand in this expression is positive away from the Casimir source. Firstly, the gradient squared term is net positive; this comes about because the cross term proportional to $\nabla A \nabla B$ in (5.73) is positive and overcompensates the negative $(\nabla B)^2$ and $(\nabla A)^2$ terms. This is very similar to the effect obtained in [17] in studying the solution to the constraint equation for short wavelength modes of $A$ and $B$, for which the constraint boils down to $\nabla^2 A = -2\nabla^2 B$ and leads to a positive Hessian despite the naïve catastrophic instability from the conformal factor gradients. Here in (5.74) we are taking a similar relationship, working at the level of a trial wavefunction rather than a solution to the constraint equation (the analogue Schrödinger equation). Secondly, in our problem $-R_G^{(7)} > 0$, with the metric $G$ describing a hyperbolic space possibly deformed in directions $h$ orthogonal to the conformal mode; the varying conformal factor in (5.72) leads to net positive curvature required (4.12) in bulk regions as described above in §5.2. These deformations $h$ are volume-preserving according to a standard decomposition [16, 59]; see also the more recent [60]. These directions are positive up through second order [16], reflecting the rigidity of hyperbolic space. This follows from Theorem 4.60 and Corollary 12.73 in [16]. Moreover, as we deform in the volume-preserving directions $h$,

we can solve the $C_6$ equation of motion as in (3.10), and there is no dependence of the $\int F_7^2$ term in $V_{eff}$ on $h$.

But this trial wavefunction $u_{t0}$ is not yet appropriate for our purposes: the integrand of (5.75) develops a large negative contribution there as $B$ becomes smaller in the Casimir region since the Casimir energy pushes down on the volume. This is reflected in our dressed background solution sketched in figure 2 and illustrated above in section 5.3. Let us instead consider an improved trial wavefunction $u_t$ which only takes the form (5.74) outside the region where Casimir becomes significant – in our setup this will be near small circles in the geometry. Before getting to the Casimir region, we may join (5.74) to a wavefunction that decays exponentially (starting where the integrand is still positive). We would first like to understand if we can construct a trial wavefunction in this (or another) way which gives a positive result overall for $\langle u_t|(-\hat{H})|u_t\rangle$, in which case the true wavefunction will give a larger positive value than this for $V_{eff}$.

Let us make some estimates of this. We can use coordinates for the cusp region

$$ds^2 = dy^2 + e^{2B}ds_\perp^2, \tag{5.76}$$

where the end of the filled cusp is at $y = y_{DF}$ and $ds_\perp^2$ contains the proper cross sectional $T^6$. We work with the trial wavefunction (5.74) in the central manifold and for $y < y_*$. Those regions give a positive contribution to $V_{eff} = \langle u|-\hat{H}|u\rangle$ as just explained, from the expression (5.75) with negligible contribution from the only negative term, the Casimir energy. For $y > y_*$ we join this to a function approximating

$$A_t = A_{t*}e^{(y-y_*)/\ell} = -2B(y_*)e^{(y-y_*)/\ell}, \tag{5.77}$$

with $B(y_*) > 0$. The contribution to $\langle u_t|-\hat{H}|u_t\rangle$ from this region is given by the earlier expression (5.73), but with the constraint solution $A|_c$ replaced by the trial function $A_t$. The function (5.77) plugged into (5.73) yields negative contributions from the gradients, in contrast to the net positive contribution in the other region (5.75). But this is multiplied by a rapidly dropping function $u_t^2 = e^{4A_t}$ (5.77). One contribution is from the $-12(\nabla A)^2$ term; taking the gradient of (5.77) we find that this goes like the transverse proper hyperbolic torus volume at the transition point $\sim e^{6y_*}$ times

$$-12 \times 4B_*^2 \int_{y_*}^\infty dy\, e^{-6(y-y_*)} \exp(-8B_*e^{(y-y_*)})e^{2(y-y_*)}e^{5B(y)} = -48\ell B_*^2 \int_1^\infty dY\, e^{-8|B_*|Y}Y^{-5}e^{5B(y)}, \tag{5.78}$$

with $B_* = B(y_*)$ and $0 < B_* \lesssim 1$, with $Y = e^{(y-y_*)/\ell}$.

An upper bound on the magnitude of this negative contribution is given by evaluating the $e^{5B}$ factor at $B = B_*$, since $B$ decreases toward the end of the cusp in our solution. Evaluating this gives

$$-48\ell B_*^2 e^{5B_*} \int_1^\infty dY\, e^{-8B_*Y}Y^{-5} \tag{5.79}$$

times the transverse volume at $y_*$. A similar estimate can be made for the other gradient terms in (5.73); for them the analogue of (5.79) is

$$-30\ell e^{5B_*} \int_1^\infty dY\, e^{-8B_*Y}Y^{-7},$$

for the $(\nabla B)^2$ term and

$$-96B_* e^{5B_*} \int_1^\infty dY\, e^{-8B_*Y}Y^{-6},$$

for the $\nabla A \cdot \nabla B$ term.

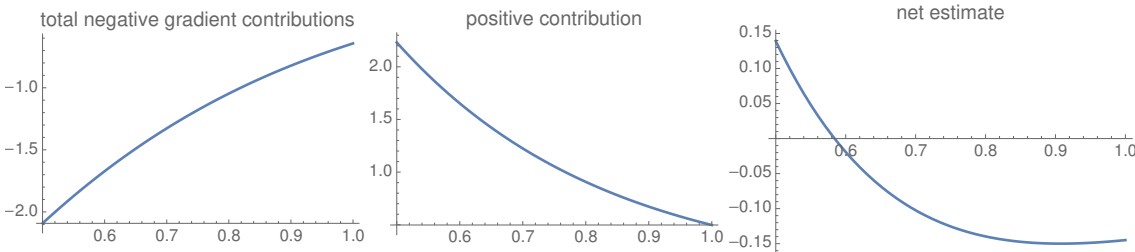

Figure 7: The relative sizes of the positive, negative, and net contributions to $\langle u_t|(-\hat{H})|u_t\rangle$ as a function of $B_*$, as roughly estimated in the text.

We would like to compare these to the positive contribution from the bulk region. A conservative estimate for this is given by $\sim \mathcal{O}(10)e^{-3B_*}$ times the bulk volume (which is larger than the volume over which the negative contribution (5.79) has support).

These crude estimates for the total negative contribution from gradients and the positive bulk contribution to $\langle u_t|(-\hat{H})|u_t\rangle$ roughly cancel each other, as depicted in figure 7. This in itself is a check of the arguments we made for the tunability of our $a$ parameter and the cosmological constant – if a trial wavefunction gave a large result it would require a correspondingly large cosmological constant (5.70). Moreover, with a conservative estimate for the positive term of $10e^{-3B_*}$ this enables a net positive result for $\langle u_t|(-\hat{H})|u_t\rangle$ which is smaller than the individual terms. For a larger coefficient (larger volume of the positive piece), the range of $B_*$ with a positive net value would be greater. The volume of the positive and negative contributions depend on our choice of $y_*$, which is limited by the volume of bulk and Casimir regions of the underlying solution. In particular, the bulk volume is limited in our $a \ll 1$ tuned models, a feature that fits with the substantial cusp volume in the class of hyperbolic manifolds (such as [25] that we are considering).[13]

If the net result for an appropriate choice of trial wavefunction is indeed positive, as suggested by this crude estimate (and the positive contributions from curvature and warping §3.1), it yields a more general result about the Hessian. We can include in $\langle u|(-\hat{H})|u\rangle$ deformations away from the solution which are small compared to a typical term in this integral but large compared to $\lambda_0$. Since we see here that the full perturbed Hamiltonian remains positive, we can conclude that such deformations increase $V_{eff}$.[14]

### 5.4.1 Perturbations of the backreacted solution from §5.3

We can study the Hessian as well as nonlinear deformations of $V_{eff}$ by perturbing the explicit radial solution in section 5.3. To do this, we must define deformations of the fields $\sigma_c = \delta \log(R_c)$ and $\sigma = \delta \log(R)$ which are non-singular, turn them on and solve the constraint equation (3.7) in order to construct the off-shell potential $V_{eff}$ as a functional of the original fields and the deformations.

The solution in §5.3 extends over a patch (5.61) with a finite range of $y$. Studying this patch in depth was motivated by the role of the cusps in our geometry supporting Casimir energy. We must include appropriate boundary terms in $V_{eff}$ for the radial patch in order to obtain a good variational problem. However, as we stressed above the boundary of the patch in §5.3 is not aligned with the polygon boundaries relevant for constructing hyperbolic manifolds

---

[13]We stress that, within these limitations, we should choose the variational parameter $y_*$ so that $\langle u_t|(-\hat{H})|u_t\rangle$ is maximized, as this will produce the strongest bound on the exact wavefunction.

[14]We note that the unstable direction found in a subset of previous perturbative de Sitter models followed a pattern [61, 62] involving an N=1 supersymmetry breaking goldstino and other features not present here. A tachyon was also found in [63], but not in [8, 13, 45] (although the last reference was less explicit).

via a gluing as described in [15] and appendix A, where a significant role is played by transverse gradients going beyond the radial ODEs solved in §5.3. For this reason, let us now focus on deformations that are supported away from the boundaries. We note as well that the flux is a subdominant contribution in the end region of the backreacted radial solution. This also simplifies our analysis of the behavior of the off-shell potential $V_{eff}$ under the deformation.

Let us define deformed metric components (5.61)

$$R[\sigma] = R_{sol}(1 + \epsilon\sigma(\hat{y})), \quad R_c[\sigma_c] = R_{csol}(1 + \epsilon\sigma_c(\hat{y})), \tag{5.80}$$

with the subscript $sol$ denoting the background solution. The deformations $\sigma, \sigma_c$ must be chosen in such a way that they are well-defined in the geometry. In the radial setup of §5.3, we impose boundary conditions to ensure a smooth filled end at $\hat{y} = 0$ in (5.61) (so that the geometry behaves like $T^5$ times the origin of polar coordinates in the $y, \theta$ directions). This requires $R'(0) = 1, R(0) = 0, R_c'(0) = 0$.

An example of a deformation satisfying these conditions is

$$\begin{aligned}
\sigma &= 25\sin(\hat{y})^2(\tanh(\frac{\hat{y} - \hat{y}_f}{\Delta\hat{y}}) - 1), \\
\sigma_c &= \frac{\cos(\frac{\hat{y}\pi}{2\hat{y}_{max}})}{\cosh(\frac{\hat{y}}{\Delta\hat{y}})},
\end{aligned} \tag{5.81}$$

where $\hat{y}_f$ is the point where the flux begins to grow; it is highly subdominant in the filling region near $\hat{y} = 0$. Here $\hat{y}_{max}$ is the largest value of $\hat{y}$ in the radial solution, and $\Delta\hat{y} \ll \hat{y}_{max}$ is much less than the range of the solution. For this type of deformation, an explicit calculation of $V_{eff}$ – including the calculation of the warp factor solving the constraint equation (3.7) for the full deformed metric (5.80) – yields a positive Hessian, with $V_{eff}$ increasing for both signs of the deformation parameter $\epsilon \ll 1$. Similar results hold for a deformation of this kind in the direction of $\sigma_c$ alone.

This provides a consistency check of the general arguments above. In so doing, this directly addresses the effect of a localized $\sigma_c$ deformation which might a priori appear dangerous since it can increase the magnitude of the negative Casimir energy.

### 5.4.2 Hessian summary

Let us now summarize the status of the Hessian calculations. First, we have a set of general statements:

• The warping model-independently contributes positively in itself as in §3.1.

• The underlying negative curvature contributes positively in itself as in [16].

• The type of tachyon identified in a subset of previous power-law stabilized de Sitter models [61] does not arise here (there is no limit with low energy supersymmetry)

• In general, there is more phase space at higher energies and most contributions to $V_{eff}$ are positive.

These generalities all go in the direction of positive Hessian in this setup. But because we have variations of the warp and conformal factors, we need a more detailed analysis to calculate the Hessian in our models. In that direction:

• The trial wavefunction explained at the beginning of this section, when estimated according to the structure of our solutions, suggests a small positive lower bound on $V_{eff}$ near the solution. These estimates are crude but conservative in the sense described above.

• In subsection 5.4.1 we explicitly computed the effect on $V_{eff}$ of deforming the metric components toward the end of the filled cusp solution of §5.3 as described, finding a positive Hessian in these directions. A calculation of the Hessian for more general deformations in particular examples would be interesting to pursue in more detail.

## 5.5 Bulk tadpole estimate

In our treatment of the inhomogeneity thus far in this section, we found that both warp and conformal factors get sourced, in a way that is consistent with the sign condition (4.12). Moreover, we displayed explicit solutions for the backreacted cusp including additional field variations, including the end region where Casimir energy is supported.

It is convenient to work with a configuration where most of the fields $h$ in (1.1) are not turned on, and estimate how much their tadpoles, combined with the gapped Hessian, will shift them. We undertake that in this section. We argued above in §4 and §5.1 that deformations in the end regions of the cusp are bounded, leaving us with a localized negative energy source. What remains is to estimate tadpoles and shifts for bulk deformations, taking into account the tadpoles in the bulk region and the scale of the gap in the Hessian.

The analysis in the previous subsections already implemented some of the required shifts. For example, §5.2.1 described the internal spatial variation of the warp and conformal factors that are sourced by the localized Casimir stress energy, and §5.3 gave some explicit radial solutions for all fields in the cusp. Any tadpoles for the deformations $h$ in (1.1) arise from the inhomogeneities. In the bulk regions this is limited by the fact that the fiducial curvature and flux terms in $V_{eff}$ are both homogeneous. There will, however, be contributions to $V_{eff}$ from the $\nabla^2 v$ contribution to the full 7-dimensional curvature $R^{(7)}$, treated in §5.2.1. This descends from the conformally rescaled internal curvature, and may lead to tadpoles in $h$.

In general, in order to present a simple analysis of (5.1)-(5.2) in our system, in this section we will start from a conservative estimate for the size of the tadpoles rather than invoking the details of solutions studied in the previous sections. As we will see shortly, this is sufficient for our purposes. We expand around our symmetric metric, $g_{\mathbb{H}n}{}^m \to g_{\mathbb{H}n}{}^m + \tilde{h}_n{}^m$, and we want to understand if the shifts $\tilde{h}_n{}^m$ are small. Here $\tilde{h}_n{}^m$ represents any metric deformation (using the tilde here to distinguish these perturbations from those in decomposition in (1.1)).

Starting from the hyperbolic metric, the tadpoles originate from the inhomogeneity of the Casimir stress energy. They are of the schematic form

$$\frac{\delta V_{eff}}{\delta \tilde{h}_n^m(y)} \sim t_{C}{}_m^n(y), \tag{5.82}$$

where $t_{C}{}_m^n$ is the difference in the Casimir stress-energy between its average and the pointwise values in (4.2).

This is approximately diagonal in the fiducial metric, with entries of order $|\bar{\rho}_C| \sim 1/R_c^{11}$ where $R_c$ is the minimal size of the circle generating the leading contribution to the Casimir energy. Expanding to quadratic order,

$$V_{eff} = V_{eff,\mathbb{H}} + \frac{1}{\ell_{11}^9} \int d^7 y \sqrt{g_{\mathbb{H}}} \left\{ \frac{1}{2} \tilde{h}_n^m \Delta_{(Total)}{}_{mp}^{nq} \tilde{h}_q^p - \ell_{11}^9 \tilde{h}_m^n t_{C}{}_n^m \right\} + \mathcal{O}(\tilde{h}^3). \tag{5.83}$$

The differential operator $\Delta_{(Total)}$ mapping symmetric tensors to symmetric tensors has a gapped spectrum of order $1/\ell^2$, given that we solve for the warp factor and $C_6$, as described above in §5.4.

From this we can express the shift $h$ in the metric as

$$\tilde{h} \sim \ell_{11}^9 \Delta_{(Total)}^{-1} t_C, \tag{5.84}$$

and we want to know whether or not $\tilde{h}$ is a small deformation. To be more specific, we may expand $t_C$ in a basis of eigentensors of $\Delta_{(Total)}$, writing

$$t_C = \sum_I \tau_I \varphi_I, \quad \Delta_{(Total)} \varphi_I = \lambda_I \varphi_I, \tag{5.85}$$

with $\varphi_I$ orthogonal to pure gauge modes in the inner product $\langle \varphi, \tilde{\varphi} \rangle = \int \sqrt{g_{\mathbb{H}}} \varphi_n^m \tilde{\varphi}_m^n$. It is natural to separate the deformations $\tilde{h}$ into $\delta B$ and $h$ (1.1), as discussed above. But since for both the eigenvalues of $\Delta_{(Total)}$ are gapped at order $1/\ell^2$, in this section we combine them for simplicity. In the bulk of the internal space, $t_C$ is slowly varying, so we will only have low-lying modes contributing to the sum (5.85).

Let us denote by $\lambda_0$ the minimal eigenvalue of $\Delta_{(Total)}$. This will generate the largest shift in $\tilde{h}$ (5.84). We have $\tau_0 \sim 1/R_c^{11}$, giving

$$\tilde{h} \sim \frac{\ell_{11}^9 \tau_0}{\lambda_0} \varphi_0 \sim \frac{\ell_{11}^9 \ell^2}{R_c^{11}} \varphi_0 \,. \tag{5.86}$$

In terms of $\epsilon$ defined above (4.11), we now apply the stabilization mechanism (4.9) to eliminate $\ell^2 = \hat{\ell}^2 \ell_{11}^2$, giving

$$\hat{\ell}^2 \sim \frac{1}{\epsilon^{1/2} \hat{R}_c^{5/2}} \,, \tag{5.87}$$

where $\hat{R}_c = R_c/\ell_{11}$. From that relation we also demand the inequality

$$\hat{\ell}^4 \gg 1 \Rightarrow \frac{1}{\hat{R}_c} \gg \epsilon^{1/5} \tag{5.88}$$

in terms of the quantities defined above in §4. As explained there, this inequality (5.88) is consistent with $\hat{R}_c \gg 1$ given $\epsilon \ll 1$. Putting together (5.87) and (5.86), we find $\tilde{h} \ll \varphi_0$ requires

$$\frac{1}{\hat{R}_c^{11+5/2} \epsilon^{1/2}} \ll 1 \,, \tag{5.89}$$

and for this to be consistent with (5.88) requires a window

$$\epsilon^{27/10} \ll \frac{1}{\hat{R}_c^{27/2}} \ll \epsilon^{1/2} \,. \tag{5.90}$$

This is available as long as $\epsilon^{11/5} \ll 1$ which is satisfied in our setup via the tuning $a \ll 1$ (4.6).

In summary, the tadpoles in the bulk of the space induced by the Casimir inhomogeneity lead to small shifts $\tilde{h} \ll \varphi_0$ away from the fiducial metric. This indicates that we can self-consistently neglect the higher order terms in (5.83). Some of the details of these shifts were laid out in the previous subsections. Altogether, these parametric estimates indicate that the de Sitter minimum suggested by the original 3-term structure explained above in §2 and §4 survives the effects of the inhomogeneity of the Casimir stress energy.

# 6 No go theorem for AdS

In the previous sections we have described a mechanism to stabilize four-dimensional compactifications with negative internal curvature, showing that with appropriate choices of the internal manifold the automatically generated Casimir energy provides enough negative energy to yield metastable de Sitter compactifications.

The simplicity of this set of ingredients is quite consistent with well-known no-go theorems [64–66]. Albeit quite powerful to constrain compactifications at the level of classical general relativity (without general orientifold planes or quantum effects), these no-go theorems only focus on a very limited set of classical contributions to the stress-energy tensors obeying standard energy conditions, as we review in detail in Appendix D. It was understood

soon after the discovery of the cosmological constant [67, 68] that negative sources such as orientifold planes, intermediate in the expansion about weak coupling and large radius are required for de Sitter [44, 45]. Orientifold planes are classical in string theory but go beyond general relativity.

It is worth stressing that classically, there is a no go theorem for metastable atoms. Quantum mechanics is essential to the stability of matter, and also enters into the dynamics of stars stabilized by degeneracy pressure. Quantum effects naturally also enter into compactification physics. Incorporating such effects to understand quantum gravity backgrounds forbidden in classical general relativity is not new; non-perturbative effects feature in the KKLT construction [55] and generalizations such as [69]. Casimir energy in particular has been used in [39, 70].

In this section, we revisit these constraints for our present class of four-dimensional M-theory compactifications, and generalize them to include Casimir energy. This yields a reversal of the classical no go theorems, enabling a simple no go for AdS solutions in a particular regime. We summarize the main point here and refer to Appendix D for more general results.

The main ingredient of our analysis is the integrated combination of stress-energy tensors derived in (D.13), which we specialize here for $D = 11, d = 4$:

$$I_{\text{tot}} \equiv \frac{1}{2} \int \sqrt{g^{(7)}} e^{2A} \left( \frac{5}{9} T^{(4)} - \frac{4}{9} e^{2A} T^{(7)} \right), \tag{6.1}$$

where $T^{(4)}$ and $T^{(7)}$ are respectively the four- and seven- dimensional traces of the total eleven-dimensional stress energy tensor, with the warping factors stripped off.

Assuming the internal space is smooth and without boundaries, which in our case is true for the filled-cusps, a combination of the eleven-dimensional Einstein equations integrated over the internal space (D.12) can be rewritten as

$$\frac{1}{G_N} R^{(4)} = I_{\text{classical}} + I_{\text{quantum}}, \tag{6.2}$$

where we have split (6.1) in its classical and quantum contributions $I_{\text{tot}} \equiv I_{\text{classical}} + I_{\text{quantum}}$.

With the help of (B.3) (B.4) (B.5) we can explicitate the various pieces in (6.1) for our present case:

$$I_{\text{Cas}} = -2 \int \sqrt{g^{(7)}} e^{4A} \rho_C(R), \qquad I_{F_7} = -\frac{4}{3\ell_{11}^9} \int \sqrt{g^{(7)}} e^{4A} |F_7|^2, \tag{6.3}$$

so that we obtain

$$\frac{1}{G_N} R^{(4)} = -\frac{4}{3\ell_{11}^9} \int \sqrt{g^{(7)}} e^{4A} |F_7|^2 + 2 \int \sqrt{g^{(7)}} e^{4A} |\rho_C(R)|, \tag{6.4}$$

where we have used that the Casimir energy density is negative in our setup. If the Casimir contribution dominates the one from the flux a dS compactification is allowed, whereas it would not be in the classical supergravity background without this term. Turning it on its head, we can also conclude that

$$\text{No AdS if:} \qquad \int \sqrt{g^{(7)}} e^{4A} |F_7|^2 \leqslant \frac{3}{2} \ell_{11}^9 \int \sqrt{g^{(7)}} e^{4A} |\rho_C(R)|. \tag{6.5}$$

In our de Sitter construction, these integrated quantities are of the same order of magnitude and satisfy this inequality. They entered above in the effective potential (3.9) and the hierarchies (5.21). The no go theorem (6.5) says that in the regime indicated, there are no AdS

extrema. As a special example of this, if we turn off the $F_7$ flux entirely, the theorem guarantees that AdS will not appear anywhere in the rich landscape descending from the quantum-corrected compactification containing the Casimir energy. de Sitter extrema are allowed. Even without flux, at least at the level of averaged sources one finds a dS saddle point at the the local maximum in figure 3, obtained by balancing the curvature and Casimir terms in the potential for the overall volume. Our models in this paper do contain $F_7$ flux, essential for the metastable dS minimum in the potential for $\ell$ as in figure 3. The result (6.5) shows that in the more general landscape with or without flux, regardless of the internal geometry, a sufficiently strong Casimir energy precludes AdS but allows dS. We should note that the right hand side of (6.5) is limited by the screening effect of the warp factor $u = e^{2A}$ as discussed in §4 and §5.1.

# 7 Axion and inflationary dynamics

Let us now generalize our construction to incorporate the dynamics of axions descending from the potential field $C_3$. Being generic bosonic fields in string/M theory, axions play a leading role in top down models of early and late universe phenomena [71], and they and their sources are crucial to quantitative entropy counts and landscape statistics.

Starting from M theory on a 7-manifold $M_7$, the 3-form potential $C_3$ of eleven dimensional supergravity descends to axion fields in the four dimensional theory,

$$c^I = \int_{\Sigma_I^{(3)}} \frac{C_3}{\ell_{11}^3}, \quad I = 1, \ldots, b_3, \tag{7.1}$$

where $\Sigma_I^{(3)}$ is an element of the homology group $H_3(M_7)$ (given a filling [19] of all the cusps to make a compact manifold – otherwise we would consider also $H_3(M_7, \partial M_7)$).

We note that in this system, as is true generically in string/M theory, the axions do not come with scalar saxion particles as would happen in the special case of low energy supersymmetry. Indeed, axions dominate the spectrum of relatively light fields by a considerable margin – their number growing with dimensionality in the general case like $2^D$ as well as proliferating with topology, while metric deformations, with a species number $\sim D^2$, generically obtain masses of order $1/\ell$ from the rigidity properties of negatively curved Einstein manifolds [16] along with warping effects [17] as we have seen in detail above.

As discussed above, the flux and curvature contributions are supported over the bulk of the internal space (rather than the regions near the small Casimir circle). In the bulk of the internal space, where the warp factor variation is negligible, the axion kinetic terms are of the form

$$\int \sqrt{-g^{(11)}} \frac{F_4^2}{\ell_{11}^9} \sim \int d^4x \sqrt{-g^{(4)}} \frac{\ell^7}{\ell_{11}^9} \dot{c}^2 \frac{1}{\hat{\ell}^6} = \int d^4x \sqrt{-g^{(4)}} f^2 \dot{c}^2 \equiv \int d^4x \sqrt{-g^{(4)}} \dot{\phi}_c^2, \tag{7.2}$$

with $\phi_c = f c$ the corresponding canonically normalized field with axion decay constant

$$f \sim \frac{M_p}{\hat{\ell}^3}, \tag{7.3}$$

(where as above we denote $\hat{\ell} = \ell/\ell_{11}$). For simplicity here we are taking all scales of order $\ell$; more generally one obtains similar results from a more precise calculation of the overlap of differential forms $\int \omega \wedge \star \omega$ that arises from plugging $C_3 = \sum_I c_I \omega_I$ into the $F_4^2$ term, with $\omega_I$ denoting an integral basis of 3-cohomology elements [1].

In the absence of 4-form flux, this would introduce $b_3$ traditional axion fields, with a periodic non-perturbative potential generated by Euclidean M2-branes. This setup may apply to N-flation [72] and other phenomenological scenarios involving axions [73, 74].

Generically, with both types of magnetic fluxes incorporated, there is a multifield axion monodromy potential in this system, introducing somewhat larger masses (still sub-KK scale), with a branched potential containing a large field range on each branch suitable for inflation. The generalized magnetic flux

$$\tilde{F}_7 = F_7 + C_3 \wedge F_4 \tag{7.4}$$

contributes an axion potential to our four-dimensional Einstein frame potential $V_{eff}$ of the form (cf (3.9))

$$\tilde{V}_7 = \frac{\ell_{11}^9}{2G_N^2} \frac{\int d^7y \sqrt{g^{(7)}} u^2|_c \left(\frac{1}{2}|\tilde{F}_7|^2\right)}{(\int d^7y \sqrt{g^{(7)}} u|_c)^2} \sim M_P^4 \frac{(N_7 + cN_4)^2}{\hat{\ell}^{21}}. \tag{7.5}$$

For an elegant derivation of this starting from the electric 4-form description, see [29]. The magnetic 4-form flux adds a contribution to the potential of the form

$$V_4 \sim M_P^4 \frac{N_4^2}{\hat{\ell}^{15}}. \tag{7.6}$$

We will work in a regime such that this is subdominant to the de Sitter potential. One way to express this is that its energy density in eleven dimensions will be much smaller than the existing terms in the potential, including the curvature. That is, it will satisfy

$$\frac{N_4^2}{\hat{\ell}^8} \ll \frac{1}{\hat{\ell}^2}. \tag{7.7}$$

In general, such 4-form flux introduces new asymmetric forces into the system, which could distinguish the overall size $\ell$ from the size $\ell_4$ of the four-cycle(s) threaded by $F_4$. But this condition (7.7) implies that the stabilizing effects of the Hessian still maintain the size $\sim \ell$ of the four-cycle threaded by $F_4$ flux: writing $\hat{\ell}_4 = e^{\delta_4}$, the relevant terms are $\frac{N_4^2}{\hat{\ell}^8} \delta\sigma_4 + \frac{1}{\hat{\ell}^2}\delta\sigma_4^2$, leading to a negligible shift $\delta\sigma_4 \sim \frac{N_4^2}{\hat{\ell}^6}$.

As noted in previous works such as [33,53], the flux potential depends on the axion fields, once the various flux quantum numbers are turned on. In the present case, the stabilization mechanism is simple, consisting of the three-term structure discussed above for the overall volume, along with the stabilizing effects of the Hessian from the Einstein action and of the warp factor in the other directions in field space. The flux contribution to the three-term structure cannot vary beyond the window in which the three terms admit a metastable solution for the volume with positive $V_{eff}$. So for simplicity, here we will work in the regime $cN_4 \ll N_7$ to avoid a significant change in the flux potential.

In other words, the 4d effective action becomes

$$\mathcal{S}^{(4)} = \mathcal{S}_{dS}^{(4)} + \int d^4x \sqrt{-g^{(4)}} \left( \dot{\phi}_c^2 - \frac{2M_P^3}{\hat{\ell}^{18}}\phi_c - \frac{M_P^2}{\hat{\ell}^{15}}\phi_c^2 \right), \tag{7.8}$$

plus generalizations for multifield models of the same sort. This expansion is useful in the regime $cN_4 \ll N_7$, with the first (linear) term in the axion potential dominating over the second (quadratic) term. This hierarchy is compatible with the large field range $\Delta\phi_c$, of order 10 Planck units, required for this form of large field inflation

$$\Delta c \ll \frac{N_7}{N_4} \Rightarrow \frac{\Delta\phi_c}{M_P} \ll \frac{N_7}{\hat{\ell}^3 N_4} \sim \frac{\hat{\ell}^3}{N_4}, \tag{7.9}$$

where in the last step we used the stabilization mechanism, balancing the flux term against the curvature term, $\frac{42}{\hat{\ell}^2} \sim \frac{N_7^2}{\hat{\ell}^{14}} \Rightarrow N_7 \sim \hat{\ell}^6$. We are free to incorporate a small $N_4$ here, which

anyway helps ensure that the 4-form flux potential not destabilize the metric moduli in the underlying de Sitter model (7.7).

We note that – as also shown explicitly in previous examples of axion monodromy from string theory such as [30,31,33] – there is not a significant buildup of low-energy fields along this field range. Since the $N_7$ contribution to the generalized flux potential dominates throughout the process, with the conditions laid out here we have negligible changes in $\ell$ and $\ell_4$, implying negligible effects on the corresponding Kaluza-Klein masses.

In this regime, the slow roll parameter $\varepsilon_V$ is given by (3.28)

$$\varepsilon_V = \frac{1}{2} \sum_I \left( \frac{\partial_{\phi_{c,I}} V_{eff}}{V_{eff}} \right)^2 \frac{1}{G_N} \sim \left( \frac{M_P^3/\hat{\ell}^{18}}{C/G_N} \right)^2 M_P^2 \sim \frac{1}{\hat{\ell}^{36}} \left( \frac{M_P}{H} \right)^4 , \qquad (7.10)$$

with $\eta \propto \partial_{\phi_c}^2 V \simeq 0$ in the regime of linear potential. Incorporating the current $2\sigma$ bound on the tensor/scalar ratio $r$ [75,76] requires roughly $H/M_P < 10^{-7}$. Thus to obtain slow roll inflation in the single-field case here requires

$$\varepsilon_V < 10^{-2} \Rightarrow \hat{\ell} > 10^{5/6} , \qquad (7.11)$$

which is a simple criterion to satisfy in our model. In the multifield context, which is more generic, one finds a similar value of $r$ but variable tilt of the primordial power spectrum according to simple statistical studies such as [77]. It would be interesting to incorporate this and other, more detailed aspects of our class of models into the axion dynamics. One aspect that requires further study and modeling is the exit from inflation.

Other forms of inflation also suggest themselves in this context. With warped regions near the ends of cusps (or other systoles), one may consider wrapped M5-brane inflation. The functional formulation of slow roll parameters in §3.2 and §C enables a study of much more general cosmological dynamics in this landscape, both analytically and numerically. On the latter front, we next set up a more general approach to numerically analyzing the landscape, in collaboration with neural networks.

# 8 Internal fields and neural networks: further exploring the landscape of hyperbolic compactifications

In the previous sections we have described our dS construction starting from a simple mechanism for stabilizing the volume, and then filling in essential details of the warp and conformal factor variation, in a large radius regime available via choice of discrete parameters. We illustrated these results and tested the general arguments for the positive Hessian via an explicit backreacted solution in a patch for each cusp. These solutions, displayed in §5.3 exhibited several properties of interest: tuning of our parameter $a$, field variations matching those predicted analytically, and second order stability. Moreover, the cusps are a significant fraction of the total volume of the space in specific examples such as those in [25] described in appendix A. But these solutions describe purely radial evolution (and even at this radial level, they involve a special choice of Dehn filling without radial evolution of the transverse torus shape). The analytic analysis points to similar behavior in the full internal solution, and we view the radial solutions as a nontrivial test of this.

A natural next step would be to find the explicit backreacted configurations in the whole internal space for a particular choice of hyperbolic manifold, providing a further numerical test of our results and determining more details of the effects of inhomogeneities of the sources. This would enable exploration of the landscape further beyond the fiducial solutions to the

constraint equation (3.7) that we discussed above. Moreover, methods for approximating internal solutions yield more general accelerated expansion related to the functional slow roll parameters discussed in §3.2 and §C. As we will see in this section, this direction of research meshes very well with modern neural network techniques.

Before continuing, however, we should stress that the finest details are not directly relevant to the existence of the metastable de Sitter solutions, and in any case the setup is subject to small additional quantum corrections as in §4.2 and subleading details of the Casimir stress energy. But it is appealing to pursue here given the concrete manifolds available [15] such as those in §A. So far we used these to establish the tuning parameters we used in our construction (5.21), but it is a well posed problem to analyze the equations of motion plus one-loop quantum stress energy in explicit examples.

For a general compactification this level of detail is often out of reach even in many supersymmetric compactifications where the fiducial Ricci flat metric is not known in general. Compactifications where the starting internal space is a Calabi-Yau manifold are well developed even when the starting metric is not known (for recent progress on Calabi-Yau metrics, see [78] which analytically constructs metrics for K3, and [79,80] which develop methods for obtaining Calabi-Yau and more general SU(3) structure metrics numerically).

The absence of an exact solution is true also in empirical physics quite generally – e.g. the standard $\Lambda CDM$ cosmological model is under extraordinarily good control without needing a specific description of each element of structure in the universe. Even there, however, the fact that cosmic censorship – the screening of singularities by horizons – is an open question is one indication that further study of PDEs and their relation to singularity theorems remains worthwhile. Indeed, the prominent role that the warp factor plays in screening would-be negative energy instabilities in compactifications [5] suggests deeper parallels between the two problems.

An interesting existing possibility for a more detailed description of string compactification is when the setup admits a cohomogeneity one formulation, where the Einstein equations reduce to a set of ODEs. There it is sometimes possible to construct explicit numerical bulk solutions with a $dS_4$ factor, although boundary variables and defects can be subtle (see [56,81] for some recent examples). This approach can be understood [63] as arising from an unfixed modulus in one higher dimension, reduced to $4d$. The unfixed scalar evolves in a radial version of FRW evolution in the extra direction. In general, a counting of boundary conditions allows for the possibility that suitable physically nonsingular objects can 'end the world' consistently within the range of the ODE solution.

Compared to previous choices of internal compactification, [13, 44, 45, 53, 55, 69], the higher dimensional hyperbolic setup considered in this paper has the advantage of both being generic, due to the dominance of negatively curved manifolds, and explicit, due to various constructive methods for building hyperbolic manifolds. In the following, we will discuss the general problem of finding explicit details of internal field configurations numerically, showing how this problem can be naturally rephrased with the language of Machine Learning. We will use our current hyperbolic example as reference to make the discussion concrete, but the general methods and ideas apply more broadly.

## 8.1 PDEs, boundary conditions and NN methods

Our problem is to solve the full set of Einstein+flux PDEs with the stress-energy tensor sourced by the flux, also taking into account the contribution of the automatically generated Casimir energy. These equations have to be paired with appropriate boundary conditions. In our current example, given the decomposition of hyperbolic manifolds in polytopes reviewed in Appendix A, these are translated to boundary conditions on the polytopes themselves. A standard method for constructing finite-volume hyperbolic manifolds is by a gluing of polytopes

via a pairwise map of their facets [15], as in the examples [25]. At each pair of facets thus glued, the boundary conditions are continuity on the fields and their first derivatives. One may alternatively work with the fields on a manifold with boundary, such as a polygon, including Neumann boundary conditions at its totally geodesic facets. We will present an explicit example of the latter in section 8.2, incorporating the physics of such boundaries in M theory [82]. Finally, since we are allowing for deformations away from the starting fiducial configurations, we also have to check a posteriori if the obtained result is self-consistent with the assumptions that entered in the stress energy tensor used. When we restricted our analysis to the constraint equation (3.7), we found that the dynamics of the warp factor limits the amount of negative energy.

As derived in [17] and discussed above, there are two equivalent ways to specify the relevant equations: from the point of view of the eleven-dimensional Einstein equations and from the point of view of the four-dimensional effective potential. These two points of view are related by the observation that the four-dimensional part of the eleven-dimensional Einstein equations, $\frac{\delta S_{11}}{\delta g_{11}^{\mu\nu}} + T_{\mu\nu}^{(Cas)} = 0$, produces the constraint (3.7), which is necessary to define the four-dimensional off-shell effective potential (3.4). This potential $V_{\text{eff}}[g_{(7)}, \phi_i]$ is an internal functional of the internal metric degrees of freedom and of the other fields of the eleven-dimensional theory, defined such that setting to zero the variations $\frac{\delta V_{\text{eff}}}{\delta g_{(7)}^{ij}}$ reproduces the internal part of the eleven-dimensional Einstein equations. Requiring stationarity of $V_{\text{eff}}$ also with respect to the matter fields $\phi_i$ enforces their eleven-dimensional equations of motion (see Appendix B.2 for more details). This approach discards the dynamics of four-dimensional vectors, which can be taken into account with an ansatz that includes non-diagonal metric terms, introducing more fields in the four-dimensional theory. As we are going to see in the next section, this way of organizing the equations, which we have employed in our analytical estimate and analyses in the rest of the paper, also has a natural application in the development of numerical methods to solve them.

Before getting into the details of the numerical methods, we can make the discussion a bit more concrete by quoting here the relevant PDEs. A way to organize the internal metric fluctuations, which we have used in §5.4 to analyze the Hessian, is to split them in volume-preserving fluctuations $h$ and conformal variations of the internal metric $B$. This results in the parametrization of the eleven-dimensional metric as in (1.1):

$$g_{11} = e^{2A}g^{(4)} + e^{2B}(g_{\mathbb{H}}^{(7)} + h). \tag{8.1}$$

For simplicity, we will now set $h = 0$. Defining $u = e^{2A}$ and $v = e^{\frac{5}{2}B}$, the external part of the eleven-dimensional equations of motion produces the constraint (3.7) specialized to the geometry (8.1) with $h = 0$:

$$\frac{\delta S_{11}}{\delta g_{11}^{\mu\nu}} : 0 = \frac{\nabla^2 u}{u} + \frac{8}{5}\frac{\nabla^2 v}{v} + 2\frac{\nabla u}{u}\frac{\nabla v}{v} - \frac{1}{3}R^{(7)} - \frac{1}{6}\frac{v^{4/5}}{u}R^{(4)} + \frac{1}{6}f_0^2\frac{v^{4/5}}{u^4} - \frac{1}{3}\ell_{11}^9\frac{\rho_c(y)}{v^{18/5}}, \tag{8.2}$$

while the internal trace, corresponding to variations with respect to $B$, gives

$$\frac{\delta V_{\text{eff}}}{\delta v} : 0 = \frac{\nabla^2 u}{u} + \frac{\nabla^2 v}{v} + 2\frac{\nabla u}{u}\frac{\nabla v}{v} + \frac{3}{8}\left(\frac{\nabla u}{u}\right)^2 - \frac{5}{24}R^{(7)} - \frac{7}{24}\frac{v^{4/5}}{u}R^{(4)} - \frac{7}{48}f_0^2\frac{v^{4/5}}{u^4} + \frac{1}{6}\ell_{11}^9\frac{\rho_c(y)}{v^{18/5}}. \tag{8.3}$$

In these variables,

$$f_0 = \ell_{11}^6 \frac{N_7}{\int d^7 y \sqrt{g}\, u^{-2} v^{14/5}}, \tag{8.4}$$

and all the metric-related quantities refer to the fiducial hyperbolic metric $g_{\mathbb{H}}^{(7)}$. Such a truncated parametrization allows us, for example, to discuss the backreaction of the conformal

factor as we did in section 5.2.1. More refined analyses, suited for example to the study the backreaction in the filled cusp, require us to take into account some of the degrees of freedom in $h$. We will see an explicit example in section 8.2. Ultimately, one would like to consider general fluctuations $\{B, h, \phi^i\}$, which is equivalent to solving all the eleven-dimensional equations of motion. We will not perform the full numerical analysis for our current example, but in the next section we propose a method to make this tractable for general flux compactifications.

### 8.1.1 Neural Network methods for solving PDEs

Recent years have seen massive development of Artificial Intelligence methods to tackle a wide range of computational problems, such as computer vision, language prediction, protein folding [83,84] and numerous applications to scientific data analysis.

The general framework is to model the solution to the problem at hand as an unknown function, and to design a method that "learns" a good approximation to this unknown function. In the most common realizations, the learning phase can be done from a fixed sets of known input-output pairs (i.e. learning by examples), by autonomously exploring a big space of possibilities and being rewarded for each success, or by learning together with an adversary that tries to fool the learner. These methods go under the broad name of Machine Learning.

The function that has to be learned is often parametrized by a Neural Network, and the fact that a good approximation exists is guaranteed by the Universal Approximation Theorem [85,86]. This is similar in spirit to Fourier analysis, but it is non-linear in nature. To make the discussion more concrete, a Feed-Forward Neural Network with $n$ layers and activation function $\sigma$ is defined as the nested function

$$N(x; W, b) \equiv W^n \sigma(W^{n-1} \sigma(\ldots \sigma(W^1 x + b^1)) + b^{n-1}) + b^n, \tag{8.5}$$

where $x \in \mathbb{R}^k$ is the input and the $W^i$ and $b^i$ are respectively matrices and vectors of appropriate dimensions, called weights and biases. The width of the $i$-th layer is the dimension of $b^i$. Together, they form the set of parameters $\theta$ that have to be learned. The nonlinear activation function $\sigma : \mathbb{R} \to \mathbb{R}$ is applied to each element of its vector argument. The Universal Approximation Theorem guarantees that (8.5) can approximate arbitrarily well a large class of functions.[15] The functional form in (8.5) is the simplest *architecture*, but many others have been developed with features adapted to the problem at hand or that enjoy some particular properties. Neural Networks with more than one layer are commonly called *deep*.

Since the equations that would give the best estimate for the parameters cannot in general be solved (in the Fourier analogy, these would be the equations that fix the the Fourier coefficients) the learning problem is often translated to an optimization problem, where, schematically, the problem of solving the equations $E_i(\theta) = 0$ is translated to the problem of minimizing $\mathcal{L}(\theta) \equiv \sum_i E_i(\theta)^2$ in the space of $\theta$'s.

Various optimization algorithms designed to tackle this problem exist, with the more common ones based on some form of gradient descent.[16] As the name suggests, this often requires the computation of the gradients of the function $\mathcal{L}(\theta)$, called *loss function*, with respect to parameters $\theta$ defining the model (or ansatz). Technically, this is done by using an algorithm called automatic differentiation, which allows to compute the gradients of very complicated functions efficiently and with small numerical error. Software packages such as *PyTorch* and *TensorFlow* have been developed to apply this technique automatically and to exploit various hardware architectures. We notice that the efficiency of these packages developed for machine

---

[15]The original version of the theorem applies to networks with a single layer and arbitrary width, but it has been refined to include more general forms, such as in [87,88].

[16]In other work with G. Panagopoulos we are developing an optimizer based on Born-Infeld dynamics with loss function related to the target spacetime warp factor.

learning tasks can also be fruitfully exploited to find numerical solutions to generic problems that require minimizing a very complicated function, even if this is not the result of Neural Network ansatz. For example, in [89–91] this has been used to directly minimize the scalar potential of $\mathcal{N} = 8$ supergravities, finding in this way new four- and five- dimensional AdS vacua.

The problems described up to now are, in a sense, algebraic. They require to find an unknown function by exploiting some known relations between its input and outputs such as training data examples in supervised learning. When dealing with differential equations such relations are not known, but what are known are relations between the derivatives (with respect to the inputs, not the parameters) of the unknown function. With this idea in mind, AI methods, in particular in their incarnation through Neural Network approximations, have also been proposed to solve partial differential equations. These were first introduced in [92] and have recently received renewed attention, see for example [93–96].

The simplest method consists of randomly sampling some points $x^m$ on the domain and directly using a Neural Network as ansatz for the solution. In this approach the loss function is taken to be

$$\mathcal{L}(\theta) = \sum_i \sum_{\{x^m\}} E_i(\theta)^2 \,, \tag{8.6}$$

where $E_i$ are now the PDEs, in which the unknown functions are substituted by $N(x; \theta)$. Evolving the parameters toward low loss function then produces approximate solutions to the PDEs. In our context, sufficiently good approximations to solutions yield inflationary dynamics with small $\varepsilon_V$ and $\eta_V$ as defined in §3.2C. To solve differential equations it is not enough to solve them in the bulk, but appropriate boundary conditions have to be imposed. Different approaches have been developed to take those into account, and can be organized in "hard" and "soft" methods. In the former case one starts with a parameterization which already enforces them, while in the latter BCs are solved at the same time as the equations. In the simplest approach one samples points $x^a$ on the boundary and considers the combined loss

$$\mathcal{L}(\theta) = \sum_i \sum_{\{x^m\}} E_i(\theta)^2 + \sum_j \sum_{\{x^a\}} BC_j(\theta)^2 \,. \tag{8.7}$$

Other approaches have also been proposed, such as adversarial ones [97] or using neural networks as a non-linear approximation of the finite differences coefficients [98] . Although there is some encouraging early success in situations where traditional numerical methods perform poorly, such as PDEs in a high number of dimensions [93] or on very complicated domains [99], to date, a complete theory of Neural Network methods for solving differential equations has not been developed.

For example, there is no guarantee that these methods converge to a solution of the original problem, since (depending on the optimizer) gradient-based optimization methods will generically converge to local minima, where $\mathcal{L}$ is generically non-zero. For more common machine learning tasks this might not be a problem, since often local minima perform better in generalizing to unknown examples, but for the application to PDE solving it can be problematic, since local minima could correspond to configurations that do not solve the original equations. This price these methods pay is often compensated by other characteristics, such as scaling much better for high-dimensional problems and being more versatile. Moreover, as noted above, approximate solutions are of interest in the present context since those may give new examples of slow roll inflation. In the remainder of this section we want to underline how the features just presented make Neural Network methods natural candidates to numerically explore the rich landscape of string/M theory compactifications beyond the part accessible with analytical methods.

As discussed in section 8.1, once the constraint (3.7) has been taken into account, the eleven-dimensional equations of motion are equivalent to minimizing the effective potential (3.4) with respect to the internal fields. This draws a direct parallel between the loss function (8.6) and the slow-roll parameter $\varepsilon_V$ as in (3.28)

$$\varepsilon_V = \frac{G_N}{8C^2} \sum_I (\partial_{\phi_c,I} V_{\text{eff}})^2 \,, \tag{8.8}$$

since the differential equations are functional derivatives of the effective potential $V_{eff}$. [17] In other words, minimizing the loss function corresponds to minimizing $\varepsilon_V$. Finding a configuration where $\mathcal{L} = 0$ would correspond to a bona fide dS solution, while a local minimum where $\mathcal{L} \propto \varepsilon_V \ll 1$, describing an approximate solution to the PDEs, would correspond to a more general accelerated expansion. It is also natural to combine such a method with analytical estimates, by using approximate analytic solutions as the starting point for the descent in the $V_{\text{eff}}$ landscape. In this process, for the effective potential to be well defined, both the constraint (3.7) and the relevant boundary conditions have to be enforced. In this paper we have checked that for our current model the starting point is well-defined by solving the constraint equation with different methods in different parts of the internal geometry. The need to solve the constraint (as is familiar in GR where one requires good initial data) provides some interesting limitations on the landscape, as discussed in [17]. The constraint and the boundary conditions have to be enforced along all the flow, and a simple strategy to enforce them could be to add these conditions to the loss function, with a big penalty factor that weights them more. This will change the shape of the resulting loss, and as for the general boundary condition problem, it would be important to develop a theory able to distinguish and organize the different possibilities.

Finally, we also notice that a more direct approach could be attempted. For a fixed choice of internal fields, which can be approximated with a Neural Network depending on some parameters $\theta$, the constraint equation (3.7) is a linear inhomogeneous equation for $u$. This equation could be directly solved for $u$ by inverting a numerical operator, defining the off-shell effective potential as an integral of $u$ as in (5.17). If the solution for $u$ is computed with a method that allows an automatic computation of the gradients with respect to the parameters $\theta$, the resulting effective potential can be directly used as a loss function, without first deriving the equations of motion. Once the boundary conditions are properly taken into account, this approach allows a direct exploration of the landscape of flux compactifications, where any local minimum corresponds to a metastable solution.

## 8.2 UV complete warmup: M theory on $\mathbb{H}_3/\Gamma$

The aim of this small section is to show a concrete example of domain and boundary conditions specification for the internal PDEs. As a simpler UV complete warm-up we can consider an internal three-dimensional space (thus describing a $\text{dS}_8$ solution), with vanishing flux. The choice of a lower number of internal dimensions is to allow for a simple numerical exploration. As an extra simplification, we can work with a single polytope, imposing Neumann boundary conditions at its faces. Physically, these boundaries correspond to Horava-Witten walls in M-theory. This further simplification is not in general required since, as described in Appendix A, we can work with more general three-dimensional hyperbolic manifolds by starting with a single polytope and acting on it with $\Gamma$, the Coxeter group of reflections along its facets.

---

[17]More precisely, if as in Monte Carlo methods the internal integral in the definition of $V_{\text{eff}}$ (see also Appendix C) is estimated by randomly sampling points $\{x^m\}$ in the internal space, the explicit slow-roll functional (8.8) is proportional to the loss defined in (8.6). In other words, from (C.15) one can see that both formulas involve an integral (sum) over points.

This would only impose the milder requirement of continuity of the functions and their first derivatives. In the following, we will describe the domain and boundary conditions in detail, together with a proof of concept of the methods introduced in section 8.1.1. The resulting configuration is not meant to be physically complete, but it provides a concrete example of an application of the neural network methods to this class of physical problems.

We can construct a simple three-dimensional polyhedron working in the upper-half space model as follows. As reviewed in Appendix A, totally geodesic submanifolds, which will be the faces of our polyhedron, are either vertical planes, or hemispheres centered on the $z = 0$ boundary. For the set of reflections along the faces to form a Coxeter group, all of them have to meet with dihedral angles $\frac{\pi}{k}$, with $k$ an integer. The easiest polytopes one can construct in this way are simplexes, which exist for $n \leq 9$ and have all been tabulated (see for example [15, Chapters 6-7] for more details). A simple three-dimensional construction consists on taking a single hemisphere centered at the origin and four vertical walls, whose cross section at $z = \text{const.}$ forms a square centered at the origin. We show this construction in Figure 8, together with the constraints that have to be satisfied in order for the resulting polyhedron to have correct diehedral angles and finite volume.

To make this geometry dynamical, we want to allow fluctuations around the hyperbolic metric. A complete analysis requires to take into account all the metric degrees of freedom, but for illustrative purposes we focus on the simpler deformation

$$ds_{11}^2 = e^{2A}ds_4^2 + e^{2Q}dz^2 + R_c^2 dx_1^2 + R^2 dx_2^2, \tag{8.9}$$

where $A, Q, R_c, R$ depend on all the internal coordinates.

Since in the fiducial hyperbolic metric $e^Q = R = R_c = z^{-1}$, at the hemisphere we impose Neumann boundary conditions (with respect to the UHS metric) for the function $A$ and for the deformations away from the hyperbolic metric: $zR_c$, $zR$, $ze^Q$. With this approach we are implicitly assuming that in the simple deformation (8.9) the internal metric approaches the hyperbolic one at the hemisphere. In a more complete setup, the full set of boundary conditions amounts to a Neumann condition for $A$ (with respect to the completely deformed internal metric) and vanishing of the internal extrinsic curvature at the boundaries of the polytope.

All in all, on the vertical walls we impose the boundary conditions

$$\partial_{x_i} R = \partial_{x_i} R_c = \partial_{x_i} A = 0 \quad \text{vertical walls}, \tag{8.10}$$

and at the hemisphere we impose

$$
\begin{aligned}
0 &= z\partial_z A + x_i \partial_{x_i} A & \text{hemisphere}, \\
0 &= z\partial_z R + R + x_i \partial_{x_i} R & \text{hemisphere}, \\
0 &= z\partial_z R_c + R_c + x_i \partial_{x_i} R_c & \text{hemisphere}, \\
0 &= z\partial_z Q + 1 + x_i \partial_{x_i} Q & \text{hemisphere}.
\end{aligned}
\tag{8.11}
$$

Finally, we implement the Dehn filling condition, which caps off the geometry smoothly at $z = z_{DF}$:

$$z\partial_z R = -1, \quad R = 0, \quad \partial_z R_c = 0, \quad \partial_z A = 0 \quad \text{at} \quad z = z_{DF}. \tag{8.12}$$

Having introduced the framework, let us present a simple example of an approximate solution to the equations (8.2) (8.3) obtained via the neural network method introduced in section 8.1.1. We use the domain in figure 8 with $L = 2, d = 1$. For simplicity we also restrict the problem to just the warp factor and the internal conformal factor, by setting $e^{2Q} = R_c^2 = R^2 = e^{2B}z^{-2}$ and imposing the boundary conditions (8.10)(8.11) at the walls and hemisphere. In the figure 9 we show an example. This solution goes beyond our purely radially evolving patchwise

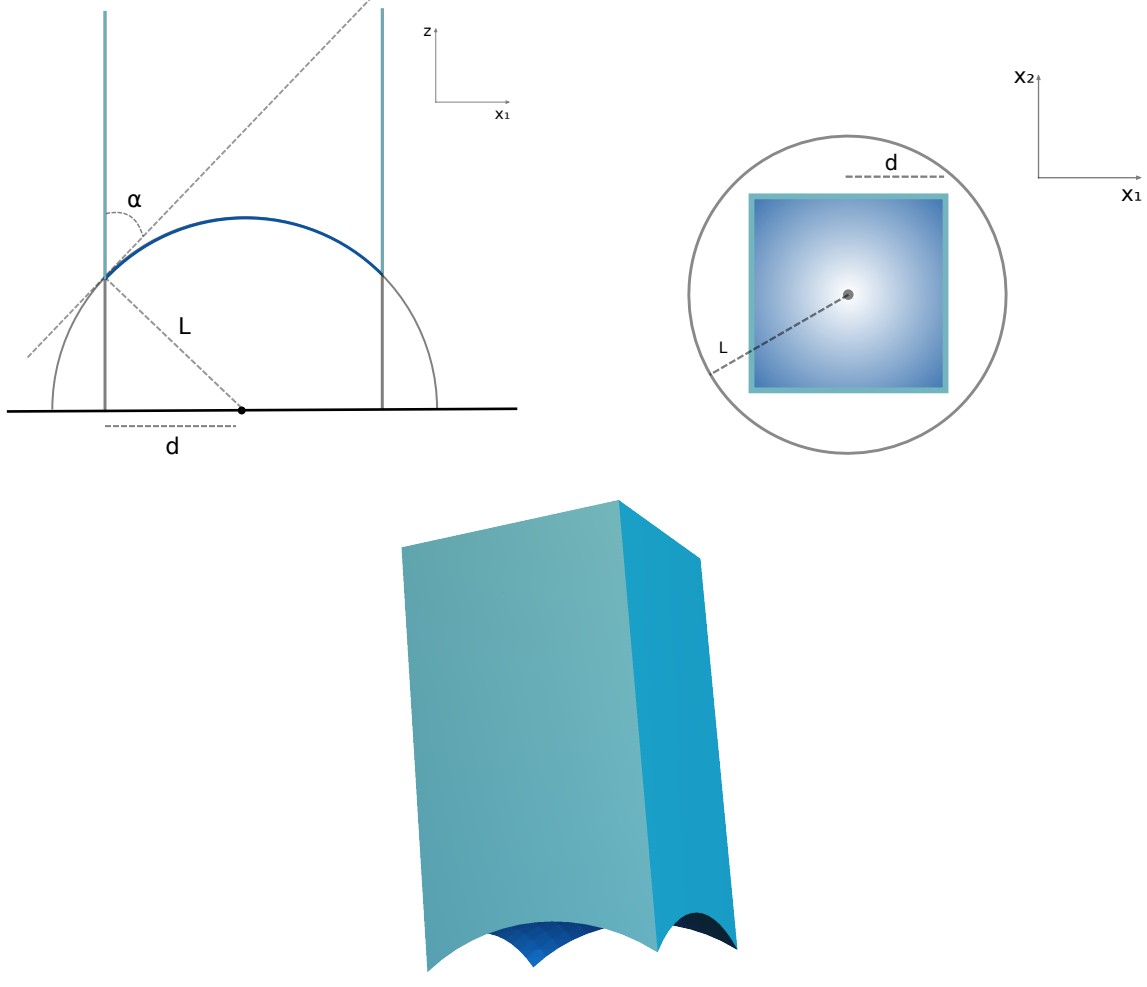

Figure 8: A simple three-dimensional polyhedron composed of a single hemisphere of radius $L$ centered at the origin of the upper-half space and four vertical walls with a square cross-section whose half-length is denoted by $d$. The dihedral angle $\alpha$ is given by $\cos \alpha = \frac{d}{L}$. For the polyhedron to have finite volume the square in the top-right view has to be inscribed in the circle, requiring $\frac{d}{L} \leq \frac{\sqrt{2}}{2}$. Imposing also $\alpha = \frac{\pi}{k}$, with $k$ an integer, leaves as the only possibilities $k = 3, 4$. At the bottom we show a 3d rendering of the $k = 3$ case.

solutions in §5.1.15.1.2 since the hemisphere boundary conditions are imposed. The solution here extends in upper half space coordinates from the hemisphere to $z_c = 7$, requiring some completion beyond that. In a human-NN collaboration, we have obtained a small $\varepsilon_V$ configuration from this by matching it to simpler geometries – solving the constraint equation (3.7) throughout, and bounding $\varepsilon_V$ (C.11). We leave the presentation of the details of this and scaled up generalizations to future works. In the $d = 4$ case of interest, a natural starting point would be manifold covers [27] of the small-volume spaces [52]. It is clear that these methods, combined with the explicit hyperbolic and Einstein geometries constructed mathematically, promise to significantly expand our reach in the landscape.

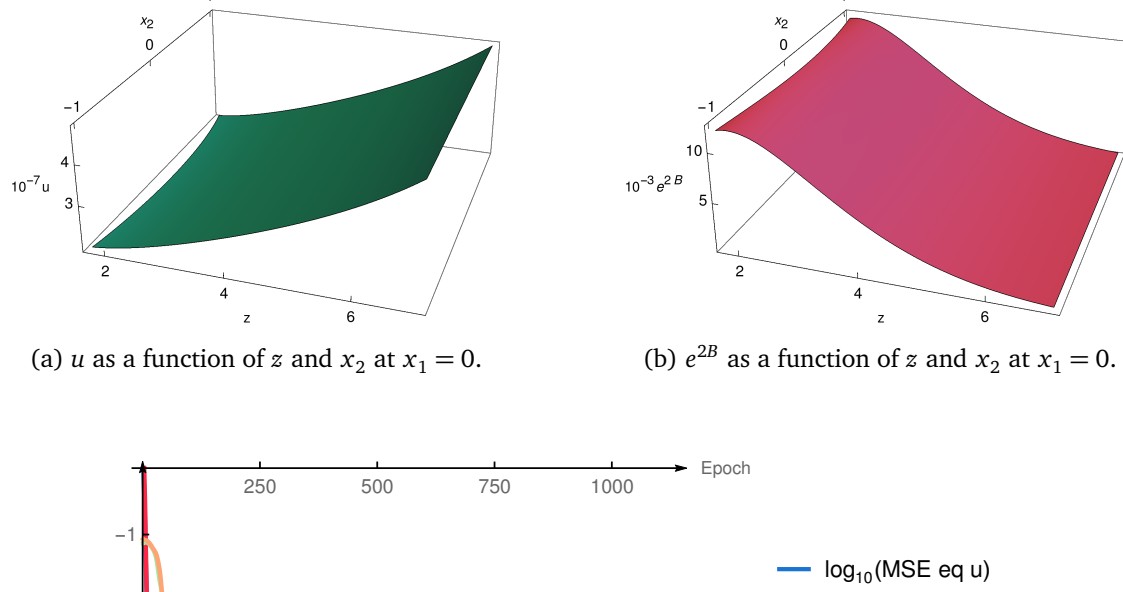

(a) $u$ as a function of $z$ and $x_2$ at $x_1 = 0$.

(b) $e^{2B}$ as a function of $z$ and $x_2$ at $x_1 = 0$.

(c) Behavior of the different components of the loss function during training.

Figure 9: An approximate neural network solution to the internal PDEs. In panel 9a, 9b we show $u$ and $e^{2B}$ as functions of the two-dimensional slice at $x_1 = 0$. In panel 9c we show the behavior of the different components of the loss function (8.7) during the training routine.

## 9 Discussion and Future directions

In this work, we obtained de Sitter and inflation models from hyperbolic compactifications of M theory. These have an effective potential (3.4) whose leading term arises from the negative curvature

$$\int \sqrt{g^{(7)}}(-R^{(7)} + \dots) = \int \sqrt{g^{(7)}} \left( \frac{42}{\ell^2} + \dots \right). \tag{9.1}$$

The internal curvature combines with flux, Casimir energy, and warping effects in $V_{eff}$ in a comprehensive mechanism to stabilize the internal dimensions, supporting accelerated expansion of the universe. The competition between the quantum Casimir energy and the classical contributions to $V_{eff}$ requires a stable small circle in the geometry, which arises in explicit hyperbolic manifolds $\mathbb{H}_7/\Gamma$ completed with the filling prescription [19] to Einstein spaces[18]. Tuning down the net curvature term, including warping effects, leads to a simple 3-term structure stabilizing the volume with Casimir energy and flux playing off against the curvature.

The rigidity of negatively curved manifolds in dimension $n \geq 3$ combined with the stabilizing effects of the warp factor dynamics [17] goes a long way toward addressing all the field deformations (with no separate dilaton as there is in perturbative string theory limits). Our analysis generalizes the stabilizing effects of warping to more general internal configura-

---

[18]and likely also other constructions of hyperbolic manifolds with small systoles [20,21]

tions, applicable near the dressed hyperbolic space augmented by appropriate varying warp and conformal factors relevant in our system (whose structure we tested in explicit radial solutions in filled cusps). General positivity results building on the warping and curvature effects, and some example calculations of masses, suggest a gapped Hessian as summarized in §5.4.2, with small residual tadpole shifts. It would be interesting to test this further and compute the Hessian in specific examples.

These results arise from detailed analytic estimates and patchwise numerical studies, controlled in a regime (5.21) that is obtained with explicitly available discrete parameters. The setup is concrete, so that further checks and details of the internal fields may be analyzed using modern numerical techniques, e.g. in collaboration with artificial neural networks. Axion physics and inflationary cosmology readily descends from this class of compactifications.

Negatively curved spaces are generic, explicitly constructed starting from a known metric, and produce a natural source of positive potential energy in four dimensions along with axion physics. This fits well with current empirical observations, which indicate positive potential energy leading to accelerated expansion of the universe, with bounds on super-partners and particle dark matter. Their explicitness and relative simplicity promise to facilitate studies of cosmological quantum gravity [8,10], with the present example a direct uplift of the M2-brane theory, a classic example of the AdS/CFT correspondence.

If this is the answer (9.1), what is the question [100]?

## 9.1 Context and further directions

de Sitter and inflationary constructions in string/M theory are not new; there are several classes of compactifications which exhibit a consistent playoff of forces. The control parameters in $\Gamma$, flux quanta, and the filling prescription are analogous to the dimensionality $D$ in [44,45,53,101], the flux superpotential $W_0$ in [55] [102,103], various topological conditions in [69] and the genus of the Riemann surfaces in [13]. The explicitness of the ingredients in the present work is analogous to [8,45,56], but simpler in several respects.

The Casimir source as the crucial negative contribution in the potential is quantum, similarly to [55], but perturbative, with explicitly known stress energy contributions to the equations of motion (as in other backgrounds such as [39,70]). It shares some features with orientifold planes (classical in string theory) which are essential in all previous de Sitter examples, as the negative stress energy is localized near the small circles with a metric that is somewhat analogous to O-plane geometries. But in our setup, the mathematical prescription for finite small circles – related to the beautiful subject of systolic geometry – combines with warp factor dynamics to produce a nonsingular configuration of sufficient negative energy. Both cases – the smooth systoles here and similar setups matching to O-planes or incorporating resolvable singularities, deserve further study. The competitive but finite negative energy contribution would likely mesh well with other modern studies of energy conditions [49,50].

There is ample evidence that different sectors of the landscape are connected, via known transitions changing topology [104–108], chirality [109], and even effective dimensionality [110–113], with dualities such as [114,115] relating curvature and dimensionality via the effective central charge. It would be very interesting to extend those relations to the present models, via M-brane dynamics. The dynamical connections indicate a unified theory. It is constrained by mathematical and physical principles, and rich with phenomena that carry an imprint of the structure of the microphysical theory [1,2,116]. These imprints affect both phenomenology descending from the theory and key microphysical aspects of quantum gravity [8,10]. As an uplift of the M2-brane theory, analogous to the uplift of the D1-D5 theory [8], the present models can potentially elucidate the holographic description of cosmology via a generalized $T\bar{T}$ deformation [117–119] including entropy counts [10,120]. The structure of de Sitter in string theory is also relevant for other related approaches such as [121–123]

and [124, 125]. For example, constructions of candidate wavefunctions of the universe, or other attempts at a measure, require at least semiclassical control of all dimensions. In the present context this essentially boils down to hyperbolic space (Euclidean AdS) internally in a warped product with 4$d$ de Sitter, with electric flux in 4$d$ and 1-loop quantum Casimir energy. The metastability – rather than stability of de Sitter in quantum gravity – a feature that is not immediately visible from the bottom up – is automathic here. This feature is crucial to existing consistency checks of cosmological quantum gravity such as those in [8] [10] and [7]. The counts of vacua and other statistics will benefit from beautiful mathematical results concerning distributions of volumes, their relations to systoles [21, 22], cusp densities and horosphere packings [126], the aforementioned counts of Einstein cusp fillings [19], and others. It would also be interesting to understand how to obtain a Standard-Model like matter content in our setup. Different avenues could be pursued here, based on Horava-Witten branes [82], chiral matter from singularities [127], or intersecting M-brane constructions. There could be novel connections with mathematical results on hyperbolic manifolds, including their topology [25] and orbifold constructions.

To sum up, what is remarkable about the present hyperbolic compactifications is their relative simplicity (in terms of the ingredient list) and naturally stabilizing rigidity properties in concert with warp factor dynamics – along with a much greater genericity than previously studied compactification spaces. This combination of simplicity and greater genericity is not unfamiliar, being somewhat similar to large-flavor expansions in physics more broadly (similarly to the large-dimension expansion). In these regions of the string/M theory landscape, positive potential energy and axion physics abound. Such regimes may well represent the typical behavior of the string/M theory landscape, and invite further work on applications to both phenomenology and abstract quantum gravity.

# Acknowledgements

We are grateful to Ian Agol and Bruno′ Martelli for extensive help navigating the rich mathematical landscape of hyperbolic geometry and many useful suggestions. We would also like to thank Michael Douglas for detailed discussions of aspects of this work, and Juan Maldacena for useful discussions and comments on a previous version of the manuscript. We would also like to thank Michael Freedman and John Lott for very helpful comments. We are additionally grateful to Alessandro Tomasiello for comments on the manuscript, and Shamit Kachru, Rafe Mazzeo, Liam McAllister, Jakob Moritz, and Steve Shenker for interesting discussions. The research of GBDL and ES was supported in part by the Simons Foundation Origins of the Universe Initiative (modern inflationary cosmology collaboration), by a Simons Investigator award, and by the National Science Foundation under grant number PHY-1720397. ES would like to thank Centro Atómico Bariloche for hospitality during part of this work. GT is supported by CONICET (PIP grant 11220150100299), ANPCyT (PICT 2018-2517), UNCuyo, and CNEA.

# A  Virtual hyperbolic yoga: Explicit examples of compactification manifolds with discrete tuning parameters

Here we briefly collect mathematical features of the compactification manifolds we use in the main text. Our construction makes use of finite-volume Einstein manifolds, hyperbolic aside from possible filled ends of cusps [19]. The key feature for our physical construction is small systoles around which fermions have antiperiodic boundary conditions, supporting Casimir energy, along with an upper bound on the overall volume where negative internal curvature

and 7-form flux are supported.

Specific examples of constructions with small systoles include:

- hyperbolic manifolds obtained via the *inbreeding* construction [20–22]

- hyperbolic manifolds with cusp ends filled in with a higher dimensional analogue of Dehn filling to form a closed Einstein space [19]. There are many examples. One particularly simple class of examples of cusped hyperbolic manifolds appears in [25]. In [19], for any given cusp there is a choice of Dehn filling corresponding to any of the simple closed geodesics on the cross sectional $T^6$, leading to finely tunable systole size. It is guaranteed that any finite volume hyperbolic space is *virtually* spinnable, i.e. a finite cover admits a spin structure [128][19]. The antiperiodic boundary conditions for fermions that we require are natural: if a small circle is contractible, the boundary conditions must be antiperiodic, in order to smoothly match to the fact that $spinor \rightarrow -spinor$ upon $2\pi$ rotation about a point (and if it is not contractible, it is an option to assign this boundary condition consistently).

We also require an upper bound on the volume in order that the three contributions to the 4d effective potential – total curvature (including warp factor gradients), Casimir energy, and flux – can compete as in (5.21). The manifold must also be consistent with the hierarchy $\ell \gg R_c \gg \ell_{11}$ that requires $\epsilon \ll 1$ in (4.11), implying an upper bound on the total volume.

The construction of hyperbolic manifolds [15] beautifully combines group theory and geometry, with numerous general theorems and explicit constructions available in the mathematical literature. These range from elementary constructions gluing polygons, to sophisticated group theoretic studies of various properties of subgroups of hyperbolic isometries. One method starts from a hyperbolic orbifold, such as a polygon whose facets are fixed points of a reflection subgroup of the group of hyperbolic isometries. Simple examples of this, with small volume, for dimension $n \leq 9$ appear in [26]. A result known as Selberg's lemma guarantees that a torsion-free subgroup $\Gamma$ exists, yielding a manifold $\mathbb{H}_n/\Gamma$ as a cover of the orbifold; such examples appear in [27]. A related method constructs a pairwise gluing of the totally geodesic facets of a set of hyperbolic polygons in a way that avoids singularities in the resulting space.

The recent paper [25] achieves this by a gluing of elementary right-angled polygons. The group $\Gamma$ is derived there as a freely action subgroup of the orbifold group defining the fundamental right-angled polygon. Although in flat Euclidean space, manifolds constructed from right-angled polygons – i.e. tori – contain moduli, in hyperbolic space these manifolds are rigid.

One example satisfying our conditions is the following. We start with the class of examples in [25]. The set of 7-manifolds in [25] includes a minimal example with 4032 cusps and volume $\sim 1.3 * 10^5$, so that the ratio $n_c/v_7$ appearing in (4.11) is $\sim 1/30$. A tuning $\Delta$volume $\ll$ volume related to the tuning described in the main text (4.13) may be achieved as follows. We fill cusps with Anderson's generalization of Dehn filling [19], choosing a different simple closed geodesic for different cusps in that construction – similar to a different value of $y_c$ in the cusp (4.1) for different cusps – in order to vary the volume of the total space. This variation in the volume is very small compared to the volume, as needed in (4.13). For our purposes, we wish to change the volume contained in bulk regions with negligible Casimir energy compared to that in Casimir regions. Indeed, as described below (4.13), we may add 'bulk' volume with negligible Casimir energy by filling some cusps such that their shortest geodesic is not small.

Another way to vary the volume, and hence the $\int -R^{(7)}$ contribution to the effective potential, is to take covers of the manifold in [25]. Manifest in that construction are totally geodesic codimension 1 submanifolds $H$ which thread through the cusps of the $n$-manifolds in that construction. Cutting the manifold open along $H$, replicating the resulting space, and joining $k$ copies together produces a cover of the original manifold which has an asymmetric

---

[19]enabling one to sign up for virtual spin classes along with yoga.

cusp extended by a factor of $k$ along one of the cross sectional directions of the $T^6$. Varying $k$ in this procedure changes the volume by an amount small compared to the total volume, again achieving $\Delta\text{Vol}_7 \ll \text{Vol}_7$. Again, this may be done in a way that varies the physical quantities of interest: the cover may proliferate some cusps while extending others, changing the relative volume contained in Casimir regions compared to bulk regions in the manifold.

More generally, there are numerous studies of the distribution of hyperbolic manifolds with various properties. There are results about the count of manifolds as a function of volume, cusp density (related to horosphere packings), and other quantities. With a wealth of explicit information about this class of compactifications, whose metric is well known, there is much room for further explicit study of this region of the M theory landscape.

## A.1 Hyperbolic manifolds from right angled polygon gluings

For readers who wish to delve into more details of the manifolds obtained in [25] as tessellations of right-angled polygons, in this section we describe more aspects of this construction and illustrate the case in $n = 3$ dimensions. The manipulations here also enable one to get a feel for the rigidity of hyperbolic manifolds.

Working in the upper half space $z > 0$ with metric

$$ds^2 = \frac{dz^2 + \sum_{i=1}^{n-1} dx_i^2}{z^2}, \tag{A.1}$$

the totally geodesic codimension one hypersurfaces, patches of which serve as facets of hyperbolic polygons, consist of

- vertical walls

$$\sum_i c_i x_i = 0, \quad c_i = const, \tag{A.2}$$

and

- hemispheres centered at $z = 0, x = x_0$

$$z^2 + \sum_i (x_i - x_{0i})^2 = R_{hem}^2, \tag{A.3}$$

for constants $x_{0i}, R_{hem}$.

There are many kinds of hyperbolic polygons whose facets are fixed loci of a reflection group [15, 129]. The examples in [25] have a further simplifying feature that they are built from right-angled polygons, for which all adjacent sides are at right angles. We illustrate this for $n = 3$ in upper half space variables in figure 10.

To construct a manifold from a right-angled polygon with a color scheme consisting of $c$ distinct colors, we mirror across the facets in such a way that we mirror on each color once. This produces a space tessellated with $2^c$ copies of $P_3$. We then glue the facets in the following way. Any facet of color $\gamma$ is identified with its image under reflection about any facet of color $\gamma$. The resulting space is a hyperbolic manifold of finite volume as depicted in figures 11-12.

Upon mirroring and gluing, each facet becomes a thrice-punctured sphere. With a little more work one can see other properties, such as the fact that the ideal vertices join up into a total of 3 cusps.

The warmup we illustrated here is for a minimal color scheme. Many other examples are possible. For example, we may start from the right angled polygon on the left in figure 11 and assign more distinct colors, e.g. to the hemisphere patches on the underside. Mirroring on each of them produces new ideal vertices that form cusps upon gluing. As already mentioned, this structure generalizes to higher dimensions, including $n = 7$. Their properties are worked out explicitly in [25].



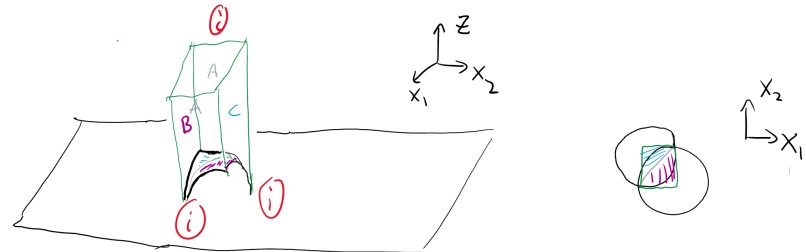

Figure 10: The polygon $P_3$ contains 6 facets drawn in the upper half space. Four are vertical walls (A.2) and two are patches of hemispheres (A.3). On the right we have shown the projection of the bottom of the figure to z=0. The vertices labelled by "i" are ideal vertices, building blocks for cusps. A minimal color scheme is included following the rules in [25], with no adjacent sides having the same color.

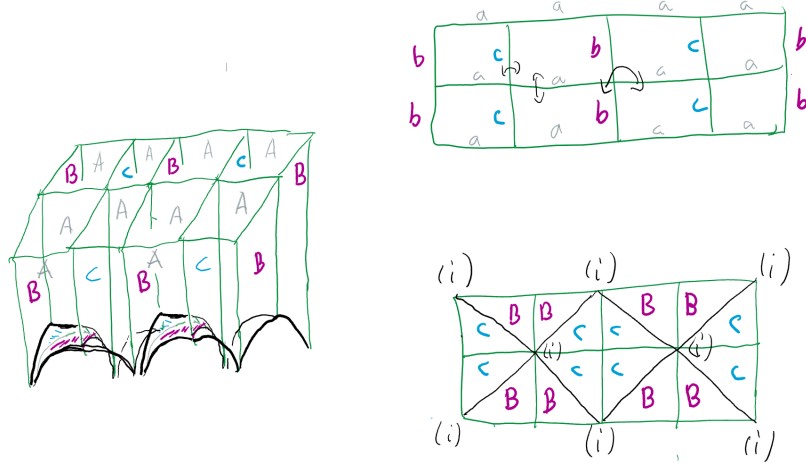

Figure 11: Sketch of the space obtained by mirroring across each color once. On the left we sketch the entire space. The opposite vertical walls are identified. The upper right is a top view giving the cross section of the cusp at $z \to \infty$ (a.k.a. its link). Although this cross section, a 6-torus, would have continuous moduli by itself, these are absent when it is connected to the rest of the space. The lower right is the mirrored color scheme on the underside of the figure. Hold this pose – the gluings for the hemisphere patches on the underside are treated in the next figure.

In this class of manifolds, the volume contained in the cusps is a significant fraction of the volume of the space. As is standard, we define the cusp volume the volume contained in (4.1) with a minimal value of the radial coordinate $y = y_{min}$ such that the cusp fits inside the manifold. At the level of an ideal vertex of the underlying right angled polytope, we similarly identify a minimal $y$ such that the ideal vertex up to $y_{min}$, with its cross sectional Euclidean $(n-1)$-polytope, fits in the $n$ dimensional polytope. These polytopes themselves are gluings of simplices, as described in [130]. A simple (computer-aided) calculation for the relevant $n = 7$ simplex has a ratio of cusp volume to full volume of .34. This is a lower bound on the fraction of volume contained in cusps for our $n = 7$ case, showing that it is a significant fraction of the total volume of the space.

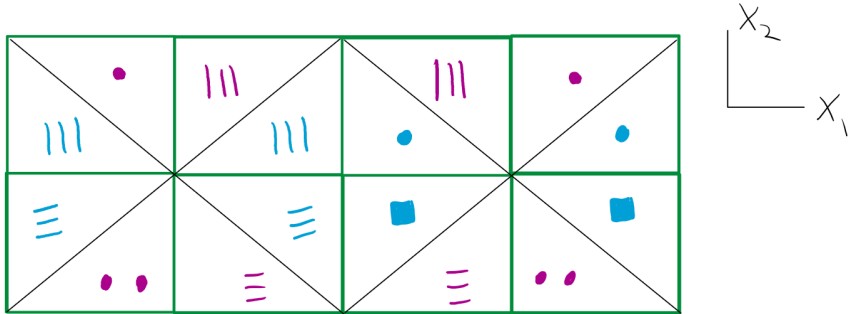

Figure 12: Here we depict the gluing procedure for the minimal hyperbolic 3-manifold in [25], gluing each color to its image under reflection about any facet of its color. As above, color B is pink and C is blue.

Two specific generalizations are of interest in our de Sitter and inflation construction. One is to vary the bulk and cusp volumes, obtaining a difference $\Delta\mathrm{Vol}_7 \ll \mathrm{Vol}_7$ related to (4.13). One way to do that is to take $k$-fold covers of our underlying manifold,[20] cutting along a totally geodesic manifold – such as the left and right walls of the left image in figure 11. Replicating this space $k$ times before identifying the walls yields a volume proportional to a tunable parameter $k$. The second generalization is to incorporate the filling developed in [19]. By varying the size of the systole which is inversely related to the length of a chosen simple closed geodesic in the filling procedure [19] (reviewed in the next subsection), we get an independent parameter useful for separately varying the integrated quantities in (5.21). In addition to explicitly providing the parameters required in our construction, concrete examples such as these may be used in more detailed numerical studies.

## A.2 Summary of the Dehn Filling to an Einstein space with small systole

Let us provide a short description of the construction [19] for Dehn filling of hyperbolic cusps by an Einstein manifold. As stressed above and in [19], via a set of discrete choices this yields a large number of Einstein manifolds for a manifold with $n_c$ cusps. Mathematically the number of choices of filling is infinite for each cusp, while physically we will restrict attention to the finite number for which the length of the systole exceeds $\ell_{11}$.

An approximation of the filling in any dimension $n$ (with $n = 7$ for our setup) is obtained as follows. There is a filling for each choice of simple closed geodesic $\sigma$ in the cusp cross sectional $T^{n-1}$. Note that there are infinitely many such geodesics, with sequences that go off to infinite length. We pick such a geodesic $\sigma$ with length $|\sigma|$ in a given torus metric $ds^2_{T^{n-1}}$ of size $\sim \ell$.[21]

We will join the cusp metric (4.1) in the form

$$ds^2 = \frac{dr^2}{r^2} + \frac{r^2}{r^2_{join}} ds^2_{T^{n-1}}, \tag{A.4}$$

---

[20]We thank I. Agol for this suggestion.

[21]A more specific specification (of this order of magnitude) is detailed in [19] in terms of the injectivity radius.

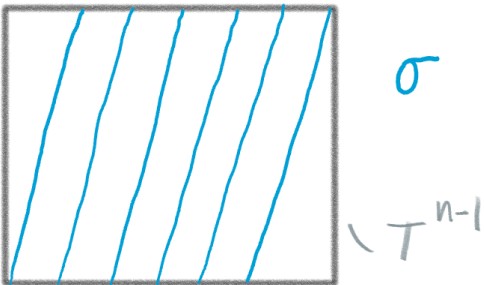

Figure 13: A long simple closed geodesic $\sigma$ (blue) in the cusp cross sectional $T^{n-1}$.

to the twisted Euclidean AdS black hole metric (equation (3.6) of [19])

$$ds_{BH}^2 = [\frac{dr^2}{V(r)} + V(r)d\theta^2 + r^2 ds_{\mathbb{R}^{n-2}}^2]/\mathbb{Z}^{n-2}\,, \tag{A.5}$$

with $\theta$ periodic with period $\beta = \frac{4\pi}{(n-1)}$ and

$$V(r) = r^2 \left(1 - \frac{1}{r^{n-1}}\right)\,. \tag{A.6}$$

The radial position $r_{join}$ is specified as follows[22]

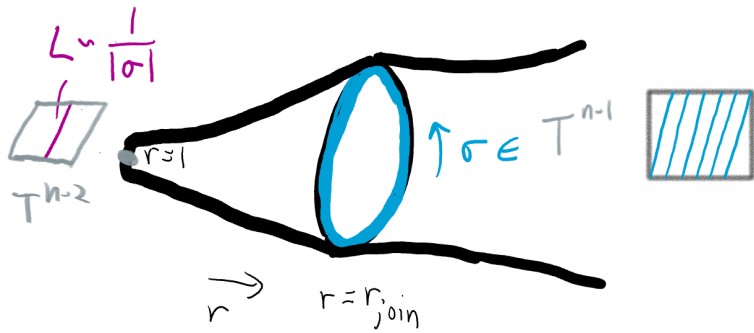

Figure 14: A schematic of the approximate filled metric in the construction [19], joined to the cusp at $r = r_{join}$, as reviewed in the text. A choice of long simple closed geodesic $\sigma$ provides a tunably small systole at the tip of the cigar. The joined metrics (A.4) and (A.5) have a corner at $r_{join}$ since $\frac{d\sqrt{V}}{dr}|_{r_{join}} > 1$, but one can smooth this out and perturb slightly to obtain an Einstein metric throughout [19].

$$\sqrt{V(r_{join})}\beta = |\sigma|\,, \tag{A.7}$$

---

[22]In the paper [19] this is denoted $R$, but we will use different notation to avoid confusion with the radius $R$ appearing in the main text.

so that the $\theta$ circle has proper size $|\sigma|$. In (A.5) the group $\mathbb{Z}^{n-2}$ acts so that the $T^{n-1}$ at $r = r_{join}$ is the $T^{n-1}$ of (A.4), with the $\theta$ circle being the simple closed geodesic $\sigma$. This works as follows. Since $\sigma$ is a simple closed geodesic, the $T^{n-1}$ may be constructed by modding out $\mathbb{R}^n$ by a $\mathbb{Z}^n$ group of translations generated by $\sigma$ and $n-2$ other generators $(b_2, \ldots, b_{n-1})$. This set of generators $(\sigma, b_2, \ldots, b_{n-1})$ can be obtained from any other set of generators, such as the simple orthonormal basis of unit vectors $\hat{e}_i$ for the square tori in [25], by an $SL(n-1, \mathbb{Z})$ transformation. As the length $|\sigma|$ grows, the $b_i$ grow in length so their integer linear combinations (in the relevant $SL(n-1, \mathbb{Z})$ transformations) yield the unit vectors $\hat{e}_i$ in that example (and similarly for more general cases).

The $\mathbb{Z}^{n-2}$ acts on the space $\left[\frac{dr^2}{V(r)} + V(r)d\theta^2 + r^2 ds^2_{\mathbb{R}^{n-2}}\right]$ in (A.5) as in equation (3.7) of [19], which we reproduce here. At each $r$, it acts on the $S^1_\theta \times \mathbb{R}^{n-2}$ as a $\mathbb{Z}^{n-2}$ with generators

$$\sigma(r) = \sqrt{\frac{V(r)}{V(r_{join})}}\,\sigma\,, \quad b_i(r) = b_i + \left(\sqrt{\frac{V(r)}{V(r_{join})}} - 1\right)\frac{b_i \cdot \sigma}{|\sigma|^2}\sigma\,. \tag{A.8}$$

We continuously join the two metrics at the radius $r_{join}$, albeit with a corner defect there. The generators (A.8) vary with $r$ giving a $T^{n-1}$ that changes shape as a function of $r$ and reaches a diameter of order $1/|\sigma|$ at the tip of the cigar $r = 1$ in the approximate filling metric (A.5) [19]. That is, we can tune the systole size smaller and smaller by choosing larger and larger simple closed geodesics:

$$\text{length(systole)} \sim \frac{\ell^2}{|\sigma|}\,. \tag{A.9}$$

It is immediately clear from (A.7) and (A.6) that the two metrics (A.4) and (A.5) agree in the range $1 \ll r \le r_{join}$ up to corrections of order $1/r^{n-1}$ (with exact agreement at $r = r_{join}$ by construction). The bulk of the paper [19] derives a correction to a smoothed out version of (A.5) that produces an Einstein metric without any corner defect at $r_{join}$. As just described, even the first step of joining the two metrics here produces for large $|\sigma|$ a space very close to the cusp, deviating from it where the $T^{n-1}$ becomes small $\sim 1/|\sigma|$.

Altogether, this construction enables fine tuning control of one combination of our model parameters as described in the main text around equation (5.21). As stressed in [19], this infinite set of Einstein metrics parallels the infinite set of Dehn filled hyperbolic metrics for $n = 3$. In our physical context, the infinity is cut off by quantum gravity effects (e.g. wrapped membrane effects) once the systole length reaches $\ell_{11}$. As described in the main text, keeping this length $R_c \gg \ell_{11}$ fits with our stabilization mechanism.

## B  Equations of motion

In this Appendix we give the equations of motion used in the main text. We take an $n$-dimensional internal space, and $d$ external space-time dimensions, with $D = d + n = 11$. The case relevant for most of the work is $d = 4, n = 7$, but for different applications it is useful to keep $(d, n)$ general.

Let us consider the equations of motion for a metric ansatz

$$\begin{aligned} ds^2 &= g^{(d)}_{\mu\nu}dx^\mu dx^\nu + g^{(n)}_{ij}dy^i dy^j \\ &= e^{2A(y)}ds^2_{\text{symm}} + e^{2B(y)}\tilde{g}^{(n)}_{ij}dy^i dy^j\,, \end{aligned} \tag{B.1}$$

where $ds^2_{\text{symm}}$ is a maximally symmetric $d$-dimensional spacetime, $g^{(n)}_{ij}$ is the metric in $n$ dimensions, and $A, B$ are the warp and conformal factors, respectively. We present first the $D$-dimensional equations, and then relate them to variations of the off-shell potential. The equations will be valid for general internal metric.

## B.1 Higher-dimensional equations of motion

The equations of motion for $A$ and $B$ follow from the $d$- and $n$-dimensional traces of

$$R_{MN} - \frac{1}{2}g_{MN}R = \frac{1}{2}(\ell_D)^{D-2}\left(T_{MN}^{(\text{flux})} + T_{MN}^{(\text{Cas})}\right), \quad T_{MN}^{(\text{matter})} = -\frac{2}{\sqrt{-g^{(D)}}}\frac{\delta S_{\text{matter}}}{\delta g^{MN}}, \tag{B.2}$$

with energy-momentum tensors from internal $n$-flux and the Casimir contribution. The flux gives

$$g^{\mu\nu}T_{\mu\nu}^{(\text{flux})} = -\frac{1}{(\ell_D)^{D-2}}\frac{d}{2}|F_n|^2, \quad g^{ij}T_{ij}^{(\text{flux})} = \frac{1}{(\ell_D)^{D-2}}\frac{n}{2}|F_n|^2. \tag{B.3}$$

Recalling (4.2), the Casimir contribution is

$$g^{\mu\nu}T_{\mu\nu}^{(\text{Cas})} = -d\,\rho_C(R_c), \tag{B.4}$$

$$g^{ij}T_{ij}^{(\text{Cas})} = -n\rho_C(R_c) - R_c\rho_C'(R_c) = d\,\rho_C(R_c), \tag{B.5}$$

where in the last step we used $\rho_C(R_c) \sim -R_c^{-d-n}$, valid when $R_c$ varies slowly in the internal space.

The $d$-dimensional equation

$$\frac{1}{\sqrt{-g^{(D)}}}\frac{\delta S}{\delta A} = -2\left(g^{\mu\nu}R_{\mu\nu} - \frac{d}{2}g^{MN}R_{MN}\right) + (\ell_D)^{D-2}T_\mu^\mu = 0 \tag{B.6}$$

then becomes

$$(d-2)e^{-2A}R_{\text{symm}}^{(d)} + dR^{(n)} - 2d(d-1)\nabla^2 A - d^2(d-1)(\nabla A)^2 - d(\ell_D)^{D-2}\rho_C(R_c) - \frac{d}{2}|F_n|^2 = 0. \tag{B.7}$$

This is proportional to the General Relativity Hamiltonian constraint in our setup with a maximally symmetric $d$-space (otherwise the constraint is just the 00 Einstein equation). The $n$-dimensional equation

$$\frac{1}{\sqrt{-g^{(D)}}}\frac{\delta S}{\delta B} = -2\left(g^{ij}R_{ij} - \frac{n}{2}g^{MN}R_{MN}\right) + (\ell_D)^{D-2}T_i^i = 0 \tag{B.8}$$

reads

$$ne^{-2A}R_{\text{symm}}^{(d)} + (n-2)R^{(n)} - 2d(n-1)\nabla^2 A - d(nd+n-2)(\nabla A)^2 + d(\ell_D)^{D-2}\rho_C(R_c) + \frac{n}{2}|F_n|^2 = 0. \tag{B.9}$$

Finally, the equation of motion for the flux reads

$$\partial_{i_1}\left(\sqrt{g^{(n)}}e^{dA}g^{i_1 j_1}\cdots g^{i_n j_n}F_{j_1\ldots j_n}\right) = 0. \tag{B.10}$$

The zero mode solution is

$$F_{i_1\ldots i_n} = f_0\,e^{-dA}\,\epsilon_{i_1\ldots i_n}, \tag{B.11}$$

and flux quantization fixes

$$\frac{1}{\ell_D^{n-1}}\int F_n \sim N_n \Rightarrow f_0 \sim \ell_D^{n-1}\frac{N_n}{\int d^n y\,\sqrt{g^{(n)}}e^{-dA}}, \quad N_n \in \mathbb{Z}. \tag{B.12}$$

We now give two equivalent forms of these equations that will be useful. In terms of the variable

$$u(y) \equiv e^{\frac{d}{2}A(y)}, \tag{B.13}$$

these equations become

$$(d-2)R^{(d)}_{\text{symm}}u^{-4/d} + d R^{(n)} - 4(d-1)\frac{\nabla^2 u}{u} - d(\ell_D)^{D-2}\rho_C(R_c) - \frac{d}{2}|F_n|^2 = 0, \qquad \text{(B.14)}$$

and

$$nR^{(d)}_{\text{symm}}u^{-4/d} + (n-2)R^{(n)} - 4(n-1)\frac{\nabla^2 u}{u} - 4\frac{d+n-2}{d}\frac{(\nabla u)^2}{u^2} + d(\ell_D)^{D-2}\rho_C(R_c) + \frac{n}{2}|F_n|^2 = 0 \tag{B.15}$$

respectively.

Lastly, let us give the equations taking into account explicitly the conformal mode $e^{2B}$ in (B.1), and using the fiducial metric $\tilde{g}^{(n)}_{ij}$. For this, we need the partial traces of the Ricci tensor

$$g^{\mu\nu}R_{\mu\nu} = e^{-2A}R^{(d)}_{\text{symm}} - de^{-2B}\left[d(\nabla_{\tilde{g}}A)^2 + \nabla^2_{\tilde{g}}A + (n-2)\nabla_{\tilde{g}}A\cdot\nabla_{\tilde{g}}B\right], \qquad \text{(B.16)}$$

$$g^{ij}R_{ij} = e^{-2B}\left[R^{(n)}_{\tilde{g}} - d\nabla^2_{\tilde{g}}A - d(\nabla_{\tilde{g}}A)^2 - 2(n-1)\nabla^2_{\tilde{g}}B - (n-1)(n-2)(\nabla_{\tilde{g}}B)^2 \right.$$
$$\left. - d(n-2)\nabla_{\tilde{g}}A\cdot\nabla_{\tilde{g}}B\right].$$

Here $\nabla_{\tilde{g}}$ is the covariant derivative with respect to $\tilde{g}^{(n)}_{ij}$, and $R^{(n)}_{\tilde{g}}$ is its Ricci scalar. The $d$-dimensional constraint then becomes

$$\begin{aligned}
0 &= (d-2)e^{-2A}R^{(d)}_{\text{symm}} + de^{-2B}R^{(n)}_{\tilde{g}} - de^{-2B}\left\{2(d-1)\nabla^2_{\tilde{g}}A + d(d-1)(\nabla_{\tilde{g}}A)^2 + 2(n-1)\nabla^2_{\tilde{g}}B\right.\\
&\quad + 2(d-1)(n-2)\nabla_{\tilde{g}}A\cdot\nabla_{\tilde{g}}B + (n-1)(n-2)(\nabla_{\tilde{g}}B)^2\left.\right\} - d(\ell_D)^{D-2}\rho_C(R_c) - \frac{d}{2}|F_n|^2.
\end{aligned} \tag{B.17}$$

For the internal $n$-dimensional trace, we obtain

$$\begin{aligned}
0 &= ne^{-2A}R^{(d)}_{\text{symm}} + (n-2)e^{-2B}R^{(n)}_{\tilde{g}} - (n-1)e^{-2B}\left\{2d\nabla^2_{\tilde{g}}A + \frac{d(nd+n-2)}{(n-1)}(\nabla_{\tilde{g}}A)^2 \right. \quad \text{(B.18)}\\
&\quad + 2(n-2)\nabla^2_{\tilde{g}}B + 2d(n-2)\nabla_{\tilde{g}}A\cdot\nabla_{\tilde{g}}B + (n-2)^2(\nabla_{\tilde{g}}B)^2\left.\right\} + d(\ell_D)^{D-2}\rho_C(R_c) + \frac{n}{2}|F_n|^2.
\end{aligned}$$

As a check, combining these two equations to eliminate $R^{(d)}_{\text{symm}}$, and approximating $A$ and $B$ as constants, reproduces the extremum of (2.3).

## B.2 Effective potential

Let us now consider the dimensionally reduced theory,

$$\mathcal{S}_d = \int d^d x \sqrt{-g^{(d)}}\left(\frac{1}{G_N}R^{(d)} - 2V_{eff}\right). \tag{B.19}$$

The off-shell potential [17] is identified from the integrated GR constraint (B.14), after adding a Lagrange multiplier $C$ that fixes the value of the $d$-dimensional Newton's constant:

$$\begin{aligned}
V_{eff} &= \frac{1}{2\ell_D^{D-2}}\int d^n y \sqrt{g^{(n)}}u^2\left(-R^{(n)} - 4\frac{d-1}{d}\frac{(\nabla u)^2}{u^2} - \frac{1}{d}(\ell_D)^{D-2}T^{\mu}_{\mu}\right)\\
&\quad + \frac{1}{2}C\left(\frac{1}{G_N} - \frac{1}{\ell_D^{D-2}}\int d^n y \sqrt{g^{(n)}}u^{2-4/d}\right). \tag{B.20}
\end{aligned}$$

This requires $T^\mu_\mu$ to be independent of $u(y)$. This is satisfied in our case,

$$-(\ell_D)^{D-2}\frac{1}{d}T^\mu_\mu = (\ell_D)^{D-2}\rho_C(R_c) + \frac{1}{2}|F_n|^2. \tag{B.21}$$

The variations

$$\frac{\delta V_{eff}}{\delta u} = 0 \,, \quad \frac{\delta V_{eff}}{\delta B} = 0 \,, \quad \frac{\delta V_{eff}}{\delta C^0} = 0 \tag{B.22}$$

reproduce the equations of motion of §B.1, with the identification $C = R^{(d)}_{\text{symm}}$. The value of the potential on the $d$-dimensional constraint, after fixing $G_N$ with the Lagrange multiplier $C$, becomes

$$V_{eff} = \frac{d-2}{2d}\frac{C}{G_N}. \tag{B.23}$$

More generally, however, we need to understand how $V_{eff}$ can reproduce all the internal equations, given that $V_{eff}$ depends on $T^\mu_\mu$, while a $T_{ij}$ needs to appear in order to match the internal Einstein equations. One needs

$$T_{ij} = -\frac{2}{d}\left(\frac{\delta T^\mu_\mu}{\delta g^{ij}} - \frac{1}{2}g_{ij}T^\mu_\mu\right). \tag{B.24}$$

If the matter action depends on the $d$-dimensional metric only through the volume element,

$$S_{matter} = \int d^D x \sqrt{-g^{(d)}}\sqrt{g^{(n)}}\,L_{matter}(g_{ij}, \phi, \ldots), \tag{B.25}$$

then

$$T_{\mu\nu} = -\frac{2}{\sqrt{-g^{(D)}}}\frac{\delta S_{matter}}{\delta g^{\mu\nu}} = g_{\mu\nu}L_{matter} \;\Rightarrow\; T^\mu_\mu = dL_{matter}. \tag{B.26}$$

In this case, (B.24) is satisfied because it gives the usual definition for the internal stress tensor:

$$T_{ij} = -2\left(\frac{\delta L_{matter}}{\delta g^{ij}} - \frac{1}{2}g_{ij}L_{matter}\right). \tag{B.27}$$

From this, we see that if $L_{matter}$ depends nontrivially on $g_{\mu\nu}$, (B.24) will not be satisfied. Examples include higher curvature corrections, or more general quantum effects. The sources we have used in this work (with the approximation where Casimir is slowly varying and dominated by $R_c$) satisfy the criterion (B.25).

## C  Slow Roll parameters from the off-shell potential

In this section, we will derive a formula for contributions to $\varepsilon_V$ from certain metric deformations (reducing to fields $\phi_{c,I}$). We will ultimately focus on a region of the cusp where we will apply our formula, including the four dimensional field corresponding to the level of asymmetry in the cusp cross section. But we will start more generally.

Let us consider the system

$$ds^2 = u(t, y)ds^2_{dS_4} + g^{(7)}_{ij}(t, y)dy^i dy^j, \tag{C.1}$$

and expand the system about a fiducial metric $\bar{g}^{(7)}_{ij}(y)$. Here, we allow for time and internal coordinate dependence in deformations away from the fiducial metric, in order to capture the kinetic normalization required for the four dimensional fields $\phi_{c,I}$ (3.28) as well as the

internal gradients. Within a portion of the cusp, we will be interested in a particular set of deformations captured by the metric

$$
\begin{aligned}
ds^2 &= u(t,\vec{y})ds^2_{dS_4} + e^{2\sigma_{rad}(t,\vec{y})}dy^2_{rad} + R(t,\vec{y})^2\sum_{i=1}^{5}dy^2_{\perp i} + R_c(t,\vec{y})^2 dy^2_c \\
&= e^{2A(t,\vec{y})}ds^2_{dS_4} + e^{2\sigma_{rad}(t,\vec{y})}dy^2_{rad} + e^{2\sigma(t,\vec{y})}\sum_{i=1}^{5}dy^2_{\perp i} + e^{2\sigma_c(t,\vec{y})}dy^2_c,
\end{aligned}
\tag{C.2}
$$

with

$$
\sigma^I = \bar{\sigma}^I(y) + \Delta\sigma^I(t,y)
\tag{C.3}
$$

describing an expansion about a fiducial metric. The asymmetry relevant to our setup is captured by the ratio $R_c/R$. In what follows, we will expand $\Delta\sigma^I$ in a basic of modes $\Phi_k$, defining an appropriate normalization to extract the $\phi_{c,I}$.

To begin, we need to generalize the derivation of $V_{eff}$ in [17] to capture the kinetic terms for the four dimensional fields. As in [17], this proceeds essentially by inserting into the eleven-dimensional action $S$ the solution to the equation of motion for the warp factor (the constraint). The action $S$ contains the 11$d$ Einstein-Hilbert terms, which generates the kinetic terms for the metric deformations. We need to replace the $\partial_\mu A$'s in the kinetic action with the solution for $A$. There is a simple way to obtain that, using the fixed Newton constant, as follows. From (3.4), in $V_{eff}$ we have the term

$$
\frac{C}{2}\left(\frac{1}{G_N} - \int d^7y \sqrt{g^{(7)}}u|_c\right),
\tag{C.4}
$$

with $C$ a Lagrange multiplier. Once we include dependencies on the four-dimensional coordinates $x^\mu$ (in particular time $t$ (C.1)(C.2)), a consistency requirement is

$$
\partial_\mu(\sqrt{g^{(7)}}u|_c) = 0,
\tag{C.5}
$$

enabling the substitution

$$
\partial_\mu A = -\frac{1}{2\sqrt{g^{(7)}}}\partial_\mu(\sqrt{g^{(7)}}).
\tag{C.6}
$$

In the particular metric (C.2), this becomes

$$
\partial_\mu(\Delta\sigma_{rad} + 5\Delta\sigma + \Delta\sigma_c + 2A) = 0.
\tag{C.7}
$$

This enables us to replace $\partial_\mu A$ in the action, giving positive kinetic terms for the four-dimensional fields (e.g. $\sigma^I$). For example, the kinetic term for the internal conformal factor $B$ becomes positive once we incorporate this substitution.

This results in an effective 4d theory of the following form check notation

$$
S = \frac{1}{2G_N}\int \sqrt{-g^{(4)}_{symm}}e^{2A+\sigma_{rad}+5\sigma+\sigma_c}\left(R^{(4)}_{symm} - G_{IJ}\partial_\mu\Delta\sigma^I\partial^\mu\Delta\sigma^J\right) - \int \sqrt{-g^{(4)}_{symm}}V_{eff},
\tag{C.8}
$$

with $G_{IJ}$ a positive symmetric constant matrix.

The next step is to expand the fields in a basis of functions $\Phi_k$.

$$
\Delta\sigma^I(x,y) = \sum_k \Delta\sigma^I_k(x)\Phi_k(y),
\tag{C.9}
$$

with completeness relations

$$e^{2A}\sqrt{g^{(7)}} \sum_k \Phi_k(y)\Phi_k(y') = \frac{1}{G_N},$$

$$\int d^7 y \, e^{2A}\sqrt{g^{(7)}} \Phi_k(y)\Phi_{k'}(y') = \frac{1}{G_N}\delta_{kk'}. \tag{C.10}$$

Given this, the $\sigma^I$ fields that we have defined (C.2), (C.3) contribute to $\varepsilon_V$ as

$$\varepsilon_{V,\sigma} = \sum_k \frac{1}{2V_{eff}^2} G^{IJ} \partial_{\Delta\sigma_k^I} V_{eff} \, \partial_{\Delta\sigma_k^J} V_{eff}. \tag{C.11}$$

A similar formula would apply to a full set of internal fields deforming away from any fiducial metric $\bar{g}_{ij}^{(7)}$ in (C.1) (related to $\Delta B, \Delta h, C_6$ in the decomposition (1.1)). In general this is given by a Kaluza-Klein reduction on the fiducial space [59, 60].

The derivatives $\partial_{\Delta\sigma_k^I} V_{eff}$ give the equations of motion for a de Sitter ansatz for our compactification. A nonzero value corresponds to a tadpole, and contributes to $\varepsilon_V$. We can use our formula for $\varepsilon_V$ to quantify how close to accelerated expansion any such configuration is. To make this more explicit, our next step is to make the definition

$$V_{eff} = \int d^7 y \sqrt{g_{fiducial}^{(7)}} \, \mathcal{L}_{eff}(g^{(7)}, C_6)$$
$$= \int d^7 y \sqrt{g_{fiducial}^{(7)}} \, \mathcal{L}_{eff}(\delta B, h, C_6) \to \int d^7 y \mathcal{L}_{eff}(\sigma^I), \tag{C.12}$$

where the last expression is the reduction to the case (C.2).

We will need the derivatives entering into $\varepsilon_V$ (C.11), computing for the cusp case

$$\frac{\partial V_{eff}}{\partial \Delta\sigma_k^I(x)} = \int d^n y \, \frac{\partial \mathcal{L}_{eff}}{\partial \Delta\sigma^I(x,y)} \frac{\partial \Delta\sigma^I}{\partial \Delta\sigma_k^I(x)}$$
$$= \int d^n y \, \frac{\partial \mathcal{L}_{eff}}{\partial \Delta\sigma^I(x,y)} \Phi_k(y). \tag{C.13}$$

Plugging back into (C.11) gives

$$\varepsilon_{\Delta\vec{\sigma}} = G^{IJ} \frac{1}{2V_{eff}^2} \sum_k \int d^n y \, d^n y' \, \frac{\partial \mathcal{L}_{eff}}{\partial \Delta\sigma^I(x,y)} \frac{\partial \mathcal{L}_{eff}}{\partial \Delta\sigma^J(x,y')} \Phi_k(y)\Phi_k(y'). \tag{C.14}$$

Applying the completeness relation (C.10) yields

$$\varepsilon_{\Delta\vec{\sigma}} = \frac{1}{2G_N} \frac{1}{V_{eff}^2} \int d^n y \, e^{-2A-\sigma_{rad}-5\sigma-\sigma_c} G^{IJ} \frac{\partial \mathcal{L}_{eff}}{\partial \Delta\sigma^I} \frac{\partial \mathcal{L}_{eff}}{\partial \Delta\sigma^J}, \tag{C.15}$$

where $V_{eff}$ is given above in (3.4).

It is useful to note the scaling with the exponentials of $A$ and $\vec{\sigma}$, and its dependence on the domain in which $\varepsilon_V$ has support. For this purpose, we collect these scalings from the formulas for the factors in (C.15), working in our case of interest $d = 4, n = 7$.

$$\frac{1}{G_N} = \int e^{2A}\sqrt{g^{(7)}} \to \int d^7 y \, e^{2A+\sigma_{rad}+5\sigma+\sigma_c}, \tag{C.16}$$

$$V_{eff} = \int d^7 y \, \mathcal{L}_{eff}(\vec{\sigma}) = \int \frac{d^7 y}{\ell_{11}^9} e^{4A+\sigma_{rad}+5\sigma+\sigma_c}\left(-R^{(7)} + \cdots - \frac{1}{2}Ce^{-2A}\right), \tag{C.17}$$

and we recall that $R^{(7)}$ contains terms scaling like $e^{-2\sigma^I}$. The derivatives $\frac{\partial \mathcal{L}_{eff}}{\partial \Delta \sigma^I}$ scale with the exponentials like the integrand of this expression (C.17). Finally, we note that $V_{eff}$ itself is simply given by (3.8),

$$V_{eff} = \frac{C}{2G_N}. \tag{C.18}$$

It is interesting to express the scaling in two cases. First, suppose that the $C/u = Ce^{-2A}$ term is (at least) of order the other terms in the expressions, as occurs without any particular tuning or if the $C$ term itself is dominant. Then, putting together these pieces (C.14)(C.16)(C.17) yields the form

$$\varepsilon_{\vec{\sigma}} \sim \frac{\int_{\Sigma_\varepsilon} d^7y \, u \, e^{\sigma_{rad}+5\sigma_\perp+\sigma_c}}{\int_{\Sigma_7} d^7y \, u \, e^{\sigma_{rad}+5\sigma_\perp+\sigma_c}} \quad C/u \; term, \tag{C.19}$$

for $\varepsilon$, where $\Sigma_\varepsilon$ denotes the domain where the equations of motion are not solved (so we get contributions to $\varepsilon$), and $\Sigma_7$ is the entire 7-manifold which contributes to $1/G_N$.

If the $C/u$ term is subdominant, on the other hand (as in the tuning of interest to obtain small 4d Hubble), the scaling of $\varepsilon$ with exponentials and volumes based on the internal curvature term is

$$\varepsilon_{\vec{\sigma}} \sim \frac{\int_{\Sigma_\varepsilon} d^7y \, e^{6A+\sigma_{rad}+5\sigma_\perp+\sigma_c} \, e^{-4\sigma_*}}{(C\ell^2)^2 \int_{\Sigma_7} d^7y \, e^{2A+\sigma_{rad}+5\sigma_\perp+\sigma_c}} \quad R^{(7)} \; terms, \tag{C.20}$$

where $\sigma_*$ is the component of $\vec{\sigma}$ that gives the strongest contribution among the terms in $R^{(7)}$ (all of which scale like $e^{-2\sigma_I}$ for some $I$). These formulas are useful for estimating epsilon for smooth internal configurations where the equations of motion are solved almost everywhere, but for which some region $\Sigma_\varepsilon$ contributes to $\varepsilon_V$. This enables a functional generalization of the analysis of [66].

# D   Derivation of no go theorems

In this Appendix we derive the inequalities governing general compactifications of $D$-dimensional gravitational theories down to $d$-dimensional vacua. As we are going to see, it is possible to constrain the $d$-dimensional cosmological constant in terms of integrated stress-energy tensors. [64–66, 131]. In the interest of making this Appendix self-contained, we repeat here the main definitions. We work in $D$-dimensional Einstein frame, such that the $D$-dimensional Einstein equations are

$$R_{MN} - \frac{1}{2}g_{MN}R = \kappa^2 \frac{1}{2}T_{MN}, \tag{D.1}$$

where $T_{MN} \equiv -\frac{2}{\sqrt{-g}}\frac{\delta S_{\text{matter}}}{\delta g^{MN}}$ and $\kappa^2 = \ell_D^{(D-2)}$. Since we want to relate the $d$-dimensional curvature directly to the matter content, we take the trace of equation (D.1) to eliminate the $D$-dimensional Ricci scalar:

$$R\left(1 - \frac{D}{2}\right) = \frac{1}{2}\kappa^2 T \quad \Rightarrow \quad R_{MN} = \frac{1}{2}\kappa^2 T_{MN} + \frac{1}{2}\kappa^2 g_{MN}\frac{T}{2-D}. \tag{D.2}$$

We now specialize this equation to space-times of the form

$$ds_D^2 = e^{2A}ds_{\text{symm}}^2 + ds_{D-d}^2, \tag{D.3}$$

where $A$ only depends on the $(D-d)$-dimensional coordinates and $ds_{\text{symm}}^2$ is a $d$-dimensional vacuum. In particular, denoting the $d$-dimensional indices by $\mu, \nu, \ldots$ this decomposition re-

sults in

$$R_{\mu\nu} = R_{\mu\nu}^{(d)} - e^{2A}g_{\mu\nu}^{(d)}(d(\nabla A)^2 + \nabla^2 A) \tag{D.4}$$

$$= R_{\mu\nu}^{(d)} - \frac{1}{d}e^{(2-d)A}g_{\mu\nu}^{(d)}\nabla^2(e^{d\,A}), \tag{D.5}$$

where $\nabla$ is the derivative with respect to $g_{D-d}$. Plugging it in (D.2) and specializing to the space-time components we get

$$R_{\mu\nu}^{(d)} = \frac{1}{2}\kappa^2 T_{\mu\nu} + g_{\mu\nu}^{(d)}\left(\frac{1}{d}e^{(2-d)A}\nabla^2(e^{dA}) + e^{2A}\frac{1}{2}\kappa^2\frac{T}{2-D}\right). \tag{D.6}$$

Tracing it with $g_{\text{symm}}$, we obtain

$$R^{(d)} = \frac{1}{2}\kappa^2 T^{(d)} + e^{(2-d)A}\nabla^2(e^{dA}) + \frac{1}{2}\kappa^2\frac{d}{2-D}e^{2A}T. \tag{D.7}$$

Finally, making use of the definitions

$$T = g^{MN}T_{MN} \tag{D.8}$$

$$= e^{-2A}g_{(d)}^{\mu\nu}T_{\mu\nu} + g^{ij}T_{ij} \tag{D.9}$$

$$\equiv e^{-2A}T^{(d)} + T^{(D-d)}, \tag{D.10}$$

we can rewrite

$$R^{(d)} = \frac{1}{2}\kappa^2\left(1 + \frac{d}{2-D}\right)T^{(d)} + e^{(2-d)A}\nabla^2(e^{dA}) + \frac{d}{2-D}e^{2A}\frac{1}{2}\kappa^2 T^{(D-d)}. \tag{D.11}$$

Multiplying this equation by $e^{(d-2)A}$ and integrating in the internal space, assuming the internal space is smooth and without boundaries, we obtain

$$\frac{1}{G_N}R^{(d)} = I_{\text{tot}}, \tag{D.12}$$

where we have defined the $d$-dimensional Newton constant as $\frac{1}{G_N} \equiv \frac{1}{\kappa^2}\int\sqrt{g_{D-d}}e^{(d-2)A}$ and the integrated quantity

$$I_{\text{tot}} \equiv \frac{1}{2}\int\sqrt{g_{D-d}}e^{(d-2)A}\left(\left(1 + \frac{d}{2-D}\right)T^{(d)} + \frac{d}{2-D}e^{2A}T^{(D-d)}\right). \tag{D.13}$$

Generically, both quantum and classical terms will contribute to the traces of stress energy tensors. If they do not mix (D.12) can be decomposed as

$$\frac{1}{G_N}R^{(d)} = I_{\text{classical}} + I_{\text{quantum}}. \tag{D.14}$$

The classical no-go theorems [64–66] are obtained by neglecting the $I_{\text{quantum}}$ contribution. For example, in a generic M-theory setting (without localized sources), $I_{\text{classical}}$ is given by

$$I_{\text{classical}} = I_{F_4} = \frac{1}{6}\int\sqrt{g_{11-d}}(e^{(d-8)A}(12-d)\hat{f}_4^2 - de^{dA}f_4^2) \leq 0, \tag{D.15}$$

where we are working with $F_4 = \star F_7$ and we have decomposed $F_4 = \hat{f}_4 + f_4$, with $\hat{f}_4$ being the space-time component (which is non-vanishing only for $d = 4$). The last inequality follows from $\hat{f}_4^2 \leq 0$ [23]. From (D.15) we see that without other energy sources $dS_d$ compactifications

---

[23]In the main text we are in the situation $d = 4, f_4 = 0, |F_7|^2 = -e^{-8A}\hat{f}_4^2$.

are forbidden. In the main text,we have shown how the quantum mechanically generated Casimir energy evade these restriction.

Localized sources also contribute to the integral $I_{\text{tot}}$. However, since their backreaction on the internal geometry can be severe, the assumptions that lead us to (D.12) are not generically valid and have to be analyzed with care. This becomes particularly important for some negative-tension sources, such as O$p$-planes in String Theory. For these objects, the very strong backreaction near their core excises a finite region of space-time in the supergravity approximation, and the assumptions of a smooth internal geometry with no boundary is false. Mathematically, they evade in this way the no-go theorems, an effect which is physically ascribed to their negative tension.[24]

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
