# Peer review of "Hyperbolic compactification of M-theory and de Sitter quantum gravity"

_SciPost Physics, doi:SciPost Phys. 12, 083 (2022)_

## Round 2 · Referee Report · Anonymous · 2021-11-2

Report
This is a significant paper constituting serious progress for the arduous task of constructing well-controlled string/M-theory vacua realizing four-dimensional meta-stable de Sitter space with a small positive cosmological constant. I recommend publication after minor revision. For the full report, please see the uploaded PDF file.
Author: Gonzalo Torroba on 2022-02-02 [id 2144]
(in reply to Report 1 on 2021-11-02)
We thank the referee for his careful and encouraging report. In the new version we have corrected the typos found by the referee. We have also addressed the bullet points as follows.
Regarding the first point on $B_\star$, we should stress that the parameter that controls it, $y_\star$, is a variational parameter. It should be chosen in order to minimize the wavefunction Hamiltonian (which maximizes the effective potential). In this sense there is no tuning here; finding the variational wavefunction that gives the largest effective potential puts the strongest bound on the exact solution. We have emphasized this in the new footnote 13.
Regarding the second bullet point, we stress two aspects. First, in order to avoid potential confusions and emphasize that our mechanism uses a large $K/a$, we have introduced the parameter $a$ from the beginning in Sec. 2. We refer there to the later Sec. 4 where we argue that it is possible to obtain $a\ll1$ with explicit discrete parameters in our construction. This decreases the relative strength of the curvature contribution, and allows the Casimir energy to compete without requiring sub-Planckian Casimir circles. This can be also confirmed in the backreacted cusp solution presented in Sec. 5.3. As explained around equation (5.68), it is possible to obtain a competitive Casimir energy with a large Casimir circle in Planck units, $\hat{R}_c \gg 1$. Since the geometry is smooth, higher-derivative curvature invariants are suppressed.
Our treatment of Axion Monodromy indeed leaves out the exit dynamics, as noted by the sentence `One aspect that requires further study is the exit from inflation.' We agree with the referee that this is a very interesting question, and that as they say it may be possible to incorporate additional couplings. One point to note intrinsic to the existing setup is that the axion $\int_{\Sigma_I} C_3$ and 4-form flux $\int_{\Sigma_4}F_4$ may be supported on cycles that are localized compared to the volume form. Although topologically $C_3\wedge F_4$ and $F_7$ are the same (deformable into each other since there is only one element of 7-cohomology), it is not clear to us that there isn't an energetic barrier to fully cancelling the $F_7$ using the $C_3\wedge F_4$ term -- deforming one into the other may require activating KK modes that cost energy. Be this as it may, we are not prepared to present a complete exit mechanism. So although we completely agree with the referee that this is important, we would like to defer this to future work (as indicated in the paper) to be able to give it the attention it deserves.
We have added in the conclusion section some comments on possible research directions for obtaining a realistic matter content. Since many of the existing constructions of this in other classes of compactifications are local, it appears possible to incorporate them here as well as potentially new mechanisms related to the geometry of hyperbolic manifolds and orbifolds. Not having delved into this in great detail yet, we still leave it as a future direction in this paper.
Author: Gonzalo Torroba on 2022-02-02 [id 2145]
(in reply to Report 2 on 2021-11-23)We thank the referee for the careful analysis of our work and comments. We have addressed the points in the new version, and here we reply explicitly to each point.
First let us make a brief comment on what is new here compared to previous negative-curvature compactifications, related to the referee's remark that ``the main new ingredient is the Casimir energy supported near small closed geodesics.''
The negative contribution (a role now played by Casimir energy) is indeed crucial and new. But we would like to stress that the role of rigidity of generic negatively curved spaces -- in addition to their positive contribution to the potential energy useful for dS and for moduli stabilization -- is also new to this class of examples, as is the direct use of the stabilizing warp factor dynamics. (Another simplification arises from the 11d SUGRA starting point with no dilaton.)
Regarding point 1), we have highlighted in a new Sec. 4.1 the explanation of how to obtain $a \ll 1$; see also the Appendix A. The parameters enabling this tuning are generically available in the setup of our work.
2) One useful toy example of the effects of the rigidity is the following. Consider first a partial cusp, \begin{equation} dy^2 + e^{-2y/\ell} d\vec x^2, ~~~~ 0\le y \le y_c \end{equation} with $\vec x$ living on a torus. This space has moduli (the moduli of the torus which preserve the volume of the space). However, a smooth finite-volume hyperbolic manifold has no such moduli. In the standard construction of such spaces as gluings of polygons, the cusps are connected to the bulk of the space along a totally geodesic submanifold, a particular region of $\mathbb{H}_{n-1}$ (as in figures 8, 10, 11), and there are no continuous parameters available in this construction; moreover, the Hessian is fully positive (see e.g. the Besse reference). As a result, the would-be moduli in the cross-sectional shape of the torus are lifted: the boundary conditions disallow those modes, and the allowed modes have positive gradient energy. This effect extends to our backreacted cusp in the sense that our numerical results in the near-cusp region have not revealed a tachyonic mode, and gluing this to the bulk will tend to increase all masses due to rigidity effects.
3) This is an important question and related to point 1) above. As we explain in the material now better highlighted in the newly created Sec. 4.1, given the wide availability of geometric and flux parameters, we present explicit discrete moves that are available among these parameters, enabling a tuning of our $a$ parameter. This, along with small tadpoles estimated in section 5, indicates that it is very likely that it is possible to parametrically control all the length scales in the full construction. Another hint for this parametric control comes from the parameter $c$ that acts on the bakcreacted cusp solution presented in Sec.~5.3. This acts both on the cusp geometry and on the flux by rescaling the length scales and the flux quanta in a way compatible with the general parametric estimates in Eq. (2.7)-(2.9) - as is also discussed below Eq.~(5.43) in the context of the auxiliary Schr\"odinger problem. This parameter of the local solution will ultimately be related to the available parameters of the core manifold to which the cusp region is glued.

---

## Round 2 · Referee Report · Anonymous · 2021-11-23

Report
In this paper, the authors proposed a scenario for constructing (meta)stable de Sitter vacua from M theory using the interplay of fluxes, curvature (hyperbolic manifold), and Casimir energy. Earlier forms of this scenario have already appeared in previous works of one of the authors including arXiv:hep-th/0411271 and arXiv:0712.1196 [hep-th]. The main new ingredient is the Casimir energy supported near small closed geodesics. The other referee has already summarized the results of this paper in detail so I do not need to repeat them here. The paper contains some interesting results and contribute to the ongoing discussions whether controlled constructions of de Sitter vacua can be found in string theory. I recommend the paper for publication but I would like the authors to address the following questions:
1) Crucial to their scenario is that the Casimir energy can give significant competing contributions. This criterion can be expressed in terms of a<<1 in their equation (4.6). If the authors can argue how likely this tuning can be realized, it would help in assessing the robustness of this scenario. For example, in other de Sitter constructions widely discussed in the literature, the Casimir energy is negligible. Does their proposed scenario only work in a contrived subset of hyperbolic manifolds? This would affect the reasoning of the landscape. For example, the Bousso-Polchinski type argument relies on having a finely-spaced discretum.
2) The authors gave some argument why the Hessian expanding around the dressed hyperbolic metric has mostly positive eigenvalues. However, a single negative eigenvalue would spoil the scenario. It’d be useful if the authors can work out one concrete model (albeit a toy one) to demonstrate that the solution is metastable.
3) Related to 1 and 2 above is whether there is a de Sitter vacuum with parametric control coming out from this construction. Having parametric control would be a real advance. If the authors can comment on whether i) they have demonstrated parametric control, or ii) they believe it is plausible but they have not been able to show that yet, or iii) it is unlikely to give parametric control within their proposed scenario, it would be useful for the readers.
I found the paper to be well written, and contains interesting results. The scenario is a bit convoluted, so I do not know if it works in details. For this reason, it would useful if the authors can comment on the above points. The results may be of interest to the broader community interested in mechanisms for de Sitter constructions.
If the authors can address the above questions, I can recommend this paper for publication in SciPost.

---

## Round 3 · Author Response

List of changes
1) We have emphasized the variational nature of the parameter $B_\star$ in the new footnote 13. 2) In order to avoid potential confusions and emphasize that our mechanism uses a large $K/a$, we have introduced the parameter $a$ from the beginning in Sec. 2. We refer there to the later Sec. 4 where we argue that it is possible to obtain $a\ll1$ with explicit discrete parameters in our construction. This decreases the relative strength of the curvature contribution, and allows the Casimir energy to compete without requiring sub-Planckian Casimir circles. 3) We have added in the conclusion section some comments on possible research directions for obtaining a realistic matter content. Since many of the existing constructions of this in other classes of compactifications are local, it appears possible to incorporate them here as well as potentially new mechanisms related to the geometry of hyperbolic manifolds and orbifolds. Not having delved into this in great detail yet, we still leave it as a future direction in this paper.

---

## Round 3 · List of Changes

1) We have emphasized the variational nature of the parameter $B_\star$ in the new footnote 13. 2) In order to avoid potential confusions and emphasize that our mechanism uses a large $K/a$, we have introduced the parameter $a$ from the beginning in Sec. 2. We refer there to the later Sec. 4 where we argue that it is possible to obtain $a\ll1$ with explicit discrete parameters in our construction. This decreases the relative strength of the curvature contribution, and allows the Casimir energy to compete without requiring sub-Planckian Casimir circles. 3) We have added in the conclusion section some comments on possible research directions for obtaining a realistic matter content. Since many of the existing constructions of this in other classes of compactifications are local, it appears possible to incorporate them here as well as potentially new mechanisms related to the geometry of hyperbolic manifolds and orbifolds. Not having delved into this in great detail yet, we still leave it as a future direction in this paper.

---

## Editorial Decision

published